# The staircase property:
# How hierarchical structure can guide deep learning

**Emmanuel Abbe**
EPFL
emmanuel.abbe@epfl.ch

**Enric Boix-Adsera**
MIT
eboix@mit.edu

**Matthew Brennan**
MIT

**Guy Bresler**
MIT
guy@mit.edu

**Dheeraj Nagaraj**
MIT
dheeraj@mit.edu

## Abstract

This paper identifies a structural property of data distributions that enables deep neural networks to learn hierarchically. We define the "staircase" property for functions over the Boolean hypercube, which posits that high-order Fourier coefficients are reachable from lower-order Fourier coefficients along increasing chains. We prove that functions satisfying this property can be learned in polynomial time using layerwise stochastic coordinate descent on regular neural networks – a class of network architectures and initializations that have homogeneity properties. Our analysis shows that for such staircase functions and neural networks, the gradient-based algorithm learns high-level features by greedily combining lower-level features along the depth of the network. We further back our theoretical results with experiments showing that staircase functions are learnable by more standard ResNet architectures with stochastic gradient descent. Both the theoretical and experimental results support the fact that the staircase property has a role to play in understanding the capabilities of gradient-based learning on regular networks, in contrast to general polynomial-size networks that can emulate any Statistical Query or PAC algorithm, as recently shown.

## 1 Introduction

It has been observed empirically that neural networks can learn hierarchically. For example, a 'car' may be detected by first understanding simpler concepts like 'door', 'wheel', and so forth in intermediate layers, which are then combined in deeper layers (c.f. [1, 2]). However, on the theoretical side, the mechanisms by which such hierarchical learning occurs are not yet fully understood. In this paper we are motivated by the following question:

*Can we identify naturally structured and interpretable classes of hierarchical functions, and show how regular[1] DNNs are able to learn them?*

This is a refinement of the generic objective of trying to understand DNNs: We identify several key desiderata for any theoretical result in this direction. (1) *Natural structure*: We aim to capture naturally occurring data of interest, so the structural assumption must make conceptual sense. (2) *Interpretability*: If we hope to clearly interpret the inner workings of neural networks, understanding both how they classify and also how they learn, then we need a model for data that is interpretable to

---

[1]The notion of regularity is specified in Definition 2.1; this means network architectures and initializations that have homogeneity properties within layers, in contrast to the emulation architectures in [3, 4].

35th Conference on Neural Information Processing Systems (NeurIPS 2021).

begin with. Interpretation of the representations occurring within a neural network is most clearly expressed with respect to structural properties of the data. Finally, (3) *Regularity of the network*: The network architecture and initialization should be symmetric in a sense defined later on. This prevents using carefully-crafted architectures and initializations to emulate general learning algorithms [3, 4]. We view this type of restriction as being partway towards considering practical neural networks that learn in a blackbox fashion. The results in this paper aim to satisfy all three high-level objectives. The relation with prior work is discussed in Section 1.1.

This paper proposes a new structurally-defined class of hierarchical functions and proves guarantees for learning by regular neural networks. In order to describe this structure, we first recall that any function $f : \{+1, -1\}^n \to \mathbb{R}$ can be decomposed in the Fourier-Walsh basis as

$$f(x) = \sum_{S \subseteq [n]} \hat{f}(S)\chi_S(x), \quad \text{where} \quad \hat{f}(S) := \langle f, \chi_S \rangle, \quad \chi_S(x) := \prod_{i \in S} x_i \tag{1}$$

and the inner product between two functions is $\langle f, g \rangle = \mathbb{E}f(X)g(X)$ for $X \sim \text{Unif}(\{+1, -1\}^n)$. This decomposition expresses $f(x)$ as a sum of components, each of which is a monomial $\chi_S(x)$, weighted by the Fourier coefficient $\hat{f}(S)$. Our definition of hierarchical structure is motivated by an observation regarding two closely related functions, "high-degree monomials" and "staircase functions", the latter of which can be learned efficiently and the former of which cannot.

**Monomials with no hierarchical structure**    The class of monomials of any degree $k$ where $k \leq n/2$ (i.e., the class $\{\chi_S\}_{S \subseteq [n], |S|=k}$) is efficiently learnable by Statistical Query (SQ) algorithms if and only if $k$ is constant [5, 6], and the same holds for noisy Gradient Descent (GD) on neural nets with polynomially-many parameters [5], and for noisy Stochastic Gradient Descent (SGD) where the batch-size is sufficiently large compared to the gradients' precision [3, 4]. This was also noted in [7] which shows that gradients carry little information to reconstruct $\chi_S$ for large $|S|$, and hence gradient-based training is expected to fail. Thus, we can think of a component $\chi_S$ as *simple* and easily learnable if the degree $|S|$ is small and *complex* and harder to learn if the degree $|S|$ is large.

**Staircase functions with hierarchical structure**    Now, instead of a single monomial, consider the following *staircase* function (and its orbit class induced by permutations of the inputs), which is a sum of monomials of increasing degree:

$$S_k(x) = x_1 + x_1 x_2 + x_1 x_2 x_3 + x_1 x_2 x_3 x_4 + \cdots + \chi_{1:k}. \tag{2}$$

Here $S_k(x)$ has a hierarchical structure, where $x_1$ builds up to $x_1 x_2$, which builds up to $x_1 x_2 x_3$, and so on until the degree-$k$ monomial $\chi_{1:k}$. Our experiments in Fig. 2 show a dramatic difference between learning a single monomial $\chi_{1:k}$ and learning the staircase function $S_k$. Even with $n = 30$ and $k = 10$, the same network with 5 ReLU ResNet layers and the same hyperparameters can easily learn $S_k$ to a vanishing error (Fig. 2b) whereas, as expected, it cannot learn $\chi_{1:k}$ even up to any non-trivial error since $\chi_{1:k}$ is a high-degree monomial (Fig. 2a).

An explanation for this phenomenon is that the neural network learns the staircase function $S_k(x)$ by first learning a degree-1 approximation that picks up the feature $x_1$, and then uses this to more readily learn a degree-2 approximation that picks up the feature $x_1 x_2$, and so on, progressively incrementing the degree of the approximation and 'climbing the staircase' up to the large degrees. We refer to Fig. 1a for an illustration. This is indeed the learning mechanism, as we can see once we plot the Fourier coefficients of the network output against training iteration. Indeed, in Fig. 2c we see that the network trained to learn $\chi_{1:10}$ cannot learn *any* Fourier coefficient relevant to $\chi_{1:10}$ whereas in Fig. 2d it is clear that the network trained to learn $S_{10}$ learns the relevant Fourier coefficients in order of increasing complexity and eventually reaches the $\chi_{1:10}$ coefficient.

**Main results**    We shed light on this phenomenon, proving that certain regular networks efficiently learn the staircase function $S_k(x)$, and, more generally, functions satisfying this structural property:

**Definition 1.1** (Staircase property). *For any $M > 1$, a function $g : \{-1, 1\}^n \to \mathbb{R}$ satisfies the $[1/M, M]$-staircase property over the unbiased binary hypercube if:*

- *for all $S \subset [n]$, if $\hat{g}(S) \neq 0$ then $|\hat{g}(S)| \in [1/M, M]$.*

- *for all $S \subset [n]$, if $\hat{g}(S) \neq 0$ and $|S| \geq 2$, there is $S' \subset S$ such that $|S \setminus S'| = 1$ and $\hat{g}(S') \neq 0$.*

*Furthermore, $g$ is said to be an $s$-sparse polynomial if $|\{S : \hat{g}(S) \neq 0\}| \leq s$.*

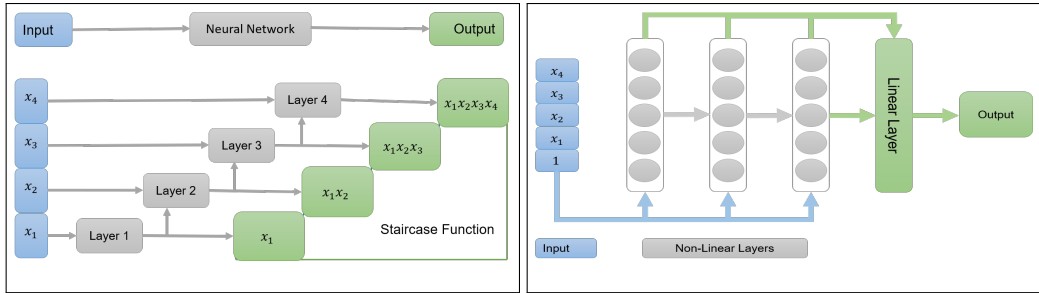

(a) An illustration of hierarchical learning where successive layers build upon the features from previous layers.

(b) An illustration of the proposed architecture. The solid blue and grey arrows represent sparse random connections

Figure 1: Hierarchical learning method and proposed architecture.

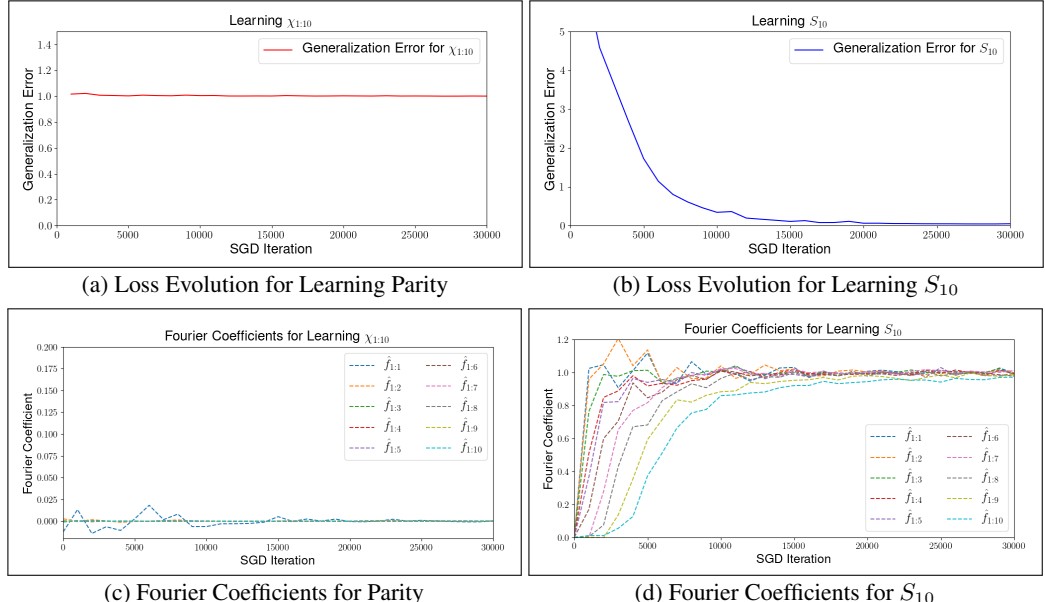

(a) Loss Evolution for Learning Parity

(b) Loss Evolution for Learning $S_{10}$

(c) Fourier Coefficients for Parity

(d) Fourier Coefficients for $S_{10}$

Figure 2: Comparison between training $\chi_{1:10}$ and $S_{10}$ with $n = 30$ on the same 5-layer ReLU ResNet of width 40. Training is SGD with constant step size on the square loss. Here $\hat{f}_{1:i}$ denotes the Fourier coefficient $\langle \chi_{1:i}, f \rangle$ corresponding to the network output $f$.

The parameters $M$ and $s$ appear naturally since a PAC-learning algorithm for $s$-sparse polynomials satisfying the $[1/M, M]$-staircase property must use a number of samples that depends polynomially on $M$ and $s$. Our theoretical result is informally summarized as follows, and we remark that the proof shows that the neural network progressively learns approximations of higher degree:

**Theorem 1.2** (Informal statement of Theorem 2.2). *Let $g : \{-1, 1\}^n \to \mathbb{R}$ be an unknown $s$-sparse polynomial satisfying the $[1/M, M]$-staircase property. Given access to random samples from $\{(x, g(x))\}_{x \sim \{-1, 1\}^n}$, there is a regular neural network architecture that approximately learns $g$ in $\mathrm{poly}(n, s, M)$ time and samples when trained with layerwise stochastic coordinate descent.*

Even though we only consider hierarchical functions over the Boolean hypercube $\{-1, 1\}^n$ in our theoretical result, we believe that the techniques used in this work can be extended to other function spaces of interest, exploiting the orthonormality of the corresponding Fourier basis functions. For this reason we give a fairly general definition of hierarchical functions in Section 3 that goes beyond the Boolean hypercube, as well as beyond the strict notion of increasing chains. This more general class of functions is of further interest because it includes as special cases well-studied classes such as biased sparse parities and decision trees in a smoothed complexity setting (see Section 3.1).

## 1.1 Related Work

**Statistical query emulation results** For the general class of polynomial-size neural network architectures with any choice of initialization, it is known that SGD on a sufficiently small batch-size can learn[2] any function class (including functions satisfying the staircase property) that is efficiently learnable from samples [3], while GD can learn any function class that is efficiently learnable from statistical queries (SQ) [4]. However, these results rely on highly non-regular architectures and initializations, with different parts of the nets responsible for different tasks that emulate the computations of general learning algorithms. In particular, it is not known how to obtain the emulation results of [4] for "regular" architectures and initializations as defined in Definition 2.1. In contrast, our architecture in Theorem 2.2 is a regular neural network in this sense, and our analysis further illustrates how features are built greedily over depth rather than by emulating a given algorithm.

Consider also the orbit class under permutations of the inputs of the "truncated staircase function", $S_{j \to k}(x) = \sum_{i=j}^{k} \chi_{i:k}(x)$, for $1 \leq j \leq k \leq n$. Note that this class is efficiently SQ-learnable when $k = n$, since the monomial $\chi_{1:n}$ is always present and one can recursively check which sub-monomial is present or not by checking at most $n$ monomials at each step. However, we conjecture that $S_{j \to n}(x)$ is not learnable by regular networks trained with gradient descent if $\min(n - j, j) = \omega(1)$. Therefore such truncated staircases provide a candidate for separating gradient-based learning on regular networks versus general, non-regular networks that allow for emulating any SQ algorithm [4].

**Hierarchical models of data** Explicitly adding hierarchical structures into machine learning algorithms such as hierarchical Bayesian modeling and hierarchical linear modeling has proved successful in various machine learning tasks beyond deep learning [8–11]. For image data, [12, 13] propose hierarchical generative models of images and use them to motivate deep convolutional architectures, although these works do not prove that deep learning learns these generative models. [14] similarly proposes a 'deep rendering model' which hierarchically models levels of abstraction present in data, but does not prove learnability. [15] gives a training algorithm for deep convolutional networks that provably learns a deep generative model of images. The paper [16] proposes a generative model of data motivated by evolutionary processes, and proves in a formal sense that "deep" algorithms can learn these models, whereas shallow algorithms cannot. In contrast to our work, the "deep" algorithms considered by [16] are not descent algorithms on regular deep neural network architectures. In [17] it is shown that during training of two-layer ReLU networks with SGD, the lower frequency components of the target function are learned first. Unfortunately, their results have an exponential dependence on the degree. In our work, we leverage depth and the hierarchical Boolean function structure to ensure that higher-level Fourier coefficients are learned efficiently. Finally, [18], studies learning Boolean circuits of depth $O(\log n)$ via neural networks under product distributions using layer-wise gradient descent. While [18] requires the architecture to match the Boolean circuit being learned, in contrast, our architecture is regular and independent of the function learned.

**Power of depth** Several works have studied how representation power depends on depth. [19] shows that deep networks can represent a class of compositionally-created functions more efficiently than shallow networks. [20] shows that certain smooth radial functions can be easily represented by three-layer networks but need exponentially-many neurons to be represented by a two-layer network. Based on an analysis of learning fractal distributions related to the Cantor set, [21] conjectures that if shallow networks are poorly represent a target function, then a deep network cannot be trained efficiently using gradient based methods. [22] presents a depth separation result by showing that deep networks can produce highly oscillatory functions by building on the oscillations layer by layer. [23] uses this phenomenon to show a sharp representation theorem for arbitrary-depth ReLU networks.

Other theoretical works have proved depth separation theorems for training. [24] prove that a two-hidden-layer neural network where the first hidden layer is kept random and only the second layer is trained, provably outperforms a just one hidden layer network. In [25, 26] it is proved that deep networks trained end-to-end with SGD and quadratic activation functions can efficiently learn a non-trivial concept class hierarchically, whereas kernel methods and lower-depth networks provably fail to do so. The class of functions studied by [25, 26] are those representable as the sum of neurons in a teacher network that is well-conditioned, has quadratic activations and has a depth of at most $\log \log n$, where $n$ is the number of inputs. This function class is expressive but incomparable to the hierarchical function class studied in our work (e.g., we can learn polynomials up to degree $n$,

---

[2]These reductions are for polynomial-time algorithms and for polynomial precisions on the gradients.

whereas [26] is limited to degree $\log(n)$). Furthermore, our function class has the advantage of being naturally interpretable, with complex features (the high-order monomials) being built in a transparent way from simple features (the low-order monomials). In [27]–[29], gradient dynamics are explored for the simplified case of deep linear networks, where an 'incremental learning' phenomenon is observed in which the singular values are learned sequentially, one after the other. This phenomenon is reminiscent of the incremental learning of the Fourier coefficients of the Boolean function in our setting (Fig. 2). For real world data sets, [30] empirically shows that in many data sets of interest, simple neural networks trained with SGD first fit a linear classifier to the data and then progressively improve the approximation, similar in spirit to the theoretical results in this paper.

**Neural Tangent Kernel and random features** A sequence of papers have studied convergence of overparametrized (or Neural Tangent Kernel regime) neural networks to the global minimizer of empirical loss when trained via gradient descent. In this regime, they show that neural networks behave like kernel methods and give training and/or generalization guarantees. Because of the reduction to kernels, these results are essentially non-hierarchical. [31, 32] in fact show that deep networks in the NTK regime behave no better than shallow networks. We refer to [26] for a review of the literature related to NTK and shallow learning. Finally, we mention the related works [33]–[35] which consider learning low-degree (degree $q$) polynomials over $\mathbb{R}^n$, without a hierarchical structure assumption. They require $n^{\Omega(q)}$ neurons to learn such functions, which is super-polynomial once $q \gg 1$. The results hold in the random features regime, known to be weaker than NTK.

## 1.2 Organization

In Section 2, we give the problem setup, network architecture and the training algorithm, and also state our rigorous guarantee that this training algorithm learns functions satisfying the staircase property. In Section 3 we discuss possible extensions, defining hierarchical functions satisfying the staircase property in a greater level of generality from Definition 1.1. We refer to Appendix A for additional experiments which validate our theory and conjectures for both the simplest definition of the staircase property in Definition 1.1 and the generalizations in Section 3.2.

## 2 Regular networks provably learn hierarchical Boolean functions

We state our main theoretical result, which proves that a regular neural network trained with a descent algorithm learns hierarchical Boolean functions in polynomial time.

### 2.1 Architecture

Our network architecture has neuron set $V$ and edge set $E$, and is defined as follows (see also Fig. 1b). The neuron set is $V = V_{\text{in}} \sqcup V_1 \sqcup \ldots V_L$. Here $V_{\text{in}} = \{v_{\text{in},0}, v_{\text{in},1}, \ldots, v_{\text{in},n}\}$ is a set of $n + 1$ inputs, and each intermediate layer consists of $|V_i| = W$ neurons. Furthermore, the edge set $E$ is a sparse, random subset of all possible directed edges:

- each $(v_0, v_i) \in V_{\text{in}} \times V_i$ is in the edge set $E$ independently with probability $p_1$, and
- each $(v_i, v_{i+1}) \in V_i \times V_{i+1}$ for $i \in [L-1]$ is in the edge set $E$ independently with probability $p_2$.

For each edge $e \in E$, let there be a weight parameter $a_e \in \mathbb{R}$. And for each neuron $v \in V \setminus V_{\text{in}}$, let there be a bias parameter $b_v \in \mathbb{R}$. The parameters of the network are therefore $a \in \mathbb{R}^E$ and $b \in \mathbb{R}^{V \setminus V_{\text{in}}}$. For simplicity of notation, we concatenate these two vectors into one vector of parameters

$$w = [a \quad b] \in \mathbb{R}^E \oplus \mathbb{R}^{V \setminus V_{\text{in}}}.$$

For each $i \in [n]$, the $i$th input, $v_{\text{in},i} \in V_{\text{in}}$, computes $x_i$, and the 0th input, $v_{\text{in},0}$, computes a constant:

$$f_{v_{\text{in},i}}(x; w) = x_i, \text{ and } f_{v_{\text{in},0}}(x; w) = 1.$$

Given a neuron $v \in V \setminus V_{\text{in}}$, the function computed at that neuron is a quadratic function of a linear combination of neurons with edges to $v$, (i.e., the activation function is quadratic). And the output of the neural network is the sum of the values of the neurons at the intermediate layers:

$$f_v(x; w) = \left( \sum_{e=(u,v) \in E} a_e f_u(x; w) \right)^2 + b_v, \text{ and } f(x; w) = \sum_{v \in V \setminus V_{\text{in}}} f_v(x; w).$$

Our architecture satisfies the following regularity condition:

**Definition 2.1** (Regular network architecture and initialization)**.** *An architecture is regular if for any $1 \leq i \leq j \leq L$, for any distinct pair of potential edges $(v_i, v_j), (v'_i, v'_j) \in V_i \times V_j$, the events that these edges are in $E$ are i.i.d.; the same holds for any distinct pair of potential edges $(u, v_j), (u', v'_j) \in V_{\text{in}} \times V_j$; the same holds for any distinct pair of potential edges $(u, v_{\text{out}}), (u', v_{\text{out}}) \in V_j \times v_{\text{out}}$ (where $v_{\text{out}}$ is the output vertex). Furthermore, the initialization is regular if it is i.i.d. over the set of present edges and each weight has a symmetric distribution.*

In our case the weight initialization is i.i.d. and symmetric since we choose it to be identically zero everywhere, which works since we escape saddle points by perturbing during training. On the other hand in our experiments the initialization is an isotropic Gaussian, which also satisfies Definition 2.1.

## 2.2 Loss function

Let the loss function be the mean-squared-error between the output $f$ of the network and a function $g : \{-1, 1\}^n \to \mathbb{R}$ that we wish to learn. Namely, for any $x \in \{-1, 1\}^n$, $a \in \mathbb{R}^E$ and $b \in \mathbb{R}^{V \setminus V_{\text{in}}}$, define the point-wise loss and population loss functions respectively, where $w = [a, \ b]$:

$$\ell(x; w) = \frac{1}{2}(f(x; w) - g(x))^2; \quad \ell(w) = \mathbb{E}_{x \sim \{-1,1\}^n} \ell(x; w) \,. \tag{3}$$

We will train the neural network parameters to minimize an $L_2$-regularized version of the loss function. Let $\lambda_1, \lambda_2 > 0$ be regularization parameters, and define the point-wise regularized loss $\ell_R(x; w) = \ell(x; w) + R(w)$ and the population regularized loss $\ell_R(w) = \ell(w) + R(w)$, where

$$R(w) = \frac{1}{2} \sum_{\substack{e=(u,v)\in E \\ u \in V_{\text{in}}}} \lambda_1 a_e^2 + \frac{1}{2} \sum_{\substack{e=(u,v)\in E \\ u \notin V_{\text{in}}}} \lambda_2 a_e^2.$$

The distinct regularization parameters $\lambda_1, \lambda_2 > 0$ for the weights of edges from the input and previous-layer neurons, respectively, are for purely technical reasons and are explained in Section 2.5.

## 2.3 Training

We train the neural network to learn a function $g : \{-1, 1\}^n \to \mathbb{R}$ by running Algorithm 1. This algorithm trains layer-wise from layer 1 to layer $L$. The $i$th layer is trained with stochastic block coordinate descent, iterating through the neurons in $V_i$ in an arbitrary fixed order, and training the parameters of each neuron $v \in V_i$ using the TRAINNEURON subroutine. Each call of TRAINNEURON runs stochastic gradient descent to train the subset of neural network parameters $w_v = \{a_e\}_{e=(u,v)\in E} \cup \{b_v\}$ directly associated with neuron $v$ (i.e., the weights of the edges that go into $v$, and the bias of $v$), keeping the other parameters $w_{-v} = \{a_e\}_{e=(u',v')\in E \text{ s.t. } v \neq v'} \cup \{b_{v'}\}_{v' \in V \setminus (\{v\} \cup V_{\text{in}})}$ fixed.

---

**Algorithm 1:** TRAINNETWORKLAYERWISE

---

**Input:** Sample access to the distribution $\{(x, g(x))\}_{x \sim \{-1,1\}^n}$. Hyperparameters
$\quad\quad W, L, p_1, p_2, \lambda_1, \lambda_2, \eta, B, \epsilon_{stop}, \alpha, \tau$.
**Output:** Trained parameters of neural network after training layer-wise.

---

1  $(V, E) \leftarrow$ random network constructed as in Section 2.1.
2  $w^0 \leftarrow \vec{0}, t \leftarrow 0$          // Initialize all weights and biases to zero.
3  **for** *layer $i = 1$ to $L$* **do**
4      **for** *neuron $v \in V_i$* **do**
        // Train the neuron parameters, $w_v$, fixing other parameters
5          $w^{t+1} \leftarrow$ TRAINNEURON$(v, w^t; \lambda_1, \lambda_2, \eta, B, \epsilon_{stop}, \alpha, \tau)$
6          $t \leftarrow t + 1$
7      **end**
8  **end**
9  Return $w^t$

---

---

**Algorithm 2:** TRAINNEURON$(v, w^0; \lambda_1, \lambda_2, \eta, B, \epsilon_{stop}, \alpha, \tau)$

---

**Input:** Neuron $v \in V \setminus V_{\text{in}}$. Initial network parameters $w^0$. Access to random samples $(x, g(x))$
for $x \sim \{-1, 1\}^n$. Hyperparameters $\lambda_1, \lambda_2, \eta, B, \epsilon_{stop}, \alpha, \tau$.

**Output:** Parameters of network after the subset of parameters $w_v$ of neuron $v$ is perturbed and
then trained with NEURONSGD, while all other parameters $w_{-v}$ remain fixed.

// To avoid saddle points, randomly perturb the neuron parameters

1   $w_v^{perturb} \leftarrow w_v^0 + z$, where $z$ is a noise vector whose entries are i.i.d. in Unif$([-\eta, \eta])$.

2   $w^{perturb} \leftarrow [w_{-v}^0, w_v^{perturb}]$

// Run stochastic gradient descent on neuron parameters, until
    approximate stationarity

3   $w^{SGD} \leftarrow$ NEURONSGD$(v, w^{perturb}; \lambda_1, \lambda_2, B, \epsilon_{stop}, \alpha)$

// Prune the neuron's small weights

4   $w_v^{\text{round}} \leftarrow w_v^{SGD}$, rounding to 0 every entry of magnitude less than $\tau$

5   Return $w^{\text{round}} = [w_{-v}^0, w_v^{\text{round}}]$

---

---

**Algorithm 3:** NEURONSGD$(v, w^0; \lambda_1, \lambda_2, \eta, B, \epsilon_{stop}, \alpha, \tau)$

---

**Input:** Neuron $v \in V \setminus V_{\text{in}}$. Initial network parameters $w^0$. Access to random samples $(x, g(x))$
for $x \sim \{-1, 1\}^n$. Hyperparameters $\lambda_1, \lambda_2, \eta, B, \epsilon_{stop}, \alpha, \tau$.

**Output:** Parameters of network after the subset of parameters $w_v$ corresponding to neuron $v$ is
trained, all other parameters $w_{-v}$ remain fixed.

1   $t \leftarrow 0$

2   **while** *true* **do**

    // Approximate $\nabla_{w_v} \ell_R$ with minibatch size $B$

3      Draw i.i.d. data samples $(x^{t,1}, g(x^{t,1})), \ldots, (x^{t,B}, g(x^{t,B}))$

4      $\xi^t \leftarrow \frac{1}{B} \sum_{i=1}^{B} \nabla_{w_v} \ell_R(x^{t,i}; w_{-v}^0, w_v^t)$

    // Stop if we have reached an approximate stationary point

5      **if** $\|\xi^t\| \leq \epsilon_{stop}$ **then** break out of the loop;

    // Update $w_v$ in direction of the approximate gradient

6      $w_v^{t+1} \leftarrow w_v^t - \alpha \xi^t$

7      $t \leftarrow t + 1$

8   Return $[w_{-v}^0, w_v^t]$

---

### 2.4 Theoretical result

We prove that Algorithm 1 learns functions satisfying the staircase property in the sense of Definition 1.1. We defer the exact bounds on the parameters considered to Appendix B.

**Theorem 2.2.** *Let $g : \{-1, 1\}^n \to \mathbb{R}$ be an unknown $s$-sparse polynomial satisfying the $[1/M, M]$-staircase property for some given $s, M > 1$. Given an accuracy parameter $\epsilon > 0$, a soundness parameter $0 < \delta < 1$, and access to random samples from $\{(x, g(x))\}_{x \sim \{-1,1\}^n}$, there is a setting of hyperparameters for Algorithm 1 that is polynomially-bounded, i.e.,*

$$1/\operatorname{poly}(n, s, M, 1/\epsilon, 1/\delta) \leq W, L, p_1, p_2, \lambda_1, \lambda_2, \eta, B, \epsilon_{stop}, \alpha, \tau \leq \operatorname{poly}(n, s, M, 1/\epsilon, 1/\delta),$$

*such that Algorithm 1 runs in $\operatorname{poly}(n, s, M, 1/\epsilon, 1/\delta)$ time and samples and with probability $\geq 1 - \delta$ returns trained weights $w$ satisfying the bound $\ell(w) \leq \epsilon$ on the population loss.*

### 2.5 Proof overview

We now briefly describe how Algorithm 1 learns, giving a high-level depiction of the training process in the case that the target function is the staircase function $S_3(x) = x_1 + x_1 x_2 + x_1 x_2 x_3$. We refer to Fig. 3 for an illustration of the training procedure, where grey neurons are 'blank' (i.e., have identically zero output) and the green neurons are 'active' (i.e., compute a non-zero function). Initially all neurons are blank and the total output of the network is 0.

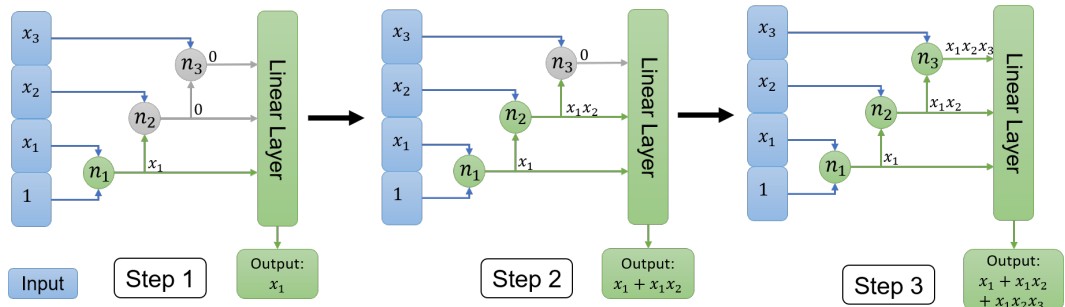

Figure 3: An illustration of the training procedure for learning $S_3(x) = x_1 + x_1x_2 + x_1x_2x_3$. The grey neurons are 'blank' and the green neurons are 'active'.

We set the random network topology connectivity hyperparameters $p_1$ and $p_2$ to be small, so that the network is sparse. We can show that the following invariant is maintained throughout training: any neuron has at most two active parents. Intuitively, this is because we can bound the number of active neurons at any iteration during training by $s + n + 1$, so the number of neuron tuples $(u_1, u_2, u_3, v) \in V^4$ such that $u_1, u_2, u_3$ are active and all have edges to $v$ is in expectation bounded by $(s + n + 1)^3 (p_1)^3 W \ll 1$. Since any neuron during training has at most two active parents, we may tractably analyze TRAINNEURON for training new neurons: in a key technical lemma, we show that every active neuron $v$ has exactly two active parents $u$ and $u'$, and approximately computes a monomial given by the product of the parents' values, $f_v(x) \approx \chi_{S_v} \approx f_u(x) f_{u'}(x)$.

We cannot set $p_1$ and $p_2$ to be too small or else the network will not be connected enough to learn. Thus, we must also set the connectivity parameters so that for any pair $(u, u') \in V_{\text{in}} \times (V \setminus V_L)$, the neurons $u$ and $u'$ share many children, and at least one of these children may learn the product if it is useful. For this it is sufficient to take the expected number of shared children $p_1 p_2 W \gg 1$ very large. We now present a run of the algorithm, breaking it up into "steps" for exposition.

**Step 1:** The algorithm iterates over neurons and trains them one by one using TRAINNEURON. Most of the neurons trained are left blank: for example, if a neuron $v$ has the two inputs $x_2$ and 1, then by our key technical lemma the neuron could either remain blank or learn the product of the inputs, $x_2 = x_2 \cdot 1$. But the mean-squared error cannot decrease by learning $x_2$, since $x_2$ is orthogonal to the staircase function in the $L_2$ sense (i.e., $\langle S_3(x), x_2 \rangle = 0$, because the staircase function does not have $x_2$ as a monomial), so the neuron $v$ remains blank. Let $t_1$ be the first iteration at which the algorithm reaches a neuron $n_1 \in V_1$ that has $x_1$ and 1 as inputs. When the network trains $n_1$ using the sub-routine, we show that it learns to output $x_1 = x_1 \cdot 1$, since that is the highest-correlated function to $S_3(x)$ that $n_1$ can output. Combined with the linear layer, the overall neural network output becomes $f(x; w^{t_1}) \approx x_1$.

**Step 2:** The error function after Step 1 is $E_1(x) = S_3(x) - f(x; w^{t_1}) \approx x_1x_2 + x_1x_2x_3$. Again, for many iterations the training procedure keeps neurons blank, until at iteration $t_2$ it reaches a neuron $n_2$ with inputs $x_1$ (due to neuron $n_1$) and $x_2$ (directly from the input). Similarly to Step 1, when we train $n_2$, we show that it learns to output $x_1x_2$, which is the function with highest correlation to $E_1(x)$ which $n_2$ can output. Thus, the neural network now learns to output $f(x; w^{t_2}) \approx x_1 + x_1x_2$, so the error function has decreased to $E_2(x) = S_3(x) - f(x; w^{t_2}) \approx x_1x_2x_3$. The training proceeds in this manner until all the monomials in $S_3(x)$ are learned by the network.

**Error Propagation and Regularization:** A significant obstacle in analyzing layer-wise training is that outputs of neurons are inherently noisy because of incomplete training, and the error may grow exponentially along the depth of the network. In order to avoid this issue, we have two distinct regularization parameters $\lambda_1, \lambda_2$ and connectivity parameters $p_1, p_2$ for edges from inputs versus edges from neurons. In our proof of Theorem 2.2, we set $\lambda_1 \ll \lambda_2$ and $p_1 \gg p_2$, which ensures that after training a neuron (say $n_2$ above) the weight from the neuron $n_1$ (which has regularization parameter $\lambda_2$) is much smaller than the weight directly from the input $x_2$ (which has regularization parameter $\lambda_1$). Since the inputs are noise-free, this disallows exponential growth of errors along the depth. We conjecture that if the network is trained end-to-end instead of layer-wise, then one can avoid this technical difficulty and set $\lambda_1 = \lambda_2$ and $p_1 = p_2$, because of a backward feature correction phenomenon [26] where the lower layers' accuracy improves as higher levels are trained.

# 3 General Hierarchical Structure

## 3.1 Extension to biased binary inputs and implications

We extend the main result of this paper to more general setting of functions over a space of i.i.d. binary variables that have zero expectation but are not necessarily supported on $\{+1, -1\}$. For instance, if $\{X_i\}_{i \in [n]}$ are i.i.d. and Boolean (on $\{+1, -1\}$) with $\mathbb{E}(X_i) = b$ for some $b \in [-1, 1]$, the centered variables $\tilde{X}_i = X_i - b$ are valued in $\{1 - b, -1 - b\}$. Over these centered variables, the Fourier coefficients of a function are given (up to normalization) by $\hat{f}(S) = \mathbb{E}_{\tilde{X}} f(\tilde{X}) \prod_{i \in S} \tilde{X}_i$ for any $S \subset [n]$. Thus, the staircase property of Definition 1.1 generalizes clearly: the function $f : \{1 - b, -1 - b\}^n \to \mathbb{R}$ satisfies the staircase property if for any $S \subset [n]$ with $|S| \geq 2$ such that $\hat{f}(S) \neq 0$, there is a subset $S' \subset S$ such that $|S \setminus S'| = 1$ and $\hat{f}(S') \neq 0$.

Showing a similar result to Theorem 2.2 for staircase functions on the variables $\tilde{X}_i$ with a quadratic activation requires a slight modification of our argument since $\tilde{X}_i^2$ is no longer constant (and equal to 1), so one cannot use the simple identity $Z_1 Z_2 = (Z_1 + Z_2)^2/2 - 1$ that holds for variables valued in $\{+1, -1\}$ to prove that a neuron learns the product of its inputs when trained. However, adding skip connections from the previous layer with quadratic activation, along with the fact that $r_1 r_2 = ((r_1 + r_2)^2 - r_1^2 - r_2^2)/2$, one can hierarchically learn new features as products of previously-learned features. Alternatively, one can change the activation so that each neuron maps a vector input $v$ to $(a \cdot v + b)^2 + c \cdot y^{\cdot 2}$, where $a, b, c$ are trainable parameters, to learn products of general binary variables. A similar proof to that of Theorem 2.2 is then expected to hold, implying that one can learn staircase functions over i.i.d. random variables that are binary and centered (beyond $\{+1, -1\}$ specifically). We will now discuss two interesting examples that fall under this setting.

**Biased sparse parities and kernel separation**    Consider the problem of learning sparse biased parities, i.e., the class of monomials of degree $\log(n)$ with a $\{+1, -1\}$-valued input distribution that is i.i.d. with $\mathbb{E}X_1 = b = 1/2$. It is shown in [36] that such a distribution class is not learnable by any kernel method with poly-many features, while it is learnable by gradient-based learning on neural networks of polynomial size. The result of [36] relies on an architecture that allows emulating an SQ algorithm – far from a regular network as considered in this paper. However, sparse biased parities are staircase functions over unbiased binary variables, with polynomially-many nonzero coefficients since the degree is logarithmic. So an extension of Theorem 2.2 to arbitrary binary centered variables would imply that regular networks can learn sparse biased parities, implying a separation between kernel-based learning and gradient-based learning on regular networks.

**Decision trees under smoothed complexity model**    Secondly, in a smoothed complexity setting, where the input distribution is drawn from the biased binary hypercube, and the bias of each variable is randomly chosen, the class of $\log(n)$-depth decision trees satisfies the general staircase property. This is because Lemma 3 in [37] implies that with high probability there is a $\mathrm{poly}(n/\epsilon)$-sparse polynomial that $\epsilon$-approximates the decision tree and satisfies the staircase property over the biased binary hypercube. Thus, the extension of our result to the case of biased binary inputs would imply that regular neural networks learn decision trees in the smoothed complexity model.[3]

## 3.2 Extension to more general $L_2$ spaces

We now give an even more general version of the staircase property in Definition 1.1. Since neural networks are efficient at representing affine transforms of the data and smooth low-dimensional functions [35], we generalize the class of hierarchical functions over the space of continuous, real valued functions $[-R, R]^n \subseteq \mathbb{R}^n$; $R \in \mathbb{R}^+ \cup \{\infty\}$ without any reference to underlying measures but with enough flexibility to add additional structures like measures and the corresponding $L^2$ norms. Set $R \in \mathbb{R}^+ \cup \{\infty\}$ and consider any sequence of functions $\mathcal{H} := \{h_k : \mathbb{R} \to \mathbb{R}\}_{k \in \mathbb{N} \cup \{0\}}$ such that $h_0$ is the constant function 1, and any affine transform $\mathcal{A} : \mathbb{R}^n \to \mathbb{R}^n$ such that $\mathcal{A}(x) = Ax + b$ for $A \in \mathbb{R}^{n \times n}; b \in \mathbb{R}^n$. We call a function $f : [-R, R]^d \to \mathbb{R}$ to be $(\mathcal{H}, \mathcal{A})$-polynomial if there exists a finite index set $I_f \subset (\mathbb{N} \cup \{0\})^n$ such that for some real numbers $(\alpha_{\mathbf{k}})_{\mathbf{k} \in I_f}$:

$$f(x) = \sum_{\mathbf{k} := (k_1, \ldots, k_n) \in I_f} \alpha_{\mathbf{k}} \prod_{i=1}^{n} h_{k_i}(y_i)$$

---

[3]Such a conjecture was recently made by [38], which leaves as an open problem in Section 1.3 whether neural networks can learn $\log(n)$-juntas in a smoothed complexity setting, and implicitly poses the same problem about the more general case of $\log(n)$-depth decision trees.

Where $y := \mathcal{A}(x)$. We also define $\mathsf{Ord}(\mathbf{k}) := |\{i : k_i \neq 0\}|$ and a partial order '$\preceq$' over $(\mathbb{N} \cup \{0\})^n$ such that $\mathbf{k}' \preceq \mathbf{k}$ iff $k_i' \in \{0, k_i\}$ for every $i \in [n]$. For $M \geq 1$, we will call a $(\mathcal{H}, \mathcal{A})$-polynomial to be $(1/M, M)$ hierarchical if

1. $1/M \leq |\alpha_{\mathbf{k}}| \leq M$ for every $\mathbf{k} \in I_f$.
2. For every $\mathbf{k} \in I_f$ such that $\mathsf{Ord}(\mathbf{k}) \geq 2$, there exists $\mathbf{k}' \in I_f$ such that $\mathsf{Ord}(\mathbf{k}') = \mathsf{Ord}(\mathbf{k}) - 1$ and $\mathbf{k}' \preceq \mathbf{k}$.

We now extend the definition to general continuous functions. Suppose $d_{\mathcal{S}}$ is a pseudo-metric on the space of bounded continuous functions $\mathcal{C}^b([-R, R]^n; \mathbb{R})$. We call $f \in \mathcal{C}^b([-R, R]^n; \mathbb{R})$ to be $(1/M, M, \mathcal{S})$ hierarchical if for every $\epsilon > 0$, there exists a $(1/M, M)$ hierarchical $(\mathcal{H}, \mathcal{A})$-polynomial $f_\epsilon$ such that: $d_{\mathcal{S}}(f, f_\epsilon) < \epsilon$. We note some examples below:

1. Let $\mu$ be the uniform measure over $\{-1, 1\}^n$, $d_{\mathcal{S}}$ be the $L^2$ norm induced by $\mu$, $\mathcal{H} = \{1, x\}$ and $\mathcal{A}$ be identity mapping. We note that functions over the unbiased Boolean hypercube satisfying the $[1/M, M]$-staircase property in Definition 1.1 correspond to $(1/M, M, L^2(\mu))$ hierarchical functions as defined above.

2. In the case when $\mu$ is the biased product measure over $\{-1, 1\}^n$, we can take $\mathcal{H} = \{1, x\}$ and $\mathcal{A}(x) = x - \mathbb{E}x$. This recovers the definition in Section 3.1.

3. When $\mu$ is the isotropic Gaussian measure, we can take $R = \infty$, $\mathcal{H}$ to be the set of 1-D Hermite polynomials and $\mathcal{A}$ to be the identity. In case $\mu = \mathcal{N}(m, \Sigma)$, we instead take $\mathcal{A} = \Sigma^{-1/2}(x - m)$.

4. When $R < \infty$ and $\mathcal{S}$ is the $L^2$ norm with respect to the Lebesgue measure. $\mathcal{C}([-R, R]^n; \mathbb{R})$, we can take $\mathcal{H} = \{\exp(i\frac{\pi k x}{R}) : k \in \mathbb{Z}\}$ and $\mathcal{A} = I$. This allows us to interpret $(\mathcal{H}, \mathcal{A})$-polynomial approximations as Fourier series approximations.

5. When $R < \infty$ and $\mathcal{S}$ is the uniform norm (or sup norm) over $\mathcal{C}([-R, R]^n; \mathbb{R})$, we can take $\mathcal{H} = \{1, x, x^2, \dots, \}$ and $\mathcal{A} = I$. Since any continuous function can be approximated by a polynomial, this presents a large class of functions of interest.

In items 1-4, we consider these specific function classes $\mathcal{H}$ in order to make $(\mathcal{H}, \mathcal{A})$ monomials orthonormal under $L^2(\mu)$. We leave it as a direction of future work to extend our theoretical learning results in Theorem 2.2 to such function classes.

### 3.3 Composable chains

Finally, we discuss a distinct way to generalize the staircase property. One can relax the strict inclusion property of Definition 1.1 with a single element removed, to more general notions of increasing chains. For instance, if $\hat{g}(S) \neq 0$, one may require that there exists an $S'$ such that $|S'| < |S|$ and $|S \Delta S'| = O(1)$, and we conjecture that regular networks will still learn sparse polynomials with this structure. More generally, one may require that for any $S$ such that $\hat{g}(S) \neq 0$, there exists a constant number of $S_j$'s such that $|S_j| < |S|$ and $\hat{g}(S_j) \neq 0$ for all $j$, and such that $S$ can be composed by $\{S_j\}$ and the input features $x_1, \dots, x_n$, where in the Boolean setting the composition rule corresponds to products. Finally, one could further generalize the results by changing the feature space, i.e., using regular networks that take not just the standard inputs $x_1, \dots, x_n$, but also have other choices of features $\phi_1(x^n), \dots, \phi_p(x^n)$ as inputs, for $p$ polynomial in $n$.

## 4 Limitations and societal impacts

For simplicity of the proofs, the architecture and training algorithm are not common in practice: quadratic activations, a sparse connectivity graph, and layer-wise training [39–41] with stochastic block coordinate descent. We also perturb the weights with noise in order to avoid saddle points [42], and we prune the low-magnitude weights to simplify the analysis (although this may not deteriorate performance much in practice [43]). We emphasize that these limitations are purely technical as they make the analysis tractable, and we conjecture from our experiments that ReLU ResNets trained with SGD efficiently learn functions satisfying the staircase property. This work does not deal directly with real world data, so may not have direct societal impacts. However, it aims to rigorously understand and interpret deep learning, which may aid us in preventing unfair behavior by AI.

**Acknowledgments**

We are grateful to Philippe Rigollet for insightful conversations. EA was supported in part by the NSF-Simons Sponsored Collaboration on the Theoretical Foundations of Deep Learning, NSF award 2031883 and Simons Foundation award 814639. EB was supported in part by an NSF Graduate Fellowship and an Apple Fellowship and NSF grant DMS-2022448. GB was supported in part by NSF CAREER award CCF-1940205. DN was supported in part by NSF grant DMS-2022448.

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
