# A  Experiments

The expository experiments given in Figure 2 compared the training of $S_k$ to the training of $\chi_{1:k}$ with each iteration of SGD drawing fresh i.i.d samples from the data. In this section, we fix the number of samples $m$ and cycle through it with some mini-batch size $B$ at each iteration. In order to maintain the comparison fair, we normalize $S_k$ in order for it to have the same $L_2$ norm as $\chi_{1:k}$ (whenever there is a comparison). For example, in the case of uniform measure over the hypercube, we replace $S_k$ with $S_k/\sqrt{k}$. We also conduct the experiments for various underlying distributions (such as Gaussians and biased product distributions on the Hypercube), the double staircase function and various choices of $n$ and $k$. When the underlying distribution, $n$ and $k$ are fixed, we will attempt to learn $S_k$ and $\chi_{1:k}$ with the same neural network along with the same parameters and hyper-parameters. We use the ReLU resnet architecture everywhere, with the same width across the layers(see [44]). We train the network by minimizing the square loss via. SGD. The errors and Fourier coefficients plotted below are all computed with fresh samples (of size $3 \times 10^4$).

In all the experiments below, we note that the functions satisfying the staircase property are learnt hierarchically - i.e, the network learns the simpler features first and then builds up to the complex features. However, the network is unable to learn just the complex features by themselves (like $\chi_{1:k}$) to any non-trivial accuracy.

**Learning with Unbiased Parities:**  We consider the same parameters as in Figure 2, but with a fixed number of samples and $S_k/\sqrt{k}$ instead of $S_k$ in order to normalize. We take $n = 30$, $k = 10$, number of samples $m = 6 \times 10^4$, mini-batch size $B = 20$, depth 5, width 40. The results are plotted in Figure 4. The Fourier coefficient $\hat{f}_S$ for $S \subseteq [n]$ denotes $\mathbb{E}f(x)\chi_S(x)$ for $f$ being either $\chi_{1:k}$ or $S_k/\sqrt{k}$.

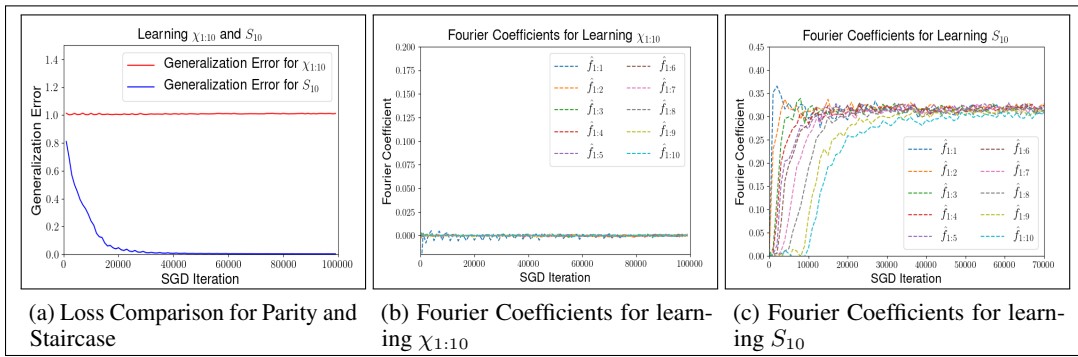

| (a) Loss Comparison for Parity and Staircase | (b) Fourier Coefficients for learning $\chi_{1:10}$ | (c) Fourier Coefficients for learning $S_{10}$ |

Figure 4: Learning Staircase and Parity functions with UnBiased Rademacher data.

**Learning with Gaussian Data:**  We draw $x$ from the standard Gaussian distribution of $\mathbb{R}^n$ instead of the uniform measure over $\{-1, 1\}^n$. This scenario is harder since monomials $\prod_{i=1}^{k} x_i$ can have heavy tails, will makes them occasionally take very large values. Hence, we take $k$ to be small and $n$ to be large. Instead of $S_k$, we consider $S_k/\sqrt{k}$ to ensure that its $L_2$ norm under the Gaussian measure is 1. In figure 5 we take $n = 100$, $k = 5$ number of samples $m = 3 \times 10^5$, mini-batch size $B = 20$, depth 8, width 50. The Fourier coefficient $\hat{f}_S$ for $S \subseteq [n]$ denotes $\mathbb{E}f(x)\chi_S(x)$ for $f$ being either $\chi_{1:k}$ or $S_k/\sqrt{k}$.

**Learning with Biased Parities:**  In Figure 6, we consider the co-ordinates of $x$ to be drawn i.i.d from $\{-1, 1\}$, but biased such that $\mathbb{P}(x_1 = 1) = p = 0.75$. In the definitions of $S_k$ and $\chi_{1:k}$, we replace $x_i$ with $\bar{x}_i := \frac{x_i - 2p + 1}{\sqrt{4p(1-p)}}$ and attempt to learn $S_k(\bar{x})/\sqrt{k}$ and $\chi_{1:k}(\bar{x})$. We take $n = 30$, $k = 7$, number of samples $m = 6 \times 10^4$, mini-batch size $B = 20$, depth 5, width 40. The Fourier coefficient $\hat{f}_S$ for $S \subseteq [n]$ denotes $\mathbb{E}f(\bar{x})\chi_S(\bar{x})$ for $f$ being either $\chi_{1:k}$ or $S_k/\sqrt{k}$.

**Learning the Double Staircase:**  We now consider learning the double staircase function, which has the structure defined in Definition 1.1. Define $S_{k,l} = S_k(x) + x_1 x_{k+1} + x_1 x_{k+1} x_{k+2} + \cdots +$

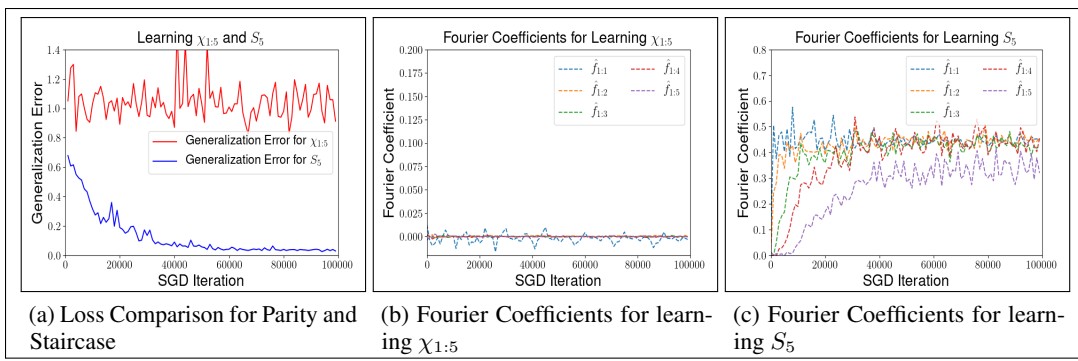

Figure 5: Learning Staircase and Parity functions with Gaussian data.

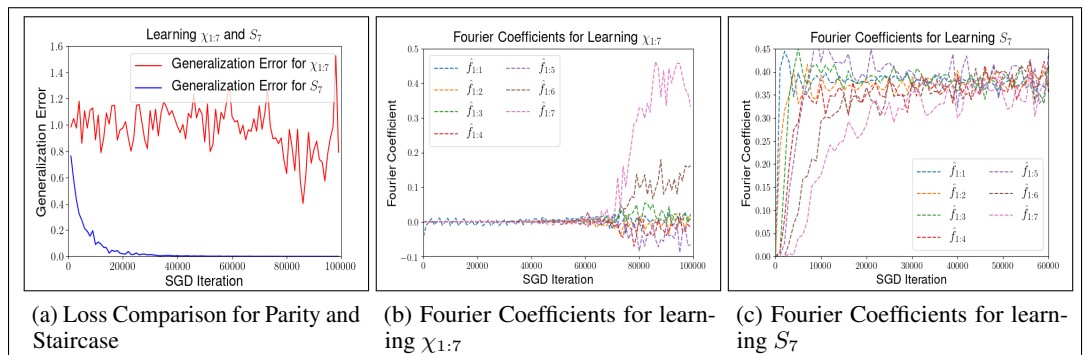

Figure 6: Learning Staircase and Parity functions with biased Rademacher data.

$x_1 \prod_{i=1}^{l-1} x_{k+i}$. We take $k = l = 7$ and $n = 30$, width 50, depth 5, mini-batch size $B = 20$ and number of samples $m = 10^5$. For simplicity, we choose the underlying distribution to be the uniform distribution over $\{-1, 1\}^n$. The Fourier coefficients here are same as that for the staircase function under the uniform measure over $\{-1, 1\}^n$.

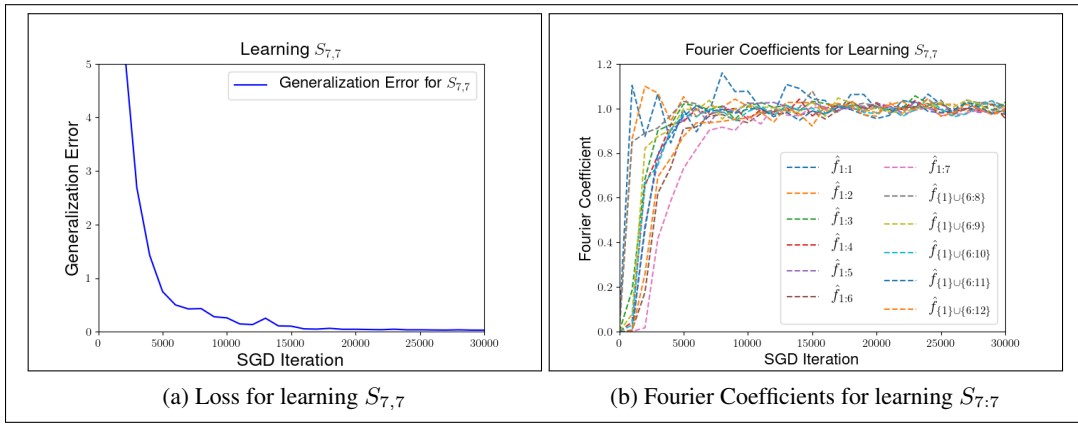

Figure 7: Learning the double-Staircase function with Rademacher data.

## B Formal Theorem Statement

We restate the main theorem, giving an explicit set of hyperparameters that works.

**Theorem B.1.** *There is a universal constant $C > 0$ such that following holds. Let $g : \{-1, 1\}^n \to \mathbb{R}$ be an unknown s-sparse polynomial satisfying $[1/M, M]$-staircase property over the unbiased*

*Boolean hypercube (Definition 1.1) for known $s, M > 1$. Given an accuracy parameter $\epsilon > 0$, a soundness parameter $0 < \delta < 1$, and access to random samples from $\{(x, g(x))\}_{x \sim \{-1,1\}^n}$, with the following setting of hyperparameters for Algorithm 1,*

$$L = n \tag{4}$$

$$W = (64M^2(n+s+1)^3L/\delta)^{24} \tag{5}$$

$$p_1 = (64M^2(n+s+1)^3L/\delta)^{-9} \tag{6}$$

$$p_2 = (64M^2(n+s+1)^3L/\delta)^{-13} \tag{7}$$

$$\tau = 1/(2^{20}M^7L) \tag{8}$$

$$\eta = 4\tau = 1/(2^{18}M^7L) \tag{9}$$

*Define*

$$\kappa = WLMs/(\epsilon\delta).$$

*For a sufficiently small constant $c_\lambda > 0$,*

$$\lambda_2 = c_\lambda \kappa^{-28} \leq 1 \tag{10}$$

$$\sqrt{\lambda_1/\lambda_2} = 1/(64M^2L) \leq 1 \tag{11}$$

*For a sufficiently small constant $c_{stop} > 0$ that may depend on $c_\lambda$,*

$$\epsilon_{stop} = c_{stop}\kappa^{-430} \tag{12}$$

*For a sufficiently small constant $c_\alpha > 0$ that may depend on $c_\lambda, c_{stop}$,*

$$\alpha = c_\alpha(\lambda_1\lambda_2)^5\kappa^{-72} \tag{13}$$

*For a sufficiently large constant $c_B > 0$ that may depend on $c_\lambda, c_{stop}, c_\alpha$,*

$$B = c_B(\lambda_1\lambda_2)^{-4}\kappa^{910} \tag{14}$$

*Then, with probability at least $1-\delta$, TRAINNETWORKLAYERWISE (Algorithm 1) runs in $O(\kappa^{2394}) = O((nsWM/(\epsilon\delta))^{172368})$ time and samples, and returns trained weights $w$ satisfying that the population loss is bounded to the desired accuracy:*

$$\ell(w) \leq \epsilon.$$

## B.1 Basic definitions

A key concept in our proof will be "blank" neurons and "active" neurons. We say that a neuron is blank if it computes the zero function identically, and also all input and output edges have zero weight:

**Definition B.2.** *A neuron $v \in V \setminus V_{\text{in}}$ is blank at parameters $w = \{a_e\}_{e \in E} \cup \{b_v\}_{v \in V \setminus V_{\text{in}}}$ of the network if:*

- $f_v(x; w) = 0$ *for all $x \in \{-1, 1\}^n$, and*

- $b_v = 0$, *and*

- $a_e = 0$ *for all $e = (u_1, u_2) \in E$ such that $v \in \{u_1, u_2\}$.*

**Definition B.3** (Active neuron). *A neuron $v \in V$ is active if and only if it is not blank.*

We will also often refer to parents of a neuron, which are the neurons that have edges into the neuron:

**Definition B.4** (Parent neurons). *The parents of a neuron $v \in V$ are the set $P_v = \{u : (u, v) \in E\}$.*

Finally, we also define what it means for a neuron to compute a monomial up to certain relative error:

**Definition B.5.** *Let $S \subset [n]$, $w$ be a setting of network parameters, and $v \in V$ be a neuron. We write that neuron $v$ computes $\chi_S(x)$ up to $\epsilon$ relative error if*

$$f_u(x; w) = r\chi_S(x) + h(x)$$

*for some scaling factor $r \in \mathbb{R}$, some function $h : \{-1, 1\}^n \to \mathbb{R}$ such that $\hat{h}(S) = \mathbb{E}_x[h(x)\chi_S(x)] = 0$, and such that $|h(x)| \leq |r|\epsilon$ for all $x$.*

## B.2 Proof organization

Our proof is organized into three modular sections, described below.

**Appendix C: NEURONSGD correctness** In this section, we prove that calling NEURONSGD will with high probability return an approximate stationary point of the loss in polynomial time and samples. The main technical difficulty in this section is to prove that the loss is smooth throughout training. To overcome this, we use the fact that the $L_2$ regularization ensures that the network's parameters are bounded during training.

**Appendix D: TRAINNEURON correctness** In this section, we analyze calls to TRAINNEURON$(v, w)$ when $v$ is a neuron with at most two active parents. Roughly speaking, we prove that if (A) $v$ has two active parents that approximately compute monomials $\chi_{S_1}(x)$ and $\chi_{S_2}(x)$, and (B) the error $\mathbb{E}_x[(f(x; w) - g(x))\chi_{S_1}(x)\chi_{S_2}(x)]$ is large, then after training $v$ approximately computes $\chi_{S_1}(x)\chi_{S_2}(x)$. Otherwise, the neuron $v$ remains blank and all the weights in the network are unchanged. The proofs in this section consist of analyzing of the stationary points of the loss, since NEURONSGD is guaranteed to train to such a stationary point.

**Appendix E: TRAINNETWORKLAYERWISE correctness** In this section, we prove Theorem B.1. We show inductively on the training iteration that during training each neuron is either blank or it approximately represents one of the nonzero monomials of $g$, up to small relative error. Because the network is taken to be quite sparse (see hyperparameter setting above), at any iteration every neuron has at most two active parents. Therefore, the guarantees that we have proved for TRAINNEURON apply to control the progress on each iteration.

## C Correctness of NEURONSGD: finds approximate stationary point

In this section, we show that with high probability NEURONSGD reaches an approximate stationary point of the regularized loss if the minibatch size is a large enough polynomial in the relevant parameters. We now introduce notation used to state and prove the main result of this section.

**Assumption C.1** (Assumptions and notation for Lemma C.2). *The inputs to* NEURONSGD *are a neuron* $v \in V \setminus V_{\text{in}}$, *and an initialization of parameters*

$$w^0 = \{a_e^0\}_{e \in E} \cup \{b_v^0\}_{v \in V \setminus V_{\text{in}}},$$

*such that the following hold:*

- *At initialization, all neurons have magnitude upper-bounded by* $U_{neur} > 1$:

$$\max_{u \in V} \max_{x \in \{-1,1\}^n} |f_u(x; w^0)| \leq U_{neur}.$$

- *Neuron* $v$ *has outward edges' weights equal to zero at initialization: i.e.,*

$$a_e^0 = 0 \text{ for all } e = (v, u) \in E.$$

- *During training, only the subset of parameters*

$$w_v = \{a_e\}_{e=(u,v) \in E} \cup \{b_v\}$$

*corresponding to the inputs to neuron* $v$. *Therefore,*

$$w_{-v}^t = w_{-v}^0 \text{ and } w^t = [w_{-v}^0, w_v^t]$$

*for any iteration* $t$. *In particular,* $w^t = \{a_e^t\}_{e \in E} \cup \{b_v^t\}_{v \in V}$.

- *Let* $T$ *denote the number of iterations, so the method returns*

$$w^T = [w_{-v}^0, w_v^T].$$

- $g : \{-1, 1\}^n \to \mathbb{R}$ *is an* $s$-*sparse polynomial satisfying the* $[1/M, M]$-*staircase property for some* $M \geq 1$, *as in Theorem B.1.*

In all of the results of this section, we assume that Assumption C.1 holds. Now we state the main result of the section. Since the hyperparameters are fixed we omit explicit dependence by writing NEURONSGD$(v, w) :=$ NEURONSGD$(v, w; \lambda_1, \lambda_2, \eta, B, \epsilon_{stop}, \alpha, \tau)$.

**Lemma C.2.** *Consider running* NEURONSGD$(v, w^0)$ *(Algorithm 3) where $v \in V \setminus V_{\text{in}}$ is a neuron, and $w^0$ are the initial parameters of the network.*

*Let $\delta > 0$ and define*
$$t_{max} = \lceil 3\ell_R(w^0)/(\alpha(\epsilon_{stop})^2) \rceil + 1.$$
*Suppose that for some large enough universal constant $C$, the mini-batch size is at least*
$$B \geq C(\lambda_1\lambda_2)^{-3}\kappa^8(U_{neur})^4(1 + \ell_R(w^0)^4)\log(2t_{max}/\delta)/\epsilon_{stop}^2,$$
*and the learning rate is at most*
$$\alpha < 1/(C(\lambda_1\lambda_2)^{-5}\kappa^{16}(U_{neur})^{16}(1 + \ell_R(w^0)^4)).$$
*Then the following hold with probability at least $1 - \delta$:*

1. *The loss does not increase:*
$$\ell_R(w^T) \leq \ell_R(w^0)$$

2. *The output $w^T$ is a $2\epsilon_{stop}$-approximate stationary point of the loss with respect to $w_v$:*
$$\|\nabla_{w_v}\ell_R(w^T)\| \leq 2\epsilon_{stop}$$

3. *The number of iterations of stochastic gradient descent until a stationary point is reached is polynomially-bounded:*
$$T \leq t_{max}$$

The proof is a standard analysis of stochastic gradient descent finding an approximate stationary point of a nonconvex loss. However, care must be taken because the loss is not uniformly smooth: if the parameters of the network grow to infinity, then the gradient of the loss may also grow to infinity. In order to overcome this technical obstacle, we prove that the $L_2$ regularization term ensures that the parameters of the network are bounded during training.

Specifically, we prove inductively on the loop iteration $t \in \{0, \dots, T\}$ that with high probability the loss $\ell_R(w^t)$ does not increase. For the inductive step, we note that the $L_2$ regularization and the upper bound on the loss implies that the parameters are polynomially upper-bounded at each iteration. In turn, this means that the loss is smooth in the neighborhood of the current iterate. And since the current iterate is not close to a stationary point (since otherwise we exit the loop), the loss decreases with high probability, completing the inductive step. The proof is given below, although several auxiliary claims must be proved first.

## C.1 Parameters are bounded by loss

In this section, we prove Claim C.6, which shows that the parameters of the network are polynomially-bounded by the loss during training. First, let us show several auxiliary results.

We observe that training the weights $w_v$ only affects the value of neuron $v$, since all output edges from the neuron $v$ have zero weight:

**Lemma C.3.** *Under Assumption C.1, for any setting of the parameters $w_v = \{a_e\}_{e=(u,v)\in E} \cup \{b_v\}$,*
$$f(x; [w^0_{-v}, w_v]) = f(x; [w^0_{-v}, \vec{0}]) + f_v(x; [w^0_{-v}, w_v]). \tag{15}$$
*And for any neuron $u \neq v \in V$,*
$$f_u(x; [w^0_{-v}, w_v]) = f(x; [w^0_{-v}, \vec{0}]) \tag{16}$$

*Proof.* For any $u \in V \setminus (V_{\text{in}} \cup \{v\})$ we claim that $f_u(x; [w^0_{-v}, w_v]) = f_u(x; [w^0_{-v}, \vec{0}])$. If $u$ is a successor of $v$ then by induction on the depth of $u$, we have that $f_u$ is independent of the value of neuron $f_v$, since all outward edges from $v$ have zero weight under $w^0_{-v}$ by Assumption C.1. On the other hand, if $u$ is not a successor of $v$ then it is independent of $w_v$.

Finally, $f_v(x; [w^0_{-v}, \vec{0}]) = 0$ for all $x \in \{-1, 1\}^n$ because all the edges to $v$ have zero weight, and $v$ has zero bias. So $f_v(x; [w^0_{-v}, w_v]) = f_v(x; [w^0_{-v}, \vec{0}]) + f_v(x; [w^0_{-v}, \vec{0}])$. Eq. (16) follows from recalling the definition $f(x; [w^0_{-v}, w_v]) = \sum_{u \in V \setminus V_{\text{in}}} f_u(x; [w^0_{-v}, w_v])$. $\square$

In order to prove Lemma C.2, we must first prove several auxiliary claims.

**Claim C.4.** *Under Assumption C.1, for any neuron $u \neq v \in V$ and setting $w_v = \{a_e\}_{e=(u,v)\in E} \cup \{b_v\}$ of the parameters of neuron $v$, define the set of network parameters $w = [w^0_{-v}, w_v]$. Then*

$$|f_u(x; w)| \leq U_{neur}.$$

*Proof.* By Eq. (16), $|f_u(x; w)| = |f_u(x; [w^0_{-v}, \vec{0}])| = |f_u(x; [w^0_{-v}, w^0])| = |f_u(x; w^0)| \leq U_{neur}$. $\square$

**Claim C.5.** *Suppose that $g$ is an $s$-sparse polynomial satisfying the $[1/M, M]$-staircase property, as in Assumption C.1. Then, $\max_x |g(x)| \leq Ms$.*

*Proof.* For any $x \in \{-1, 1\}^n$, by the Hölder inequality,

$$|g(x)| = |\sum_{S \subset [n]} \hat{g}(S)\chi_S(x)| \leq |\{\hat{g}(S) \neq 0\}| \cdot \max_S |\hat{g}(S)| \leq Ms.$$

$\square$

**Claim C.6** (Parameters are upper-bounded during training). *For any setting $w_v = \{a_e\}_{e=(u,v)\in E} \cup \{b_v\}$ of the parameters of neuron $v$, define the set of network parameters $w = [w^0_{-v}, w_v]$. Then*

$$\max_{e \in E} |a_e| \lesssim (\lambda_1\lambda_2)^{-1/2}\sqrt{\ell_R(w)} \tag{17}$$

$$|b_v| \lesssim (\lambda_1\lambda_2)^{-1}\kappa^2(U_{neur})^2\ell_R(w) \tag{18}$$

*In particular, we obtain the following bound for the parameters $w_v$ associated with neuron $v$:*

$$\|w_v\|_\infty \lesssim (\lambda_1\lambda_2)^{-1}\kappa^2(U_{neur})^2 \max(1, \ell_R(w)).$$

*Proof.* For any $e \in E$, the bound on $|a_e|$ follows because of the $L_2$ regularization term

$$\frac{1}{2}\lambda_1\lambda_2 \cdot (a_e)^2 \leq \frac{1}{2}\lambda_1 \cdot (a_e)^2 \leq R(w) \leq \ell_R(w).$$

We now prove the bound on $|b_v|$, using the above bound on $|a_e|$. For any $x \in \{-1, 1\}^n$,

$$|f(x; w)| \geq |f_v(x; w)| - \sum_{u \in V \setminus (V_{in} \cup \{v\})} |f_u(x; w)|$$

$$\geq |f_v(x; w)| - WLU_{neur} \qquad \text{using Claim C.4}$$

$$\geq |f_v(x; w)| - \kappa U_{neur}$$

Furthermore, for any $x \in \{-1, 1\}^n$, recall that $f_v(x; w) = \left(\sum_{e=(u,v)\in E} a_e f_u(x; w)\right)^2 + b_v$, so

$$|f_v(x; w)| \geq |b_v| - (|\{(u, v) \in E\}| \cdot \max_e |a_e| \cdot \max_{u \in V \setminus \{v\}} |f_u(x; w)|)^2$$

$$\geq |b_v| - ((2W)2(\lambda_1\lambda_2)^{-1/2} \cdot \sqrt{\ell_R(w)} \cdot U_{neur})^2 \qquad \text{using Eq. (17)}$$

$$\geq |b_v| - 16(\lambda_1\lambda_2)^{-1}\kappa^2(U_{neur})^2\ell_R(w)$$

Recall from Claim C.5 that $|g(x)| \leq Ms \leq \kappa$ for any $x \in \{-1, 1\}^n$. This implies

$$\ell_R(w) \geq \mathbb{E}_{x\sim\{-1,1\}^n}\frac{1}{2}(f(x; w) - g(x))^2$$

$$\geq \mathbb{E}_{x\sim\{-1,1\}^n}\frac{1}{2}(\max(0, |f_v(x; w)| - \kappa U_{neur} - \kappa))^2$$

$$\geq (|b_v| - 16(\lambda_1\lambda_2)^{-1}\kappa^2(U_{neur})^2\ell_R(w) - 2\kappa U_{neur})^2,$$

so we must have

$$|b_v| \leq \sqrt{\ell_R(w)} + 16(\lambda_1\lambda_2)^{-1}\kappa^2(U_{neur})^2\ell_R(w) - 2\kappa U_{neur}$$

$$\lesssim (\lambda_1\lambda_2)^{-1}\kappa^2(U_{neur})^2\ell_R(w).$$

$\square$

## C.2 Stochastic gradient approximation is close during training

The main result of this section is Claim C.10, which proves that if the loss is bounded during training, then the stochastic gradient approximations $\xi^t$ are close to the true gradients with high probability.

First, we prove that if the parameters associated with the neuron $v$ are bounded, then the magnitude of the function computed by the network at each neuron is polynomially upper-bounded:

**Claim C.7** (Neurons are upper-bounded during training). *Under Assumption C.1, for any setting $w_v$ of the parameters of neuron $v$, define the network parameters $w = [w^0_{-v}, w_v]$. Then,*

$$\max_{u \in V} \max_{x \in \{-1,1\}^n} |f_u(x;w)| \lesssim (\lambda_1 \lambda_2)^{-1} \kappa^2 (U_{neur})^2 \ell_R(w).$$

*Proof.* The bound holds for all $u \in V \setminus \{v\}$ by Claim C.4. For $v$, recall that $f_v(x;w) = \left( \sum_{e=(u,v) \in E} a_e f_u(x;w) \right)^2 + b_v$, so

$$
\begin{aligned}
|f_v(x;w)| &\leq (|\{(u,v) \in E\}| \cdot \max_{e=(u,v)} |a_e| \cdot \max_{u \in V \setminus \{v\}} |f_u(x;w)|)^2 + |b_v| \\
&\lesssim (2W \cdot (\lambda_1 \lambda_2)^{-1/2} \sqrt{\ell_R(w)} \cdot U_{neur})^2 + (\lambda_1 \lambda_2)^{-1} \kappa^2 (U_{neur})^2 \ell_R(w) \quad \text{by Claim C.6} \\
&\lesssim (\lambda_1 \lambda_2)^{-1} \kappa^2 (U_{neur})^2 \ell_R(w).
\end{aligned}
$$

$\square$

**Claim C.8** (Gradient of neuron $v$ is upper-bounded during training). *Under Assumption C.1, for any setting $w_v$ of the parameters of neuron $v$, the gradient of $f_v$ with respect to $w_v$ is bounded:*

$$\max_{x \in \{-1,1\}^n} \|\nabla_{w_v} f_v(x;w)\|_\infty \lesssim \max(\kappa U_{neur} \max_{e=(u,v) \in E} |a_e|, 1).$$

*Proof.* For any $x \in \{-1,1\}^n$,

$$
\begin{aligned}
\|\nabla_{w_v} f_v(x;w)\|_\infty &= \|\nabla_{w_v}((\sum_{e=(u,v) \in E} a_e f_u(x;w))^2 + b_v)\|_\infty \\
&= \max \left( \left( 2 \sum_{e=(u,v) \in E} a_e f_u(x;w) \right) \cdot \max_{e=(u,v) \in E} |f_u(x;w)|, 1 \right) \\
&\leq \max(4W U_{neur} \max_{e=(u,v) \in E} |a_e|, 1) \\
&\lesssim \max(\kappa U_{neur} \max_{e=(u,v) \in E} |a_e|, 1)
\end{aligned}
$$

$\square$

**Claim C.9** (Gradient of loss is upper-bounded during training). *Under Assumption C.1, for any $x \in \{-1,1\}^n$ and for any setting $w_v$ of the parameters of neuron $v$, the gradient of the loss with respect to $w_v$ is bounded. Namely, defining the set of network parameters $w = [w^0_{-v}, w_v]$, we have*

$$\|\nabla_{w_v} \ell_R(x;w)\|_\infty \lesssim (\lambda_1 \lambda_2)^{-3/2} \kappa^4 (U_{neur})^3 \max(\ell_R(w)^2, 1).$$

*Proof.* For the subsequent arguments, define the "error" function

$$\zeta(x;w) = f(x;w) - g(x).$$

This is the gap between the learned function $f$ from the true function $g$. The definition of $\zeta$ allows us to write the gradient of the unregularized loss at $x \in \{-1,1\}^n$ as:

$$
\begin{aligned}
\nabla_{w_v} \ell(x;w) &= \frac{1}{2} \nabla_{w_v} (f(x;w) - g(x))^2 \\
&= \frac{1}{2} \nabla_{w_v} (\zeta(x;[w^0_{-v}, \vec{0}]) + f_v(x;w))^2 \qquad \text{by Eq. (15)} \\
&= (\zeta(x;[w^0_{-v}, \vec{0}]) + f_v(x;w)) \nabla_{w_v} f_v(x;w)
\end{aligned}
$$

So we may upper-bound the gradient of the unregularized loss at $x$ by:

$\|\nabla_{w_v}\ell(x;w)\|_\infty$

$$\leq |\zeta(x;[w^0_{-v},\vec{0}])| + f_v(x;w)|\cdot\|\nabla_{w_v}f_v(x;w)\|_\infty$$

$$= |f(x;w)-g(x)|\cdot\|\nabla_{w_v}f_v(x;w)\|_\infty, \qquad\qquad \text{by Eq. (15)}$$

$$\lesssim (|f(x;w)-g(x)|)\max(\kappa U_{neur}\max_{e=(u,v)\in E}|a_e|,1) \qquad\qquad \text{by Claim C.8}$$

$$\lesssim (|f(x;w)-g(x)|)\max((\lambda_1\lambda_2)^{-1/2}\kappa U_{neur}\sqrt{\ell_R(w)},1) \qquad\qquad \text{by Claim C.6}$$

$$\lesssim (|f(x;w)|+Ms)\max((\lambda_1\lambda_2)^{-1/2}\kappa U_{neur}\sqrt{\ell_R(w)},1) \qquad\qquad \text{by Claim C.5}$$

$$\lesssim ((\lambda_1\lambda_2)^{-1}\kappa^3(U_{neur})^2\ell_R(w)+Ms)\max((\lambda_1\lambda_2)^{-1/2}\kappa U_{neur}\sqrt{\ell_R(w)},1) \quad \text{by Claim C.7}$$

$$\lesssim (\lambda_1\lambda_2)^{-3/2}\kappa^4(U_{neur})^3\max(\ell_R(w)^2,1)$$

Finally, the triangle inequality implies an upper-bound on the gradient of the regularized loss at $x$:

$$\|\nabla_{w_v}\ell_R(x;w)\|_\infty \leq \|\nabla_{w_v}\ell(x;w)\|_\infty + \max(\lambda_1,\lambda_2)\max_{e=(u,v)\in E}|a_e|$$

$$\lesssim (\lambda_1\lambda_2)^{-3/2}\kappa^4(U_{neur})^3\max(\ell_R(w)^2,1),$$

by Claim C.6, and using $0 < \lambda_1,\lambda_2 \leq 1$. $\qquad\qquad\square$

Finally, we use the above bounds to prove that with high probability the stochastic gradients computed by NEURONSGD are close to the true gradients if the minibatch size is taken to be a large enough polynomial.

**Claim C.10** (Stochastic gradient approximation is close during training). *Under Assumption C.1, there is a large enough constant $C$ such that for any $\epsilon > 0, \delta > 0$ and iteration $0 \leq t \leq T$, if*

$$B \geq C(\lambda_1\lambda_2)^{-3}\kappa^8(U_{neur})^4\log(1/\delta)\max(\ell_R(w^t)^4,1)/\epsilon^2,$$

*then*

$$\mathbb{P}[\|\xi^t - \nabla_{w_v}\ell_R(w^t)\| > \epsilon \mid w^t] \leq \delta.$$

*Proof.* Recall the definition of $\xi^t$ from Lines 3 and 4 of Algorithm 3. Namely, draw i.i.d. $x^{t,1},\dots,x^{t,B} \sim \{-1,1\}^n$, and define the random variable $\xi^t$ as follows:

$$\xi^t = \frac{1}{B}\sum_{i=1}^B \nabla_{w_v}\ell_R(x^{t,i};w^t).$$

By linearity of expectation and differentiation, $\xi^t$ is an unbiased estimator of the true gradient:

$$\mathbb{E}[\xi^t] = \frac{1}{B}\sum_{i=1}^B \nabla_{w_v}\mathbb{E}[\ell_R(x^{t,i};w^t)] = \nabla_{w_v}\mathbb{E}_{x\sim\{-1,1\}^n}[\ell_R(x;w^t)] = \nabla_{w_v}\ell_R(w^t).$$

So it suffices to prove that $\xi^t$ concentrates around its mean. We will use the Hoeffding inequality.

$$\mathbb{P}[\|\xi^t - \nabla_{w_v}\ell_R(x;w^t)\| > \epsilon \mid w^t] \leq \mathbb{P}[\|\xi^t - \nabla_{w_v}\ell_R(x;w^t)\|_\infty > \epsilon/\sqrt{2W} \mid w^t]$$

$$\leq (2W)\exp(-2B\epsilon^2/(2\sqrt{2W}\max_x\|\nabla_{w_v}\ell_R(x;w^t)\|_\infty)^2).$$

The first inequality uses that $\xi^t$ is of length at most $2W$, since there are at most $W + n + 1 \leq 2W$ parameters associated with the neuron $v$, since there are at most $W$ edges to $v$ from the previous layer, at most $n$ edges to $v$ from the inputs, and one bias parameter. The second inequality is the Hoeffding bound. Therefore, the inequality

$$\mathbb{P}_{x\sim\{-1,1\}^n}[\|\xi - \nabla_{w_v}\ell_R(x;w^t)\| > \epsilon] \leq \delta$$

follows by using Claim C.9 to upper bound $\|\nabla_{w_v}\ell_R(x;w^t)\|_\infty$, and choosing the constant $C$ in the statement of the claim large enough. $\qquad\qquad\square$

## C.3 Objective is smooth during training

The final claim bounds the smoothness of the loss function at each iterate during training. This is needed to prove that with high probability NEURONSGD does not increase the loss.

**Claim C.11.** *Under Assumption C.1, given a setting of parameters $w_v$ for neuron $v$, let $w = [w^0_{-v}, w]$. Then the Hessian of the loss at $w$ with respect to the parameters $w_v$ has bounded norm*

$$\|\nabla^2_{w_v} \ell_R(w)\| \lesssim (\lambda_1 \lambda_2)^{-1} \kappa^4 (U_{neur})^4 \max(\ell_R(w), 1).$$

*Proof.* For the subsequent arguments, define the error function

$$\zeta(x; w) = f(x; w) - g(x),$$

in the same way as defined for the proof of Claim C.9. We now write the Hessian of the loss with respect to the parameters $w_v = \{a_e\}_{e=(u,v)\in E} \cup \{b_v\}$ at any point $x \in \{-1, 1\}^n$. For any $e = (u, v), e' = (u', v) \in E$:

$$\frac{\partial^2 \ell(x; w)}{\partial a_{e'} \partial a_e}$$

$$= \frac{\partial}{\partial a_{e'}} \frac{\partial \ell(x; w)}{\partial a_e}$$

$$= \frac{\partial}{\partial a_{e'}} ((\zeta(x; [w^0_{-v}, \vec{0}]) + f_v(x; w)) \frac{\partial f_v(x; w)}{\partial a_e})$$

$$= \left( \frac{\partial f_v(x; w)}{\partial a_{e'}} \right) \left( \frac{\partial f_v(x; w)}{\partial a_e} \right) + (\zeta(x; [w^0_{-v}, \vec{0}]) + f_v(x; w)) \cdot \left( \frac{\partial^2 f(x; w)}{\partial a_{e'} \partial a_e} \right)$$

$$= \left( \frac{\partial f_v(x; w)}{\partial a_{e'}} \cdot \frac{\partial f_v(x; w)}{\partial a_e} \right)$$

$$\quad + (\zeta(x; [w^0_{-v}, \vec{0}]) + f_v(x; w)) \cdot \left( \frac{\partial}{\partial a_{e'}} 2 \sum_{e''=(u'',v)\in E} a_{e''} f_{u''}(x; w) f_u(x; w) \right)$$

$$= \left( \frac{\partial f_v(x; w)}{\partial a_{e'}} \cdot \frac{\partial f_v(x; w)}{\partial a_e} \right) + (\zeta(x; [w^0_{-v}, \vec{0}]) + f_v(x; w)) \cdot (2 f_{u'}(x; w) f_u(x; w))$$

So

$$\left| \frac{\partial^2 \ell(x; w)}{\partial a_{e'} \partial a_e} \right|$$

$$\leq \left| \frac{\partial f_v(x; w)}{\partial a_{e'}} \cdot \frac{\partial f_v(x; w)}{\partial a_e} \right| + 2|\zeta(x; [w^0_{-v}, \vec{0}]) + f_v(x; w)| \cdot |f_{u'}(x; w) f_u(x; w)|$$

$$\leq \max_{e''=(u'',v)\in E} \left| \frac{\partial f_v(x; w)}{\partial a_{e''}} \right|^2 + 2|\zeta(x; [w^0_{-v}, \vec{0}]) + f_v(x; w)| \cdot (U_{neur})^2 \qquad \text{by Claim C.4}$$

$$\lesssim \max(\kappa^2 (U_{neur})^2 \max_{e''=(u'',v)\in E} |a_{e''}|^2, 1) + 2|\zeta(x; [w^0_{-v}, \vec{0}]) + f_v(x; w)| \cdot (U_{neur})^2 \qquad \text{by Claim C.8}$$

$$\lesssim (\lambda_1 \lambda_2)^{-1} \kappa^2 (U_{neur})^2 \max(\ell_R(w), 1) + 2|\zeta(x; [w^0_{-v}, \vec{0}]) + f_v(x; w)| \cdot (U_{neur})^2 \qquad \text{by Claim C.6}$$

$$= (\lambda_1 \lambda_2)^{-1} \kappa^2 (U_{neur})^2 \max(\ell_R(w), 1) + 2|f(x; w) - g(x)| \cdot (U_{neur})^2 \qquad \text{by Eq. (15)}$$

$$\leq (\lambda_1 \lambda_2)^{-1} \kappa^2 (U_{neur})^2 \max(\ell_R(w), 1) + 2|f(x; w)| \cdot (U_{neur})^2 + 2Ms \cdot (U_{neur})^2 \qquad \text{by Claim C.5}$$

$$\lesssim (\lambda_1 \lambda_2)^{-1} \kappa^2 (U_{neur})^2 \max(\ell_R(w), 1) + 2(\lambda_1 \lambda_2)^{-1} \kappa^3 (U_{neur})^4 \ell_R(w) + 2Ms \cdot (U_{neur})^2 \qquad \text{by Claim C.7}$$

$$\lesssim (\lambda_1 \lambda_2)^{-1} \kappa^3 (U_{neur})^4 \max(\ell_R(w), 1).$$

Similarly, for any $e = (u, v) \in E$,

$$
\begin{aligned}
\left| \frac{\partial^2 \ell(x; w)}{\partial a_e \partial b_v} \right| &= \left| \frac{\partial}{\partial b_v} ((\zeta(x; [w^0_{-v}, \vec{0}]) + f_v(x; w)) \frac{\partial f_v(x; w)}{\partial a_e}) \right| \\
&= \left| \frac{\partial f_v(x; w)}{\partial a_e} \cdot \frac{\partial f_v(x; w)}{\partial b_v} \right| \\
&= \left| \frac{\partial f_v(x; w)}{\partial a_e} \right| && \text{since } \frac{\partial f_v(x;w)}{\partial b_v} = 1 \\
&\lesssim \max(\kappa U_{neur} \max_{e' = (u', v) \in E} |a_{e'}|, 1) && \text{by Claim C.8} \\
&\lesssim (\lambda_1 \lambda_2)^{-1/2} \kappa (U_{neur})^2 \max(\sqrt{\ell_R(w)}, 1) && \text{by Claim C.6.}
\end{aligned}
$$

And similarly:

$$
\left| \frac{\partial^2 \ell(x; w)}{\partial b_v \partial b_v} \right| = \left| \frac{\partial f_v(x; w)}{\partial b_v} \cdot \frac{\partial f_v(x; w)}{\partial b_v} \right| = 1.
$$

Finally, this allows us to bound the operator norm of the Hessian of the regularized loss:

$$
\begin{aligned}
\| \nabla^2_{w_v} \ell_R(w) \| &= \| \mathbb{E}[\nabla^2_{w_v} \ell_R(x; w)] \| \\
&\leq \| \mathbb{E}[\nabla^2_{w_v} \ell(x; w)] \| + \max(\lambda_1, \lambda_2) \\
&= \mathbb{E}[\max_{\phi : \|\phi\| = 1} \phi^T (\nabla^2_{w_v} \ell(x; w)) \phi] + \max(\lambda_1, \lambda_2) && \text{by the Courant-Fischer Theorem} \\
&\lesssim 2W (\lambda_1 \lambda_2)^{-1} \kappa^3 (U_{neur})^4 \max(\ell_R(w), 1) && \text{since } w_v \text{ has length at most } 2W \\
&\lesssim (\lambda_1 \lambda_2)^{-1} \kappa^4 (U_{neur})^4 \max(\ell_R(w), 1).
\end{aligned}
$$

Note that we use that $w_v$ has length at most $2W$, which is true since there are at most $W + n < 2W$ edge parameters and 1 bias parameter associated with neuron $v$. $\qquad\square$

## C.4 Loss decreases if gradient approximation is good

In this section, we prove Claim C.14, which shows that if the gradient approximation $\xi^t$ on iteration $t$ in NEURONSGD is sufficiently accurate, then the loss decreases. In order to show this, we first prove a claim that is essentially a converse of Claim C.6: namely, we show that if the parameters $w_v$ of neuron $v$ are upper-bounded, then the loss is upper-bounded as well.

**Claim C.12** (Bounded change in parameters implies bounded change in loss). *For a given setting $w_v$ of the parameters of neuron $v$, define the network parameters $w = [w^0_{-v}, w_v]$. Furthermore, for any real-valued vector of parameters $\mu$ of the same length as $w_v$, define $w'_v = w_v + \mu$ and $w' = [w^0_{-v}, w'_v]$. Then the following holds:*

$$
\ell_R(w') \lesssim (\lambda_1 \lambda_2)^{-4} \kappa^{12} (U_{neur})^{12} (1 + \ell_R(w)^4 + \|\mu\|_\infty^4)
$$

*Proof.*

$$\ell(w') = \mathbb{E}_{x \sim \{-1,1\}^n} \ell(x; w')$$

$$= \frac{1}{2} \mathbb{E}_{x \sim \{-1,1\}^n} \left( f(x; w') - g(x) \right)^2$$

$$= \frac{1}{2} \mathbb{E}_{x \sim \{-1,1\}^n} \left( f(x; [w^0_{-v}, 0]) + f_v(x; w') - g(x) \right)^2 \qquad \text{by Eq. (15)}$$

$$\leq \mathbb{E}_{x \sim \{-1,1\}^n} \left( Ms + WLU_{neur} + |f_v(x; w')| \right)^2 \qquad \text{by Claim C.5 and Claim C.4}$$

$$\lesssim \mathbb{E}_{x \sim \{-1,1\}^n} \left( \kappa U_{neur} + |f_v(x; w')| \right)^2$$

$$\lesssim \kappa^2 U_{neur}^2 + \mathbb{E}_{x \sim \{-1,1\}^n} |f_v(x; w')|^2$$

$$\lesssim \kappa^2 U_{neur}^2 + \mathbb{E}_{x \sim \{-1,1\}^n} \left( |b'_v| + \left( \sum_{e=(u,v)\in E} a'_e f_u(x; w') \right)^2 \right)^2$$

$$\lesssim \kappa^2 U_{neur}^2 + \left( (2W)^2 \|w'_v\|_\infty^2 U_{neur}^2 \right)^2 \qquad \text{by Claim C.4}$$

$$\lesssim \kappa^4 U_{neur}^4 \max(1, \|w'_v\|_\infty^4)$$

$$\lesssim \kappa^4 U_{neur}^4 (1 + \|w_v\|_\infty^4 + \|\mu\|_\infty^4).$$

Furthermore, by Claim C.6,

$$\|w_v\|_\infty \lesssim (\lambda_1 \lambda_2)^{-1} \kappa^2 (U_{neur})^2 \max(1, \ell_R(w)).$$

So

$$\ell_R(w') - \ell_R(w) \leq \ell(w') + \frac{1}{2} \sum_e \max(\lambda_1, \lambda_2) \|w'_v\|_\infty^2$$

$$\lesssim \kappa^4 U_{neur}^4 \max(\|w_v\|_\infty^4 + \|\mu\|_\infty^4) + (2W) \max(\lambda_1, \lambda_2)(\|w_v\|_\infty^2 + \|\mu\|_\infty^2)$$

$$\lesssim (\lambda_1 \lambda_2)^{-4} \kappa^{12} (U_{neur})^{12} (1 + \ell_R(w)^4 + \|\mu\|_\infty^4),$$

using $\lambda_1, \lambda_2 \leq 1$ for the last line. $\qquad \square$

**Definition C.13.** *For any iteration $0 \leq t \leq T$, let $E_{good,t}$ be the event that for all $0 \leq t' \leq t$ we have*

$$\|\xi^{t'} - \nabla_{w_v} \ell_R(w^{t'})\| \leq \epsilon_{stop}/3.$$

**Claim C.14.** *Suppose that the learning rate $\alpha$ satisfies $\alpha < (\lambda_1 \lambda_2)^5 / (C\kappa^{16}(U_{neur})^{16}(1+\ell_R(w^0)^4))$ for some large enough universal constant $C$, and let $0 \leq t \leq T - 1$. If $E_{good,t}$ holds, then*

$$\ell_R(w^{t+1}) \leq \ell_R(w^t) - \alpha \|\xi^t\|^2/3.$$

*Proof.* The proof is by induction on $t$. For any $0 \leq t < T$, suppose that $E_{good,t}$ holds. By Taylor's theorem there is $\theta_t \in (0, 1)$ such that:

$$\ell_R(w^{t+1}) = \ell_R(w^t) - \alpha \xi^t \cdot \nabla_{w_v} \ell_R(w^t) + \frac{\alpha^2}{2} ((\nabla^2_{w_v} \ell_R(\theta_t w^t + (1 - \theta_t) w^{t+1})) \xi^t) \cdot \xi^t$$

We note that $E_{good,t}$ implies

$$\xi^t \cdot \nabla_{w_v} \ell_R(w^t) \geq \|\xi^t\|^2 - \epsilon_{stop} \|\xi^t\|/3 \geq (2/3) \|\xi^t\|^2,$$

where the second inequality is due to $\|\xi^t\| > \epsilon_{stop}$, because $t < T$.

Furthermore, we note that

$$\ell_R(w^t) \leq \ell_R(w^0).$$

If $t = 0$ then the above inequality is trivial, and if $t \geq 1$ then it is true by the inductive hypothesis, since $E_{good,t}$ implies $E_{good,t-1}$. This allows us to prove that the loss is smooth in the neighborhood

of $w^t$:

$$|((\nabla^2_{w_v}\ell_R(\theta_t w^t + (1-\theta_t)w^{t+1}))\xi^t)\cdot\xi^t|$$
$$\leq \|\nabla^2_{w_v}\ell_R(\theta_t w^t + (1-\theta_t)w^{t+1})\|\|\xi^t\|^2$$
$$\lesssim (\lambda_1\lambda_2)^{-1}\kappa^4(U_{neur})^4\max(\ell_R(\theta_t w^t + (1-\theta_t)w^{t+1}),1)\|\xi^t\|^2 \qquad \text{by Claim C.11}$$
$$= (\lambda_1\lambda_2)^{-1}\kappa^4(U_{neur})^4\max(\ell_R([w^0_{-v}, w^t_v + \alpha(1-\theta_t)\xi^t]),1)\|\xi^t\|^2$$
$$\lesssim (\lambda_1\lambda_2)^{-5}\kappa^{16}(U_{neur})^{16}(1 + \ell_R(w^t)^4 + \|\alpha(1-\theta_t)\xi^t\|^4_\infty)\|\xi^t\|^2 \qquad \text{by Claim C.12}$$

So

$$|((\nabla^2_{w_v}\ell_R(\theta_t w^t + (1-\theta_t)w^{t+1}))\xi^t)\cdot\xi^t|(\lambda_1\lambda_2)^5\kappa^{-16}(U_{neur})^{-16}$$
$$\lesssim (1 + \ell_R(w^t)^4 + \|\alpha\xi^t\|^4_\infty)\|\xi^t\|^2$$
$$\lesssim (1 + \ell_R(w^t)^4 + (\alpha\epsilon_{stop})^4 + \|\alpha\nabla_{w_v}\ell_R(w^t)\|^4_\infty)\|\xi^t\|^2 \qquad \text{by } E_{good,t}$$
$$\lesssim (1 + \ell_R(w^t)^4 + \|\alpha\nabla_{w_v}\ell_R(w^t)\|^4_\infty)\|\xi^t\|^2 \qquad \text{by } \alpha, \epsilon_{stop} \leq 1$$
$$\lesssim (1 + \ell_R(w^t)^4 + (\alpha(\lambda_1\lambda_2)^{-3/2}\kappa^4(U_{neur})^3\max(\ell_R(w^t)^2,1))^4)\|\xi^t\|^2 \qquad \text{by Claim C.9}$$
$$\lesssim (1 + \ell_R(w^t)^4)\|\xi^t\|^2,$$

where the last line is by making the learning rate $\alpha$ small enough that it satisfies $\alpha \leq (\lambda_1\lambda_2)^{3/2}\kappa^{-4}(U_{neur})^{-3}\min(\ell_R(w^t)^{-2},1)$. So plugging these bounds back into Taylor's theorem:

$$\ell_R(w^{t+1}) \leq \ell_R(w^t) - \alpha((2/3) - \alpha C(\lambda_1\lambda_2)^{-5}\kappa^{16}(U_{neur})^{16}(1 + \ell_R(w^t)^4))\|\xi^t\|^2,$$

where $C$ is some universal constant. Taking $\alpha < (\lambda_1\lambda_2)^5/(3C\kappa^{16}(U_{neur})^{16}(1 + \ell_R(w^0)^4)) \leq (\lambda_1\lambda_2)^5/(3C\kappa^{16}(U_{neur})^{16}(1 + \ell_R(w^t)^4))$, we conclude that

$$\ell_R(w^{t+1}) \leq \ell_R(w^t) - \alpha\|\xi^t\|^2/3.$$

$\square$

## C.5  Proof of Lemma C.2

We now combine the above claims to prove the main result of this section: i.e., that NEURONSGD returns an approximate stationary point in a polynomial number of iterations.

*Proof of Lemma C.2.* Recall that $t_{max} = \lceil 3\ell_R(w^0)/(\alpha(\epsilon_{stop})^2)\rceil + 1$. We make the following claim:

**Claim C.15.** *Under the setting of Assumption C.1, suppose that the learning rate $\alpha$ satisfies*

$$\alpha < 1/(C(\lambda_1\lambda_2)^{-5}\kappa^{16}(U_{neur})^{16}(1 + \ell_R(w^0)^4))$$

*and that the minibatch size is at least*

$$B \geq C(\lambda_1\lambda_2)^{-3}\kappa^8(U_{neur})^4(1 + \ell_R(w^0)^4)\log(2t_{max}/\delta)/\epsilon^2_{stop}$$

*for some large enough universal constant C. Then, for any $t \geq 0$ we have*

$$\mathbb{P}[E_{good,\min(t+1,T)} \mid E_{good,\min(t,T)}] \geq 1 - \delta/(t_{max} + 1).$$

*Proof.* We split into two cases. If $T \leq t$:

$$\mathbb{P}[E_{good,\min(t+1,T)} \mid E_{good,\min(t,T)} \text{ and } T \leq t] = \mathbb{P}[E_{good,\min(t,T)} \mid E_{good,\min(t,T)} \text{ and } T \leq t] = 1.$$

Otherwise, if $T > t$, then

$$\mathbb{P}[E_{good,\min(t+1,T)} \mid E_{good,\min(t,T)} \text{ and } T > t]$$
$$= \mathbb{P}[\|\xi^{t+1} - \nabla_{w_v}\ell_R(w^{t+1})\| \leq \epsilon_{stop}/3 \mid E_{good,\min(t,T)} \text{ and } T > t]$$
$$= \mathbb{P}[\|\xi^{t+1} - \nabla_{w_v}\ell_R(w^{t+1})\| \leq \epsilon_{stop}/3 \mid E_{good,\min(t,T)} \text{ and } T > t \text{ and } \ell_R(w^{t+1}) \leq \ell_R(w^0)]$$
$$\geq 1 - \delta/(t_{max} + 1)$$

where in the second-to-last inequality we used Claim C.14, and in the last inequality we used Claim C.10 and the fact that $\xi^{t+1}$ is independent of $w^0, \ldots, w^t$ conditioned on $w^{t+1}$. $\square$

Combining Claim C.15 with a union bound for all $t \in \{0, \ldots, t_{max}\}$, and noting that Claim C.10 implies $\mathbb{P}[E_{good,0}] \geq 1 - \delta/(t_{max} + 1)$, it follows that

$$\mathbb{P}[E_{good,\min(t_{max},T)}] \geq 1 - \delta.$$

We claim that if $E_{good,\min(t_{max},T)}$ holds, then we must have $T \leq t_{max}$. Indeed, otherwise, the event $E_{good,t_{max}}$ holds, so applying Claim C.14 we must have

$$\ell_R(w^{t_{max}+1}) \leq \ell_R(w^0) - \alpha \sum_{t=0}^{t_{max}} \|\xi^t\|^2/3 \leq \ell_R(w^0) - \alpha t_{max}(\epsilon_{stop})^2/3 < 0,$$

which is a contradiction because the loss cannot be negative. Therefore, we conclude that:

$$\mathbb{P}[E_{good,T} \text{ and } T \leq t_{max}] \geq 1 - \delta.$$

If $E_{good,T}$ holds, Claim C.14 implies that $\ell_R(w^T) \leq \ell_R(w^0)$. Furthermore, under the event $E_{good,T}$ we must have $\|\xi^T\| \leq \epsilon_{stop}$, so $\|\nabla_{w_v} \ell_R(w^T)\| \leq \epsilon_{stop} + \epsilon_{stop}/3 \leq 2\epsilon_{stop}$. $\qquad\square$

## D  Correctness of TRAINNEURON: learns product of inputs

The main results of this section are Lemmas D.7, D.8 and D.11 to D.13, which control how TRAINNEURON (Algorithm 2) updates individual neurons during the training of the entire network. Because of the sparsity of the network, in this section we only reason about how TRAINNEURON updates neurons with at most two active inputs. These will be the only results that will be needed to prove correctness of TRAINNETWORKLAYERWISE in Appendix E. We also assume that each of the neurons in the previous layers is either blank (i.e., always computes zero), or it represents a monomial $\chi_S(x)$ up to some small relative error, since this will hold true inductively on the training iteration.

Suppose that we train an initially blank neuron $v$ by running TRAINNEURON$(v, w^0) :=$ TRAINNEURON$(v, w^0; \lambda_1, \lambda_2, \eta, B, \epsilon_{stop}, \alpha, \tau)$. If $v$ has at most one active parent, then we prove in Lemma D.7 that with high probability $v$ remains blank after training. This is because by analyzing the stationary points of the loss one can see that the $L_2$ regularization term sends the weights of the input edges to $v$ to close to zero, and these are rounded to exactly zero in Line 4 of TRAINNEURON.

If instead $v$ has two active parents $u_1$ and $u_2$, then the situation is more delicate. Suppose in this case that $u_1$ approximately computes a monomial $\chi_{S_1}(x)$, and $u_2$ approximately computes a monomial $\chi_{S_2}(x)$. We prove that training the neuron $v$ allows it to approximately compute the product of these two inputs: i.e., the monomial $\chi_{S_1}(x)\chi_{S_2}(x)$. If the error function does not have a large component in the direction of $\chi_{S_1}(x)\chi_{S_2}(x)$, then the $L_2$ regularization will again prevail and send the input weights to zero, and $v$ will remain blank after training (proved in Lemma D.8). On the other hand, if the error function does have a large component in the direction of $\chi_{S_1}(x)\chi_{S_2}(x)$ then the regularization will be relatively insignificant to the decrease in the loss from learning $\chi_{S_1}(x)\chi_{S_2}(x)$, and so with lower-bounded probability the neuron $v$ will learn to approximately compute $\chi_{S_1}(x)\chi_{S_2}(x)$ (proved in Lemmas D.11 to D.13). Thus, training neuron $v$ computes a monomial equal to the product of monomials computed by neurons at lower depth only if it significantly decreases the loss, and so this ensures that a bounded number of neurons in the network are active during training.

We also note that an obstacle to applying TRAINNEURON to train the network layerwise is the possible exponential error blow-up along the depth in the approximation of each neuron computed by the monomials. In order to overcome this obstacle, we must carefully bound the blow-up in the relative error of the new neuron created. For this, we roughly prove in Lemma D.12 that if neuron $u_1$ has relative error $\epsilon_1$, and neuron $u_2$ is an input in $V_{in}$ and therefore has relative error $\epsilon_2 = 0$, then the new neuron trained will have relative error at most

$$\epsilon_{newrel} = \epsilon_1 \left(1 + O\left(\sqrt{\frac{\lambda_1}{\lambda_2}} \kappa^{O(1)}\right)\right) + \sqrt{\lambda_1\lambda_2} \kappa^{O(1)}.$$

By taking the ratio of $\lambda_1$ and $\lambda_2$ sufficiently small, it holds that

$$\epsilon_{newrel} = \epsilon_1(1 + O(1/L)) + \sqrt{\lambda_1\lambda_2} \kappa^{O(1)},$$

so the relative error of a neuron can blow up to at most $\sqrt{\lambda_1\lambda_2} \kappa^{O(1)}$ by the $L$th layer. This is very small if we take $\sqrt{\lambda_1\lambda_2}$ sufficiently small, and so the relative error of the neurons is controlled throughout training.

## D.1 At most two active inputs: assumption and notation

Formally, the following assumption is shared by the main results of this section:

**Assumption D.1** (At most two parent vertices are active). *TRAINNEURON is run with a neuron $v \in V \setminus V_{\text{in}}$, and a parameter initialization $w^0$, such that the following hold:*

- *There are two parent vertices $u_1, u_2 \in P_v$ as well as constants $r_1, r_2, \epsilon_1, \epsilon_2 \in \mathbb{R}$, sets $S_1, S_2 \subset [n]$ and functions $h_1, h_2 : \{-1, 1\}^n \to \mathbb{R}$ such that for each $i \in \{1, 2\}$ and $x \in \{-1, 1\}^n$ the following holds:*

$$f_{u_i}(x; w^0) = r_i \chi_{S_i}(x) + h_i(x) \text{ and } |h_i(x)| \leq |r_i| \epsilon_i,$$

  *and $\hat{h}_i(S_i) = \mathbb{E}_{x \sim \{-1,1\}^n}[h_i(x)\chi_{S_i}(x)] = 0$ for each $i \in \{1, 2\}$.*

- *On the other hand, for any vertex $u' \in P_v \setminus \{u_1, u_2\}$, for all $x \in \{-1, 1\}^n$ we have*

$$f_{u'}(x; w^0) = 0.$$

- *The neuron $v$ is blank at initialization (i.e., all input and output weights and the bias associated with $v$ are zero):*

$$a_e^0 = 0 \text{ for all } e \in E \text{ such that } v \in e, \text{ and } b_v^0 = 0.$$

- *We use $\gamma_i \in \{\lambda_1, \lambda_2\}$ to denote the regularization parameter associated with $a_{(u_i,v)}$. Namely, $\gamma_i = \lambda_1$ if $u_i \in V_{\text{in}}$ and $\gamma_i = \lambda_2$ otherwise.*

- *We use $S = S_1 \cup S_2 \setminus (S_1 \cap S_2)$ to denote the symmetric difference between $S_1$ and $S_2$.*

- *We write the error at parameters $w$ as:*

$$\zeta(x; w) = f(x; w) - g(x),$$

  *and its Fourier coefficients for $S \subset [n]$ is:*

$$\hat{\zeta}(S; w) = \mathbb{E}_{x \sim \{-1,1\}^n}[\zeta(x; w)\chi_S(x)].$$

Under Assumption D.1, we may decompose the function learned during training as follows:

**Claim D.2** (Decomposition of learned function). *Suppose that Assumption D.1 holds, and write and write $e_i = (u_i, v) \in E$ for each $i \in \{1, 2\}$ for shorthand. For any setting of parameters $w_v = \{a_e\}_{e=(u,v)\in E} \cup \{b_v\}$, we have*

$$f(x; [w_{-v}^0, w_v]) = f(x; w^0) + f_v(x; [w_{-v}^0, w_v]) \tag{19}$$

$$= f(x; w^0) + \left(\sum_{i \in [2]} a_{e_i} f_{u_i}(x; w^0)\right)^2 + b_v \tag{20}$$

$$= f(x; w^0) + \left(\sum_{i \in [2]} a_{e_i}(r_i \chi_{S_i}(x) + h_i(x))\right)^2 + b_v. \tag{21}$$

*Proof.* The first line follows from the definition of $f$ in Section 2.1, using that $a_e^0 = 0$ for all $e = (v, u) \in E$. The second line uses that $f_{u'}(x; w^0) = 0$ for all $(u', v) \in E$ such that $u' \notin \{u_1, u_2\}$. The third line uses that $f_{u_i}(x; w^0) = r_i \chi_{S_i}(x) + h_i(x)$. $\qquad\square$

## D.2 Reduction to analyzing the idealized loss

The main technical challenge in Appendix D is to analyze the approximate stationary points of the loss function $\ell_R([w_{-v}^0, w_v])$ with respect to $w_v$. In order to do this, we introduce an "idealized loss

function", which will be a close approximation to the true loss. Let $w = [w^0_{-v}, w_v]$. If $S_1 \neq S_2$, the idealized loss function is defined as:

$$\tilde{\ell}(w) = \frac{1}{2}(2r_1 r_2 a_{e_1} a_{e_2} + \hat{\zeta}(S; w^0))^2 + \frac{1}{2}((r_1 a_{e_1})^2 + (r_2 a_{e_2})^2 + b_v + \hat{\zeta}(\emptyset; w^0))^2$$
$$+ \frac{1}{2} \sum_{\substack{S' \subset [n] \\ S' \neq \emptyset, S}} (\hat{\zeta}(S'; w^0))^2$$

And if $S_1 = S_2$, it is defined as:

$$\tilde{\ell}(w) = \frac{1}{2}((r_1 a_{e_1} + r_2 a_{e_2})^2 + b_v + \hat{\zeta}(\emptyset; w^0))^2 + \frac{1}{2} \sum_{\substack{S' \subset [n] \\ S' \neq \emptyset}} (\hat{\zeta}(S'; w^0))^2.$$

Similarly, define the regularized version:

$$\tilde{\ell}_R(w) = \tilde{\ell}(w) + R(w).$$

As we will see below, $\tilde{\ell}$ is the loss function that would arise if we had $h_1(x) = h_2(x) = 0$ (i.e., if all of the parents to vertex $v$ computed a monomial noiselessly). We prove in Lemma D.3 that $\tilde{\ell}$ is close to the true unregularized loss $\ell$ and that the gradients of $\tilde{\ell}$ with respect to $w_v$ are close to the gradients of $\ell$ with respect to $w_v$. The benefit of this result is that in the proofs we may analyze the stationary points of the simpler loss $\tilde{\ell}$ instead of the actual loss $\ell$.

**Lemma D.3.** *Suppose Assumption D.1 holds on the initialization $w^0$. Then for any parameter vector $w_v = \{a_e\}_{(u,v) \in E} \cup \{b_v\}$, and letting $w = [w^0_{-v}, w_v]$, we have*

$$|\tilde{\ell}(w) - \ell(w)| \lesssim \max_{u \in V} \max_{x \in \{-1,1\}^n} (\|w_v\|_\infty^4 + 1)(|f_u(x; w^0)|^3 + 1)(\max_{i \in \{1,2\}} \epsilon_i), \tag{22}$$

$$\|\nabla_{w_v} \ell(w) - \nabla_{w_v} \tilde{\ell}(w)\|_\infty \lesssim \max_{u \in V} \max_{x \in \{-1,1\}^n} (\|w_v\|_\infty^3 + 1)(|f_u(x; w^0)|^3 + 1)(\max_{i \in \{1,2\}} \epsilon_i). \tag{23}$$

*Proof.* First, for any $x \in \{-1,1\}^n$, define the functions $\tilde{\Upsilon}$ and $\Upsilon$:

$$\tilde{\Upsilon}(x; w) = \left( \sum_{i \in [2]} a_{e_i} r_i \chi_{S_i}(x) \right)^2 + b_v + \zeta(x)$$

$$\Upsilon(x; w) = \left( \sum_{i \in [2]} a_{e_i} (r_i \chi_{S_i}(x) + h_i(x)) \right)^2 + b_v + \zeta(x)$$

The reason for these definitions is that we may write the idealized and actual loss functions in terms of $\tilde{\Upsilon}$ and $\Upsilon$, respectively. First, by Parseval's theorem on the Boolean hypercube, the idealized loss function is:

$$\tilde{\ell}(w) = \frac{1}{2} \mathbb{E}_{x \sim \{-1,1\}^n} \Big[ \big( (2r_1 r_2 a_{e_1} a_{e_2} + \hat{\zeta}(S; w^0)) \chi_S(x) + (r_1^2 (a_{e_1})^2 + b_v + \hat{\zeta}(\emptyset; w^0))$$
$$+ \sum_{\substack{S' \subset [n] \\ S' \neq \emptyset, S}} \hat{\zeta}(S'; w^0) \chi_S(x) \big)^2 \Big]$$

$$= \frac{1}{2} \mathbb{E}_x [\tilde{\Upsilon}(x; w)^2].$$

Furthermore, the actual loss function may be written as:

$$\ell(w) = \mathbb{E}_{x \sim \{-1,1\}^n} [\ell(x; w)]$$

$$= \frac{1}{2} \mathbb{E}_{x \sim \{-1,1\}^n} \left[ \left( \left( \sum_{i \in [2]} a_{e_i} (r_i \chi_{S_i}(x) + h_i(x)) \right)^2 + b_v + \zeta(x) \right)^2 \right] \quad \text{by Eq. (21)}$$

$$= \frac{1}{2} \mathbb{E}_x [\Upsilon(x; w)^2].$$

We bound $\tilde{\ell}(w) - \ell(w)$ by bounding $\tilde{\Upsilon}$ and $\tilde{\Upsilon} - \Upsilon$ pointwise for any $x \in \{-1,1\}^n$. First,

$$
\begin{aligned}
|\tilde{\Upsilon}(x; w)| &\leq (2 \max_i |a_{e_i} r_i|)^2 + |b_v| + |\zeta(x; w^0)| \\
&\leq 4 \max_i |a_{e_i}|^2 (\max_u |f_u(x; w^0)| + \epsilon_i)^2 + |b_v| + |\zeta(x; w^0)| \\
&\lesssim (\|w_v\|_\infty^2 + 1)(\max_u |f_u(x; w^0)| + 1)^2 + |\zeta(x; w^0)| \\
&\leq (\|w_v\|_\infty^2 + 1)(\max_u |f_u(x; w^0)| + 1)^2 + (Ms + WL \max_u |f_u(x; w^0)|) \\
&\lesssim \kappa(\|w_v\|_\infty^2 + 1)(\max_u |f_u(x; w^0)|^2 + 1) := U_1(x).
\end{aligned}
$$

Let us compare $\Upsilon$ to $\tilde{\Upsilon}$:

$$
\begin{aligned}
|\Upsilon(x; w) - \tilde{\Upsilon}(x; w)| &= \left| \left( \sum_{i \in [2]} a_{e_i}(r_i \chi_{S_i}(x) + h_i(x)) \right)^2 - \left( \sum_{i \in [2]} a_{e_i} r_i \chi_{S_i}(x) \right)^2 \right| \\
&\lesssim (\max_i |a_{e_i}|)^2 (|r_1| + |r_2| + |\epsilon_1| + |\epsilon_2|)(|\epsilon_1| + |\epsilon_2|) \\
&\lesssim \|w_v\|_\infty^2 (\max_u |f_u(x; w^0)| + 1)(\max_{i \in \{1,2\}} \epsilon_i) := U_2(x).
\end{aligned}
$$

Of course, by the triangle inequality we also have:

$$
|\Upsilon(x; w)| \leq U_1(x) + U_2(x) := U_3(x).
$$

This lets us prove the first bound in the claim:

$$
\begin{aligned}
|\tilde{\ell}(w) - \ell(w)| &= \left| \mathbb{E}_x[\tilde{\Upsilon}(x; w)^2 - \Upsilon(x; w)^2] \right| \\
&\leq \mathbb{E}_x\left[ \left| \tilde{\Upsilon}(x; w)^2 - \Upsilon(x; w)^2 \right| \right] \\
&\lesssim \mathbb{E}_x\left[ \left| \tilde{\Upsilon}(x; w) + \Upsilon(x; w) \right| \left| \tilde{\Upsilon}(x; w) - \Upsilon(x; w) \right| \right] \\
&\lesssim \mathbb{E}_x[(U_1(x) + U_3(x))(U_2(x))] \\
&\lesssim \max_{u \in V} \max_{x \in \{-1,1\}^n} \kappa(\|w_v\|_\infty^4 + 1)(|f_u(x; w^0)|^3 + 1)(\max_{i \in \{1,2\}} \epsilon_i).
\end{aligned}
$$

For the second part of the claim, we bound the gradient of $\tilde{\ell} - \ell$. For this, let us first bound and compare the gradients of $\tilde{\Upsilon}$ and $\Upsilon$:

$$
\begin{aligned}
\|\nabla_{w_v} \tilde{\Upsilon}(x; w)\|_\infty &\lesssim \max(1, \max_{i \in \{1,2\}} |a_{e_i} r_i| \cdot \max_{i' \in \{1,2\}} |r_i|) \\
&\lesssim (\|w_v\|_\infty + 1) \cdot (\max_{u \in V} |f_u(x; w)|^2 + 1) := U_4(x).
\end{aligned}
$$

Let us compare $\nabla_{w_v} \tilde{\Upsilon}(x; w)$ to $\nabla_{w_v} \Upsilon(x; w)$:

$$
\begin{aligned}
\|\nabla_{w_v} \tilde{\Upsilon}(x; w) - \nabla_{w_v} \Upsilon(x; w)\|_\infty &\lesssim \left( \max_{i \in \{1,2\}} |a_{e_i}| \right) \cdot (|r_1| + |r_2| + |\epsilon_1| + |\epsilon_2|)(|\epsilon_1| + |\epsilon_2|) \\
&\lesssim \|w_v\|_\infty (\max_u |f_u(x; w^0)| + 1)(\max_{i \in \{1,2\}} \epsilon_i) := U_5(x).
\end{aligned}
$$

And by triangle inequality we have:

$$
\|\nabla_{w_v} \Upsilon(x; w)\|_\infty \leq U_4(x) + U_5(x) := U_6(x).
$$

The above bounds may be combined to prove that the gradient of the true loss is close to the gradient of the idealized loss:

$$
\begin{aligned}
\|\nabla_{w_v} \ell(w) - \nabla_{w_v} \tilde{\ell}(w)\|_\infty &= \|\mathbb{E}_x[\Upsilon(x; w)(\nabla_{w_v} \Upsilon(x; w)) - \tilde{\Upsilon}(x; w)(\nabla_{w_v} \tilde{\Upsilon}(x; w))]\|_\infty \\
&= \|\mathbb{E}_x[(\Upsilon(x; w) - \tilde{\Upsilon}(x; w))(\nabla_{w_v} \Upsilon(x; w)) \\
&\qquad - \tilde{\Upsilon}(x; w)(\nabla_{w_v} \tilde{\Upsilon}(x; w) - \nabla_{w_v} \Upsilon(x; w))]\|_\infty \\
&\leq \max_x U_2(x) U_6(x) + U_1(x) U_5(x) \\
&\lesssim \max_{u \in V} \max_{x \in \{-1,1\}^n} (\|w_v\|_\infty^3 + 1)(|f_u(x; w^0)|^3 + 1)(\max_{i \in \{1,2\}} \epsilon_i).
\end{aligned}
$$

$\square$

## D.3 Approximate stationarity of $w^{SGD}$, and loss does not increase

In this subsection, we prove that with large enough minibatch size $B$ and small enough learning rate, with high probability the vector $w^{SGD}$ computed in Line 3 of TRAINNEURON (i) is an approximate stationary point of the idealized loss $\tilde{\ell}([w^0_{-v}, w_v])$ with respect to the parameters $w_v$, and (ii) satisfies $\ell_R(w^{SGD}) \leq \ell_R(w^{perturb})$. This is proved by appealing to the guarantees for NEURONSGD in Lemma C.2 and the fact proved in Lemma D.3 that the idealized loss $\tilde{\ell}$ and the true loss $\ell$ are close. First we prove a helper lemma bounding $\ell_R(w^{perturb})$ and $\max_{u \in V} \max_{x \in \{-1,1\}^n} |f_u(x; w^{perturb})|$.

**Claim D.4.** *Under Assumption D.1, the following bounds are satisfied:*

$$\max_{u \in V} \max_{x \in \{-1,1\}^n} |f_u(x; w^{perturb})| \lesssim \max_{u \in V \setminus \{v\}} |f_u(x; w^0)|^2 + 1, \text{ and}$$

$$\ell_R(w^{perturb}) \lesssim \kappa^2 (\max_u |f_u(x; w^0)|^4 + 1) + \ell_R(w^0).$$

*Proof.* First, note that $w^0_v = 0$ by Assumption C.1. So since the noise added at Line 2 has each entry in $\text{Unif}[-\eta, \eta]$, we must have $\|w^{perturb}_v\|_\infty \leq \eta$. This is the input to the call of NEURONSGD in Line 3 of TRAINNEURON, and because of Lemma C.3 it satisfies

$$\max_{x \in \{-1,1\}^n} |f_v(x; w^{perturb})| \leq \eta + (2\eta \max_{u \in V} \max_{x \in \{-1,1\}^n} |f_u(x; w^0)|)^2$$

$$\lesssim (\max_{u \in V} \max_{x \in \{-1,1\}^n} |f_u(x; w^0)|)^2 + 1$$

Therefore,

$$\max_{u \in V} \max_{x \in \{-1,1\}^n} |f_u(x; w^{perturb})| \leq \max(\max_{u \in V \setminus \{v\}} |f_u(x; w^0)|, |f_v(x; w^{perturb})|)$$

$$\lesssim (\max_{u \in V \setminus \{v\}} |f_u(x; w^0)|)^2 + 1.$$

Furthermore, by splitting the loss into the unregularized part and the regularization terms:

$\ell_R(w^{perturb})$

$$\leq \ell(w^{perturb}) + \frac{\max(\lambda_1, \lambda_2)}{2}(2W\eta^2 + \sum_{e \in E} |a^0_e|^2)$$

$$\leq \max_x \frac{1}{2}(g(x) - WL \max_u f_u(x; w^{perturb}))^2 + \frac{\max(\lambda_1, \lambda_2)}{2}(2W\eta^2 + \sum_{e \in E} |a^0_e|^2) \quad \text{by Claim C.5}$$

$$\lesssim \kappa^2 (\max_u |f_u(x; w^0)|^4 + 1) + \frac{\max(\lambda_1, \lambda_2)}{2}(2W\eta^2 + \sum_{e \in E} |a^0_e|^2)$$

$$\lesssim \kappa^2 (\max_u |f_u(x; w^0)|^4 + 1) + W\eta^2 + \ell_R(w^0)$$

$$\lesssim \kappa^2 (\max_u |f_u(x; w^0)|^4 + 1) + \ell_R(w^0).$$

$\square$

The main result of the subsection may now be stated and proved:

**Lemma D.5.** *Consider running* TRAINNEURON$(v, w^0)$ *(Algorithm 2), where the assumptions Assumption D.1 hold. There is a large enough constant $C'$ such that for any $\delta > 0$, if we define*

$$t_{max} = C'(\kappa^2 (\max_{u \in V} \max_{x \in \{-1,1\}^n} |f_u(x; w^0)|^4 + 1) + \ell_R(w^0))/(\alpha(\epsilon_{stop})^2),$$

*and if the minibatch size is at least*

$$B \geq \max_{u \in V} \max_{x \in \{-1,1\}^n} C'(\lambda_1 \lambda_2)^{-3} \kappa^8 (|f_u(x; w^0)|^8 + 1)(\kappa^8 (|f_u(x; w^0)|^{16} + 1) + \ell_R(w^0)^4) \log(2t_{max}/\delta)/\epsilon^2_{stop},$$

*and if the learning rate is at most*

$$\alpha < \min_{u \in V} \min_{x \in \{-1,1\}^n} 1/(C'(\lambda_1 \lambda_2)^{-5} \kappa^{16} (|f_u(x; w^0)|^{32} + 1)(\kappa^8 (|f_u(x; w^0)|^{16} + 1) + \ell_R(w^0)^4)),$$

*then $\mathbb{P}[E_{stat}] \geq 1 - \delta$, where $E_{stat}$ is the event that the following hold:*

1. *The loss at $w^{SGD}$ is not larger than the loss at $w^{perturb}$:*
$$\ell_R(w^{SGD}) \le \ell_R(w^{perturb})$$

2. *The parameters $w^{SGD}$ are an approximate stationary point with respect to $w_v$:*
$$\|\nabla_{w_v}\ell_R(w^{SGD})\|_\infty \le 2\epsilon_{stop}$$
$$\|\nabla_{w_v}\tilde{\ell}_R(w^{SGD})\|_\infty \le \epsilon_{stat}(w^0, \epsilon_1, \epsilon_2) = \epsilon_{stat},$$
   *where*
$$\epsilon_{stat} = 2\epsilon_{stop} + C' \max_{u \in V} \max_{x \in \{-1,1\}^n} (\lambda_1\lambda_2)^{-3}\kappa^{12}(\ell_R(w^0)^3 + 1)(|f_u(x;w^0)|^{30} + 1)(\max_{i \in \{1,2\}} \epsilon_i)$$

3. *The call to* TRAINNEURON *runs in time $O(\kappa B t_{max})$.*

4. *We have the following bound on the returned parameters:*
$$\|w_v^{SGD}\|_\infty \le U_{stat}(w^0) = U_{stat},$$
   *where $U_{stat} = C'(\lambda_1\lambda_2)^{-1}\kappa^4(\max_u \max_{x \in \{-1,1\}^n} |f_u(x;w^0)|^8 + 1)(\ell_R(w^0) + 1)$.*

*Proof.* The proof is by plugging the bounds of Claim D.4 into Lemma C.2, which provides guarantees for NEURONSGD.

Let $C$ be the constant from Lemma C.2. For large enough constant $C'$, we have
$$t_{max} \ge 3\ell_R(w^{perturb})/(\alpha(\epsilon_{stop})^2),$$

$$B \ge C'(\lambda_1\lambda_2)^{-3}\kappa^8(\max_{u \in V \setminus \{v\}} |f_u(x;w^0)|^8 + 1)(\kappa^8(\max_u |f_u(x;w^0)|^{16} + 1) + \ell_R(w^0)^4)\log(2t_{max}/\delta)/\epsilon_{stop}^2$$

$$\ge C(\lambda_1\lambda_2)^{-3}\kappa^8(\max_{u \in V \setminus \{v\}} |f_u(x;w^0)|^2 + 1)^4(1 + (\kappa^2(\max_u |f_u(x;w^0)|^4 + 1) + \ell_R(w^0))^4)\log(2t_{max}/\delta)/\epsilon_{stop}^2$$

$$\ge C(\lambda_1\lambda_2)^{-3}\kappa^8(\max_u |f_u(x;w^{perturb})|^4)(1 + \ell_R(w^{perturb})^4)\log(2t_{max}/\delta)/\epsilon_{stop}^2,$$

and, similarly,
$$\alpha \le 1/(C(\lambda_1\lambda_2)^{-5}\kappa^{16}(\max_{u \in U} \max_{x \in \{-1,1\}^n} |f_u(x;w^{perturb})|)^{16}(1 + \ell_R(w^{perturb})^4)).$$

In particular, the bounds in NEURONSGD hold with probability at least $1 - \delta$. Let $E_{stat}$ be the event that they hold. Under $E_{stat}$, Item 1 of the lemma immediately follows. Furthermore, since the NEURONSGD method runs for at most $t_{max}$ iterations and each iteration takes at most $B\kappa$ time, Item 3 follows. Finally, since
$$\|\nabla_{w_v}\ell_R(w^{SGD})\| \le 2\epsilon_{stop},$$
we conclude that for some large enough constant $C''$ we have
$$\|\nabla_{w_v}\tilde{\ell}_R(w^{SGD})\|_\infty$$
$$\le 2\epsilon_{stop} + \|\nabla_{w_v}\tilde{\ell}_R(w^{SGD}) - \nabla_{w_v}\ell_R(w^{SGD})\|_\infty$$
$$\le 2\epsilon_{stop} + C'' \max_{u \in V} \max_{x \in \{-1,1\}^n} (\|w_v^{SGD}\|_\infty^3 + 1)(|f_u(x;w^{perturb})|^3 + 1)(\max_{i \in \{1,2\}} \epsilon_i), \quad \text{by Lemma D.3.}$$

This may be further bounded by noting that by Claim C.6 and Claim D.4,
$$\|w_v^{SGD}\|_\infty \lesssim (\lambda_1\lambda_2)^{-1}\kappa^2(\max_u \max_{x \in \{-1,1\}^n} |f_u(x;w^{perturb})|^2 + 1)(\ell_R(w^{SGD}) + 1)$$
$$\lesssim (\lambda_1\lambda_2)^{-1}\kappa^2(\max_u \max_{x \in \{-1,1\}^n} |f_u(x;w^{perturb})|^2 + 1)(\ell_R(w^{perturb}) + 1)$$
$$\lesssim (\lambda_1\lambda_2)^{-1}\kappa^4(\max_u \max_{x \in \{-1,1\}^n} |f_u(x;w^0)|^8 + 1)(\ell_R(w^0) + 1).$$

Thus, for large enough constant $C'''$ and assuming $C'$ is also large enough, by again applying Claim D.4,
$$\|\nabla_{w_v}\tilde{\ell}_R(w^{SGD})\|_\infty$$
$$\le 2\epsilon_{stop} + C' \max_{u \in V} \max_{x \in \{-1,1\}^n} (\lambda_1\lambda_2)^{-3}\kappa^{12}(\ell_R(w^0)^3 + 1)(|f_u(x;w^0)|^{30} + 1)(\max_{i \in \{1,2\}} \epsilon_i).$$

In the above, the second inequality follows from applying Claim D.4. This proves Item 2, concluding the proof of the lemma. $\qquad\square$

In the subsequent proofs of this section, for brevity of notation write $\epsilon_{stat} = \epsilon_{stat}(w^0, \epsilon_1, \epsilon_2)$ and $U_{stat} = U_{stat}(w^0)$.

## D.4 Blank input weights are trained to zero

Before proving the main lemmas in this section, let us prove one last helper claim, which states that the parameters on which $f_v$ does not depend are set to zero by TRAINNEURON.

**Claim D.6** (Blank neuron weights are zero). *Under Assumption D.1, and if the event $E_{stat}$ of Lemma D.5 holds, and if*

$$\tau > 2\epsilon_{stat}/\min(\lambda_1, \lambda_2) := \epsilon^{(1)}, \tag{24}$$

*then for all $e = (u, v) \in E$ such that $f_u(x; w^0) = 0$ for all $x$ (i.e., parents $u$ that are blank at initialization), it holds that $a_e^{\text{round}} = 0$.*

*Proof.* Recall that $w^{SGD} = [w^0_{-v}, w^{SGD}_v]$ (i.e., all parameters except for the parameters to neuron $v$ are frozen during training). For any $e = (u, v) \in E$ such that $u$ is blank, the derivative of the unregularized loss at $x \in \{-1, 1\}^n$ with respect to $a_e$ is:

$$\frac{\partial \ell(x; w^{SGD})}{\partial a_e} = \frac{\partial}{\partial a_e}(\frac{1}{2}(f(x; w^0) + f_v(x; w^{SGD}) - g(x))^2), \qquad \text{by Eq. (19)}$$

$$= (f(x; w^0) + f_v(x; w^{SGD}) - g(x)) \cdot \frac{\partial}{\partial a_e} f_v(x; w^{SGD})$$

$$= (f(x; w^0) + f_v(x; w^{SGD}) - g(x)) \cdot 0 = 0 \qquad \text{since } f_u(x; w^t) = 0$$

Therefore

$$|\frac{\partial \ell_R(w^t)}{\partial a_e}| \geq \min(\lambda_1, \lambda_2)|a_e^t| - |\frac{\partial \ell(w^t)}{\partial a_e}| = \min(\lambda_1, \lambda_2)|a_e^t|.$$

So in particular

$$|a_e^t| \leq 2\epsilon_{stop}/\min(\lambda_1, \lambda_2) \leq \epsilon_{stat}/\min(\lambda_1, \lambda_2) < \tau,$$

so by the truncation step of Line 4, the algorithm returns trained weights $w^{\text{round}}$ with $a_e^{\text{round}} = 0$. $\square$

## D.5 TRAINNEURON correctness : training a neuron with at most one active input (Lemma D.7)

We may now state and prove the first main result of this section – i.e., if a neuron with at most one active input is trained, then it remains blank after training.

**Lemma D.7** (TRAINNEURON correctness: at most one active input). *Suppose that Assumption D.1 holds, the event $E_{stat}$ from Lemma D.5 holds, and also $r_2 = \epsilon_2 = 0$ (i.e., neuron $u_2$ is blank). Suppose also that*

$$\tau > 2\epsilon_{stat}/\min(\lambda_1, \lambda_2) := \epsilon^{(1)}, \tag{25}$$

$$\tau > |\hat{\zeta}(\emptyset; w^0)| + \epsilon_{stat} + (r_1)^2|2\epsilon_{stat}/\min(\lambda_1, \lambda_2)|^2, \tag{26}$$

*and*

$$\epsilon_{stat} < \min(\lambda_1, \lambda_2)/(2r_1)^2. \tag{27}$$

*Then after running TRAINNEURON$(v, w^0)$, we have $w^{\text{round}} = w^0$, so the weights do not change during training and the neuron $v$ remains blank.*

*Proof.* All neurons $u \in P_v \setminus \{u_1\}$ are blank at the initialization $w^0$ by the assumptions in the lemma statement (for the case of $u = u_2$, this follows because because $r_2 = \epsilon_2 = 0$, so $f_{u_2}(x; w^0) = 0$ for all $x \in \{-1, 1\}^n$). Therefore, by Claim D.6 and Eq. (25), for edge $e = (u, v)$ the algorithm TRAINNEURON returns weight $a_e^{\text{round}} = 0$.

Now consider the parameters $b_v$ and $a_{e_1}$. We compute the partial derivatives of the idealized loss:

$$\frac{\partial \tilde{\ell}}{\partial b_v} = ((r_1 a_{e_1})^2 + b_v - \hat{\zeta}(\emptyset; w^0)) \text{ and } \frac{\partial \tilde{\ell}}{\partial a_{e_1}} = ((r_1 a_{e_1})^2 + b_v - \hat{\zeta}(\emptyset; w^0))(2(r_1)^2 a_{e_1}).$$

Since the event $E_{stat}$ holds, by Item 2 of Lemma D.5, we have

$$\|\nabla_{w_v}\tilde{\ell}_R(w^{SGD})\|_\infty \le \epsilon_{stat},$$

which implies

$$|(r_1 a_{e_1}^{SGD})^2 + b_v^{SGD} - \hat{\zeta}(\emptyset; w^0)| \le \epsilon_{stat},$$

and

$$(\gamma_1 - 2(r_1)^2|(r_1 a_{e_1}^{SGD})^2 + b_v^{SGD} - \hat{\zeta}(\emptyset; w^0)|)|a_{e_1}^{SGD}| \le \epsilon_{stat},$$

which means that

$$(\gamma_1 - 2(r_1)^2\epsilon_{stat})|a_{e_1}^{SGD}| \le \epsilon_{stat},$$

so, by Eq. (27),

$$|a_{e_1}^{SGD}| \le \epsilon_{stat}/(\gamma_1/2) = 2\epsilon_{stat}/\gamma_1,$$

and hence

$$|b_v^{SGD}| \le |\hat{\zeta}(\emptyset; w^0)| + \epsilon_{stat} + (r_1)^2|2\epsilon_{stat}/\gamma_1|^2.$$

Thus, in Line 4 of TRAINNEURON since $\tau > \max(2\epsilon_{stat}/\gamma_1, |\hat{\zeta}(\emptyset; w^0)| + \epsilon_{stat} + (r_1)^2|2\epsilon_{stat}/\gamma_1|^2)$ by Eq. (25) and Eq. (26), we have $b_v^{round} = 0$ and $a_{e_1}^{round} = 0$. So overall we have $w_v^{round} = \vec{0}$ for all parameters, so $w^{round} = [w_{-v}^0, w_v^{round}] = [w_{-v}^0, \vec{0}] = w^0$, and the neuron remains blank. $\square$

## D.6 TRAINNEURON correctness: training a neuron with two active inputs whose product is not useful (Lemma D.8)

The next main result of this section is the correctness of TRAINNEURON in the case in which both $u_1$ and $u_2$ are active neurons, but the monomial $\chi_S(x)$ that is approximately computed by their product only has low correlation with the error function $\zeta(x; w^0)$. In this case learning the product of the active inputs would not significantly decrease the loss, and the $L_2$ regularization on the weights dominates. Thus the neuron remains blank after training because of the rounding step in Line 4 of TRAINNEURON.

**Lemma D.8** (TRAINNEURON correctness: two active inputs, product not useful). *Define*

$$\epsilon^{(1)} := 2\epsilon_{stat}/\min(\lambda_1, \lambda_2)$$

$$\epsilon^{(2)} = (1 + \max_{i \in \{1,2\}} r_i^2 U_{stat})\epsilon_{stat}$$

$$\epsilon^{(3)} := 8\epsilon^{(2)}(1 + (U_{stat})^2 + |\hat{\zeta}(S; w^0)|)\max(1, |r_1 r_2|^2)/\min(\lambda_1, \lambda_2)^2$$

$$\epsilon^{(4)} := 2(\sqrt{\frac{\max(\lambda_1, \lambda_2)}{\min(\lambda_1, \lambda_2)}}\sqrt{\max(\lambda_1, \lambda_2) + |\hat{\zeta}(S; w^0)|} + \epsilon^{(2)}\max(\lambda_1, \lambda_2))/\min(1, |r_1 r_2|^2)$$

*Suppose that Assumption D.1 holds and that event $E_{stat}$ from Lemma D.5 holds. Suppose also that*

$$\tau > \max(\epsilon^{(1)}, \epsilon^{(3)}, \epsilon^{(4)}) \tag{28}$$

$$\tau > |\hat{\zeta}(\emptyset; w^0)| + 2(r_1^2 + r_2^2)(\max(\epsilon^{(1)}, \epsilon^{(3)}, \epsilon^{(4)}))^2 + \epsilon_{stat} \tag{29}$$

$$4(r_1^2 + r_2^2)\epsilon_{stat}/\min(\lambda_1, \lambda_2) \le 1/2 \tag{30}$$

*Then after running TRAINNEURON$(v, w^0)$, the weights are not changed during training (i.e., $w^{round} = w^0$) and so the neuron $v$ remains blank.*

Before proving this lemma, let us prove a helper claim:

**Claim D.9.** *Suppose that Assumption D.1 and the event $E_{stat}$ from Lemma D.5 both hold, and also that $S_1 \ne S_2$. Define*

$$\rho = (2r_1 r_2 a_{e_1}^{SGD} a_{e_2}^{SGD} + \hat{\zeta}(S; w^0))(2r_1 r_2). \tag{31}$$

*Then, for any distinct $i, j \in \{1, 2\}$*

$$|\rho a_{e_j}^{SGD} + \gamma_i a_{e_i}^{SGD}| \le (1 + \max_{i \in \{1,2\}} r_i^2 U_{stat})\epsilon_{stat} := \epsilon^{(2)}, \tag{32}$$

*and*

$$|\gamma_1 \gamma_2 - \rho^2||a_{e_i}^{SGD}| \le (\gamma_j + |\rho|)\epsilon^{(2)}. \tag{33}$$

*Proof.* First, write the derivatives of the idealized loss with respect to the parameters $b_v$, $a_{e_1}$, and $a_{e_2}$:

$$\frac{\partial \tilde{\ell}_R}{\partial b_v} = \left| (r_1 a_{e_1})^2 + (r_2 a_{e_2})^2 + b_v + \hat{\zeta}(\emptyset; w^0) \right|.$$

Further, for any distinct $i, j \in \{1, 2\}$,

$$\frac{\partial \tilde{\ell}_R}{\partial a_{e_i}} = (2 r_1 r_2 a_{e_1} a_{e_2} + \hat{\zeta}(S; w^0))(2 r_1 r_2 a_{e_j}) + \left( \frac{\partial \tilde{\ell}}{\partial b_v} \right)(2 r_i^2 a_{e_i}) + \gamma_i a_{e_i}.$$

By the guarantee in Item 2 of Lemma D.5 and the event $E_{stat}$, we have $\|\nabla_{w_v} \tilde{\ell}_R(w^{SGD})\|_\infty \leq \epsilon_{stat}$. It follows that

$$|(2 r_1 r_2 a_{e_1}^{SGD} a_{e_2}^{SGD} + \hat{\zeta}(S; w^0))(2 r_1 r_2 a_{e_j}^{SGD}) + \gamma_i a_{e_i}^{SGD}| \leq \epsilon_{stat}(1 + r_i^2 a_{e_i}^{SGD}).$$

Finally, by Item 4 of Lemma D.5 we also have the bound $\|w_v^{SGD}\|_\infty \leq U_{stat}$, which combined with the above equation implies

$$|(2 r_1 r_2 a_{e_1}^{SGD} a_{e_2}^{SGD} + \hat{\zeta}(S; w^0))(2 r_1 r_2 a_{e_j}^{SGD}) + \gamma_i a_{e_i}^{SGD}| \leq \epsilon_{stat}(1 + r_i^2 U_{stat}),$$

which is the claimed inequality Eq. (32) when rewritten in terms of $\rho$.

Multiplying Eq. (32) for $i = 1, j = 2$ by $\gamma_2$:

$$|\gamma_2 \rho a_{e_2}^{SGD} + \gamma_1 \gamma_2 a_{e_1}^{SGD}| \leq \gamma_2 \epsilon^{(2)},$$

and multiplying Eq. (32) for $i = 2, j = 1$ by $|\rho|$:

$$|\rho^2 a_{e_1}^{SGD} + \gamma_2 \rho a_{e_2}^{SGD}| \leq |\rho| \epsilon^{(2)}.$$

Combining the above two inequalities by the triangle inequality,

$$|\gamma_1 \gamma_2 - \rho^2||a_{e_1}^{SGD}| \leq (\gamma_2 + |\rho|) \epsilon^{(2)}.$$

Eq. (33) follows by a symmetric argument. $\square$

Now we may prove the main result of this subsection:

*Proof of Lemma D.8.* We claim that

$$|a_{e_1}^{SGD}|, |a_{e_2}^{SGD}| \leq \max(\epsilon^{(1)}, \epsilon^{(3)}, \epsilon^{(4)}). \tag{34}$$

This is proved below, but first let us see the consequences. Plugging Eq. (34) into the stationarity condition $\left| \frac{\partial \tilde{\ell}}{\partial b_v} \big|_{w=w^{SGD}} \right| \leq \epsilon_{stat}$ guaranteed by Item 2 of Lemma D.5, we obtain

$$|b_v^{SGD}| \leq |\hat{\zeta}(\emptyset; w^0)| + 2(r_1 a_{e_1}^{SGD})^2 + 2(r_2 a_{e_2}^{SGD})^2 + \epsilon_{stat}$$
$$\leq |\hat{\zeta}(\emptyset; w^0)| + 2(|r_1|^2 + |r_2|^2)(\max(\epsilon^{(1)}, \epsilon^{(3)}, \epsilon^{(4)}))^2 + \epsilon_{stat}.$$

Therefore $|a_{e_1}^{SGD}|, |a_{e_2}^{SGD}| < \tau$ by Eq. (28) and $|b_v^{SGD}| < \tau$ by Eq. (29). So Line 4 of TRAINNEURON rounds $a_{e_1}^{SGD}, a_{e_2}^{SGD}$ and $b_v^{SGD}$ to $a_{e_1}^{round} = a_{e_2}^{round} = b_v^{round} = 0$. Furthermore, Claim D.6 and Eq. (28) imply $a_{e'}^{round} = 0$ for all $e' = (u', v) \in E$ such that $u' \notin \{u_1, u_2\}$. Overall, this implies $w^{round} = w^0$, since $w_v^{round} = \vec{0} = w_v^0$.

Therefore, it only remains to show (34). We prove it with a case analysis.

**Case 1**: If $S_1 = S_2$, we have $S = S_1 \cup S_2 \setminus (S_1 \cap S_2) = \emptyset$. In this case, Item 2 of Lemma D.5 guarantees the stationarity conditions $\left| \frac{\partial \tilde{\ell}_R}{\partial b_v} \big|_{w=w^{SGD}} \right| \leq \epsilon_{stat}$ and $\left| \frac{\partial \tilde{\ell}_R}{\partial a_{e_i}} \big|_{w=w^{SGD}} \right| \leq \epsilon_{stat}$ for any $i \in \{1, 2\}$, i.e.,

$$|(r_1 a_{e_1}^{SGD} + r_2 a_{e_2}^{SGD})^2 + b_v^{SGD} + \hat{\zeta}(S; w^0)| \leq \epsilon_{stat}.$$

$$\left| 2 r_i (r_1 a_{e_1}^{SGD} + r_2 a_{e_2}^{SGD})((r_1 a_{e_1}^{SGD} + r_2 a_{e_2}^{SGD})^2 + b_v^{SGD} + \hat{\zeta}(S; w^0)) + \gamma_i a_{e_i}^{SGD} \right| \leq \epsilon_{stat}.$$

Combining these two inequalities and the triangle inequality, we obtain
$$\left|\gamma_i a_{e_i}^{SGD}\right| \leq (1 + |2r_i(r_1 a_{e_1}^{SGD} + r_2 a_{e_2}^{SGD})|)\epsilon_{stat},$$

So
$$\max_i |a_{e_i}^{SGD}| \leq (1 + 4(r_1^2 + r_2^2)\max_i |a_{e_i}^{SGD}|)\epsilon_{stat}/\min(\lambda_1, \lambda_2)$$

$$\leq \epsilon_{stat}/\min(\lambda_1, \lambda_2) + \frac{1}{2}\max_i |a_{e_i}^{SGD}|. \qquad\qquad \text{by Eq. (30)}$$

This means that
$$\max_i |a_{e_i}^{SGD}| \leq \epsilon_{stat}/\min(\lambda_1, \lambda_2) = \epsilon^{(1)},$$

concluding the analysis of this case.

**Case 2**: Otherwise, we are in the case that $S_1 \neq S_2$. Let $\rho = (2r_1 r_2 a_{e_1}^{SGD} a_{e_2}^{SGD} + \hat{\zeta}(S; w^0))(2r_1 r_2)$ be defined as in Eq. (31).

**Case 2a**: If $|\gamma_1\gamma_2 - \rho^2| \geq \gamma_1\gamma_2/2$, then by Eq. (33), which is guaranteed by Claim D.9,

$\max\limits_{i \in [2]} |a_{e_i}^{SGD}|$

$\leq (\max\limits_{j \in [2]} \gamma_j + |\rho|)\epsilon^{(2)}/|\gamma_1\gamma_2 - \rho^2|$

$\leq (\max\limits_{j \in [2]} \gamma_j + |\rho|)\epsilon^{(2)}/(\gamma_1\gamma_2/2)$

$\leq (\max(\lambda_1, \lambda_2) + |\rho|)\epsilon^{(2)}/(\gamma_1\gamma_2/2)$

$\leq (\max(\lambda_1, \lambda_2) + |(2r_1 r_2 a_{e_1}^{SGD} a_{e_2}^{SGD} + \hat{\zeta}(S; w^0))(2r_1 r_2)|)\epsilon^{(2)}/(\gamma_1\gamma_2/2)$

$\leq (1 + |2r_1 r_2|^2|a_{e_1}^{SGD} a_{e_2}^{SGD}| + |2r_1 r_2||\hat{\zeta}(S; w^0)|)\epsilon^{(2)}/(\gamma_1\gamma_2/2)$

$\leq 8\epsilon^{(2)}(1 + |a_{e_1}^{SGD} a_{e_2}^{SGD}| + |\hat{\zeta}(S; w^0)|)\max(1, |r_1 r_2|^2)/(\gamma_1\gamma_2)$

$\leq 8\epsilon'(1 + (U_{stat})^2 + |\hat{\zeta}(S; w^0)|)\max(1, |r_1 r_2|^2)/(\gamma_1\gamma_2) \qquad\qquad \text{by Item 4 of Lemma D.5}$

$\leq \epsilon^{(3)}.$

**Case 2b**: Otherwise, if $|\gamma_1\gamma_2 - \rho^2| \leq \gamma_1\gamma_2/2$, then
$$|\rho| \in [\sqrt{\gamma_1\gamma_2/2}, \sqrt{3\gamma_1\gamma_2/2}]. \qquad\qquad (35)$$

In this case,
$$|2r_1 r_2 a_{e_1}^{SGD} a_{e_2}^{SGD}| \leq |\rho/(2r_1 r_2)| + |\hat{\zeta}(S; w^0)|$$
$$\leq \sqrt{3\gamma_1\gamma_2/2}/|2r_1 r_2| + |\hat{\zeta}(S; w^0)| \qquad\qquad \text{by Eq. (35)}$$
$$\leq \sqrt{\gamma_1\gamma_2}/|r_1 r_2| + |\hat{\zeta}(S; w^0)|.$$

Therefore, $\min_i |a_{e_i}^{SGD}| \leq \sqrt{\sqrt{\gamma_1\gamma_2}/(2|r_1 r_2|^2) + |\hat{\zeta}(S; w^0)|/|2r_1 r_2|}$ Also, by Eq. (32) of Claim D.9, for any distinct $i, j \in \{1, 2\}$ we have
$$|a_{e_j}^{SGD} + \gamma_i a_{e_i}^{SGD}/\rho| \leq \epsilon^{(2)}/|\rho|,$$

Therefore, by the triangle inequality:
$$\max_i |a_{e_i}^{SGD}| \leq (\max(\gamma_1, \gamma_2)\min_i |a_{e_i}^{SGD}| + \epsilon^{(2)})/|\rho|$$
$$\leq (\max(\gamma_1, \gamma_2)\min_i |a_{e_i}^{SGD}| + \epsilon^{(2)})/\sqrt{\gamma_1\gamma_2/2} \qquad\qquad \text{by Eq. (35).}$$

Therefore
$$\max_i |a_{e_i}^{SGD}| \leq (\max(\gamma_1, \gamma_2)\sqrt{\sqrt{\gamma_1\gamma_2}/(2|r_1 r_2|^2) + |\hat{\zeta}(S; w^0)|/|2r_1 r_2|} + \epsilon^{(2)})/\sqrt{\gamma_1\gamma_2/2}$$

$$\leq 2(\sqrt{\frac{\max(\gamma_1, \gamma_2)}{\min(\gamma_1, \gamma_2)}}\sqrt{\sqrt{\gamma_1\gamma_2} + |\hat{\zeta}(S; w^0)|} + \epsilon^{(2)}\sqrt{\gamma_1\gamma_2})/\min(1, |r_1 r_2|^2)$$

$$\leq \epsilon^{(4)}.$$

$\square$

### D.7 TRAINNEURON correctness: training a neuron with two active inputs whose product is useful (Lemmas D.11 to D.13)

We now prove Lemmas D.11 to D.13, which are our final main results on TRAINNEURON's correctness. These results state that if a neuron with two active inputs is trained, and if learning the product of the inputs would significantly contribute to reducing the loss, then with polynomially lower bounded probability the neuron learns the product up to some small relative error, and remains blank otherwise.

For the following definition recall that $\rho = (2r_1 r_2 a_{e_1}^{SGD} a_{e_2}^{SGD} + \hat{\zeta}(S; w^0))(2r_1 r_2)$ as defined in Eq. (31).

**Definition D.10.** *Let $E_{newactive}$ be the event that $|\gamma_1 \gamma_2 - \rho^2| < \gamma_1 \gamma_2/2$.*

In our analysis, when $|\hat{\zeta}(S; w^0)|$ is sufficiently large (i.e., when the learning a neuron that represents $\chi_S$ would significantly reduce the loss, then the event $E_{newactive}$ corresponds to when TRAINNEURON creates an active neuron.

**Lemma D.11** (Two active inputs, product is useful, case when neuron remains blank). *Suppose that Assumption D.1 holds, and the event $E_{stat} \cap (\neg E_{newactive})$ holds, and $S_1 \neq S_2$. Finally, recall the definitions of $\epsilon^{(1)}, \epsilon^{(2)}, \epsilon^{(3)}$*

$$\epsilon^{(1)} = 2\epsilon_{stat} / \min(\lambda_1, \lambda_2)$$

$$\epsilon^{(2)} = \epsilon_{stat}(1 + \max_{i \in \{1,2\}} r_i^2 U_{stat})$$

$$\epsilon^{(3)} = 8\epsilon^{(2)}(1 + (U_{stat})^2 + |\hat{\zeta}(S; w^0)|) \max(1, |r_1 r_2|^2) / \min(\lambda_1, \lambda_2)^2$$

*and suppose that the following hold:*

$$\tau > \max(\epsilon^{(1)}, \epsilon^{(3)}) \tag{36}$$

$$\tau > |\hat{\zeta}(\emptyset; w^0)| + 2(|r_1|^2 + |r_2|^2)(\epsilon^{(3)})^2 + \epsilon_{stat} \tag{37}$$

*Then $w^{\mathrm{round}} = w^0$ (and $v$ remains a blank neuron).*

*Proof.* Since $S_1 \neq S_2$, and the event $\neg E_{newactive}$ implies $|\gamma_1 \gamma_2 - \rho^2| \geq \gamma_1 \gamma_2/2$, the proof of this lemma is identical to the proof for Case 2a in Lemma D.8. □

**Lemma D.12** (Two active inputs, product is useful, case when new active neuron is created). *Suppose that Assumption D.1 holds, and the event $E_{stat} \cap E_{newactive}$ holds. Suppose also that $f_{u_1}(x; w^0)$ depends only on variables in $S_1$, and $f_{u_2}(x; w^0)$ depends only on variables in $S_2$, and that $S_2 \neq \emptyset$ and $S_1 \cap S_2 = \emptyset$. Suppose also that $\epsilon_1 \leq 1$ and $\epsilon_2 = 0$. Finally, recall the definition*

$$\epsilon^{(1)} = 2\epsilon_{stat} / \min(\lambda_1, \lambda_2),$$

*and suppose also that*

$$\tau > \epsilon^{(1)} \tag{38}$$

$$|\hat{\zeta}(S; w^0)| \geq \sqrt{3} \max(\lambda_1, \lambda_2) / |r_1 r_2|. \tag{39}$$

$$\epsilon^{(2)} \leq \min(\lambda_1, \lambda_2) \sqrt{|\hat{\zeta}(S; w^0)| / |r_1 r_2|} / 8 \tag{40}$$

$$\tau < \frac{1}{8} \sqrt{\frac{\lambda_1}{\lambda_2} |\hat{\zeta}(S; w^0)| / |r_1 r_2|}, \tag{41}$$

$$\tau < (|\hat{\zeta}(S; w^0)|/4)(\min_i |r_i|)/(\max_i |r_i|) - |\hat{\zeta}(\emptyset; w^0)| - \epsilon_{stat} \tag{42}$$

*for some large enough universal constant $C > 0$.*

*Then*

1. We may write $f_v(x; w^{\text{round}}) = r\chi_S(x) + h(x)$, such that $\hat{h}(S) = 0$, the error is bounded by $|h(x)| \le |r|\epsilon_{newrel}$, where

$$\epsilon_{newrel} = (4\epsilon_{stop} + 2|\hat{\zeta}(\emptyset; w^0)|)/|\hat{\zeta}(S; w^0)| + 32\frac{\lambda_2}{\lambda_1}|\epsilon_1|^2|r_1/r_2| + \epsilon_1(8\frac{|r_1|}{|r_2|}\sqrt{\frac{\gamma_2}{\gamma_1}} + 1)$$

and the scaling factor $r$ is close to $-\hat{\zeta}(S; w^0)$:

$$|r + \hat{\zeta}(S; w^0)| \le \frac{4\sqrt{\gamma_1\gamma_2}}{|r_1 r_2|}.$$

2. The weights after training are bounded: $a_{e'}^{\text{round}} = 0$ for all $e' = (u', v) \in E$ such that $u' \notin \{u_1, u_2\}$, and

$$|a_{e_1}^{\text{round}}|, |a_{e_2}^{\text{round}}| \le 4\sqrt{\frac{\lambda_2}{\lambda_1}|\hat{\zeta}(S; w^0)|/|r_1 r_2|}.$$

3. The error bias is bounded:
$$|\hat{\zeta}(\emptyset; w^{\text{round}})| \le \epsilon_{stat}.$$

**Lemma D.13** (Two active inputs, product is useful: two cases and probability lower bound). *Suppose that Assumption D.1 holds, as well as the conditions of Lemma D.11. Suppose also that $r_1, r_2 \ne 0$, $S_1 \ne S_2$, and that the following inequalities hold, where $C > 0$ is a large enough universal constant,*

$$\eta \ge 4\tau \tag{43}$$

$$|2r_1 r_2 \eta^2| < |\hat{\zeta}(S; w^0)|/16 \tag{44}$$

$$|2r_1 r_2 \tau^2| < |\hat{\zeta}(S; w^0)|/16 \tag{45}$$

$$|r_1 r_2 \eta^2 \hat{\zeta}(S; w^0)| \ge 16(r_1^2 \eta^2 + r_2^2 \eta^2 + |\hat{\zeta}(\emptyset; w^0)|)^2 \tag{46}$$

$$\frac{1}{32}|r_1 r_2 \eta^2 \hat{\zeta}(S; w^0)| - \lambda_2 W\eta^2 - C\max_{u \in V}\max_{x \in \{-1,1\}^n}(|f_u(x; w^0)|^3 + 1)(\max_{i \in \{1,2\}} \epsilon_i) > 0 \tag{47}$$

*Then*
$$\mathbb{P}[E_{newactive} \cap E_{stat} \mid w^0] \ge \min(1, \sqrt{|r_1 r_2 \hat{\zeta}(S; w^0)|}/8) - \mathbb{P}[\neg E_{stat} \mid w^0].$$

### D.7.1 Proof of Lemma D.12

*Proof of Lemma D.12.* The proof is modularized into several claims:

**Claim D.14** (Input weights from blank neurons are sent to zero). $a_{e'}^{\text{round}} = 0$ *for all $e' = (u', v) \in E$ such that $u' \notin \{u_1, u_2\}$.*

*Proof.* By Claim D.6, since the precondition Eq. (38) holds, as well as Assumption D.1 and the event $E_{stat}$. □

So it only remains to examine $a_{e_1}^{\text{round}}$, $a_{e_2}^{\text{round}}$, and $b_v^{\text{round}}$.

**Claim D.15.** *The following bounds on $|a_{e_i}^{SGD}|$ hold:*

$$\max_{i \in \{1,2\}} |a_{e_i}^{SGD}| \ge \sqrt{|\hat{\zeta}(S; w^0)|/|4r_1 r_2|}. \tag{48}$$

$$\min_{i \in \{1,2\}} |a_{e_i}^{SGD}| \le \sqrt{|\hat{\zeta}(S; w^0)|/|r_1 r_2|} \tag{49}$$

*Proof.* Since $E_{newactive}$ holds we must have $\rho^2 \in [\gamma_1\gamma_2/2, 3\gamma_1\gamma_2/2]$, so

$$|\rho| \in [\sqrt{\gamma_1\gamma_2/2}, \sqrt{3\gamma_1\gamma_2/2}]. \tag{50}$$

Plugging Eq. (50) into the definition of $\rho$ implies that

$$||2r_1r_2 a_{e_1}^{SGD} a_{e_2}^{SGD}| - |\hat{\zeta}(S; w^0)|| \leq \sqrt{3\gamma_1\gamma_2/2}/|2r_1r_2|.$$

Since $|\hat{\zeta}(S; w^0)| \geq \sqrt{3}\max(\lambda_1, \lambda_2)/|r_1r_2| > 2\sqrt{3\gamma_1\gamma_2/2}/|2r_1r_2|$ by Eq. (39), this means

$$|2r_1r_2 a_{e_1}^{SGD} a_{e_2}^{SGD}| \in [|\hat{\zeta}(S; w^0)|/2, 2|\hat{\zeta}(S; w^0)|]. \tag{51}$$

Eqs. (48) and (49) immediately follow. $\qquad\square$

**Claim D.16.** *For any distinct $i, j \in \{1, 2\}$ we have*

$$\frac{1}{4}|a_{e_i}^{SGD}| \leq \sqrt{\frac{\gamma_j}{\gamma_i}}|a_{e_j}^{SGD}| \leq 4|a_{e_i}^{SGD}|. \tag{52}$$

*Proof.* Moreover, plugging Eq. (50) into Eq. (32), for all distinct $i, j \in \{1, 2\}$, we also have

$$
\begin{aligned}
|a_{e_j}^{SGD} + \gamma_i a_{e_i}^{SGD}/\rho| &\leq \epsilon^{(2)}/|\rho| \\
&\leq \epsilon^{(2)}/\sqrt{\gamma_1\gamma_2/2} \\
&\leq \sqrt{|\hat{\zeta}(S; w^0)|/|4r_1r_2|}/2 && \text{by Eq. (40)} \\
&\leq \max_{i^* \in \{1,2\}} |a_{e_{i^*}}^{SGD}|/2 && \text{by Eq. (48)} \tag{53}
\end{aligned}
$$

Let $i^*, j^* \in \{1, 2\}$ be distinct indices such that $|a_{e_{i^*}}^{SGD}| = \max_{i \in \{1,2\}} |a_{e_i}^{SGD}|$ and $|a_{e_{j^*}}^{SGD}| = \min_{j \in \{1,2\}} |a_{e_j}^{SGD}|$. Therefore,

$$|a_{e_{i^*}}^{SGD} + \gamma_{j^*} a_{e_{j^*}}^{SGD}/\rho| \leq |a_{e_{i^*}}^{SGD}|/2 \qquad \text{by Eq. (53)}$$

As a consequence,

$$\frac{1}{2}|a_{e_{i^*}}^{SGD}| \leq |\gamma_{j^*} a_{e_{j^*}}^{SGD}/\rho| \leq \frac{3}{2}|a_{e_{i^*}}^{SGD}|.$$

And because of the bounds in Eq. (50), we have

$$\frac{1}{4}|a_{e_{i^*}}^{SGD}| \leq \sqrt{\frac{\gamma_{j^*}}{\gamma_{i^*}}}|a_{e_{j^*}}^{SGD}| \leq 4|a_{e_{i^*}}^{SGD}|.$$

This immediately implies Eq. (52). $\qquad\square$

We now use the above claims to bound the range of $[\min_i |a_{e_i}^{SGD}|, \max_i |a_{e_i}^{SGD}|]$:

**Claim D.17.**

$$\min_i |a_{e_i}^{SGD}| \geq \frac{1}{8}\sqrt{\frac{\lambda_1}{\lambda_2}|\hat{\zeta}(S; w^0)|/|r_1r_2|} \tag{54}$$

$$\max_i |a_{e_i}^{SGD}| \leq 4\sqrt{\frac{\lambda_1}{\lambda_2}|\hat{\zeta}(S; w^0)|/|r_1r_2|} \tag{55}$$

*Proof.* We first show Eq. (54):

$$
\begin{aligned}
\min_i |a_{e_i}^{SGD}| &\geq \frac{1}{4}\sqrt{\frac{\min(\gamma_1, \gamma_2)}{\max(\gamma_1, \gamma_2)}}\max_i |a_{e_i}^{SGD}| && \text{by Eq. (52)} \\
&\geq \frac{1}{4}\sqrt{\frac{\min(\gamma_1, \gamma_2)}{\max(\gamma_1, \gamma_2)}|\hat{\zeta}(S; w^0)|/|4r_1r_2|} && \text{by Eq. (48)} \\
&\geq \frac{1}{8}\sqrt{\frac{\lambda_1}{\lambda_2}|\hat{\zeta}(S; w^0)|/|r_1r_2|} && \text{since } \lambda_1 \leq \lambda_2 \text{ and } \gamma_1, \gamma_2 \in \{\lambda_1, \lambda_2\}
\end{aligned}
$$

And similarly we show Eq. (55):

$$\max_i |a_{e_i}^{SGD}| \leq 4\sqrt{\frac{\max(\gamma_1, \gamma_2)}{\min(\gamma_1, \gamma_2)}} \min_i |a_{e_i}^{SGD}| \qquad \text{by Eq. (55)}$$

$$\leq 4\sqrt{\frac{\max(\gamma_1, \gamma_2)}{\min(\gamma_1, \gamma_2)}} \sqrt{|\hat{\zeta}(S; w^0)|/|r_1 r_2|}$$

$$\leq 4\sqrt{\frac{\lambda_2}{\lambda_1}|\hat{\zeta}(S; w^0)|/|r_1 r_2|}.$$

$\square$

**Claim D.18.** $a_{e_1}^{\text{round}} = a_{e_1}^{SGD}$, $a_{e_2}^{\text{round}} = a_{e_2}^{SGD}$ and $b_v^{\text{round}} = b_v^{SGD}$.

*Proof.* First, from the previous claim,

$$\min_i |a_{e_i}^{SGD}| \geq \frac{1}{8}\sqrt{\frac{\lambda_1}{\lambda_2}|\hat{\zeta}(S; w^0)|/|r_1 r_2|} \qquad \text{by Eq. (54)}$$

$$> \tau \qquad \text{by Eq. (41).}$$

Furthermore, by the stationarity condition $\left|\frac{\partial \tilde{\ell}_R}{\partial b_v}\big|_{w=w^{SGD}}\right| \leq \epsilon_{stat}$, which is guaranteed by the event $E_{stat}$ and Item 2 from Lemma D.5:

$$|(r_1 a_{e_1}^{SGD})^2 + (r_2 a_{e_2}^{SGD})^2 + b_v^{SGD} + \hat{\zeta}(\emptyset; w^0)| \leq \epsilon_{stat}.$$

which means that

$$|b_v^{SGD}| \geq (r_1 a_{e_1}^{SGD})^2 + (r_2 a_{e_2}^{SGD})^2 - |\hat{\zeta}(\emptyset; w^0)| - \epsilon_{stat}$$

$$\geq \max_i (r_i a_{e_i}^{SGD})^2 - |\hat{\zeta}(\emptyset; w^0)| - \epsilon_{stat}$$

$$\geq \min_i (r_i)^2 |\hat{\zeta}(S; w^0)|/|4 r_1 r_2| - |\hat{\zeta}(\emptyset; w^0)| - \epsilon_{stat} \qquad \text{by Eq. (48)}$$

$$\geq (|\hat{\zeta}(S; w^0)|/4)(\min_i |r_i|)/(\max_i |r_i|) - |\hat{\zeta}(\emptyset; w^0)| - \epsilon_{stat}$$

$$> \tau \qquad \text{by Eq. (42)}$$

Therefore, since $|a_{e_1}^{SGD}|, |a_{e_2}^{SGD}|, |b_v^{SGD}| > \tau$, the rounding in Line 4 of TRAINNEURON keeps the weights from NEURONSGD unchanged. $\square$

We may now begin to prove the items of Lemma D.12.

**Claim D.19.** *Item 2 holds.*

*Proof.* This is true because by Eq. (55), we have $\max_i |a_{e_i}^{SGD}| \leq 4\sqrt{\frac{\lambda_2}{\lambda_1}|\hat{\zeta}(S; w^0)|/|r_1 r_2|}$, and by the previous claim we have $a_{e_1}^{SGD} = a_{e_1}^{\text{round}}$ and $a_{e_2}^{SGD} = a_{e_2}^{\text{round}}$. $\square$

We now proceed to analyze the relative error of the active neuron that is created.

**Claim D.20.** *Neuron $v$ becomes active, with low relative error: i.e., $f_v(x; w^{\text{round}}) = r\chi_S(x) + h(x)$, where $r = 2r_1 r_2 a_{e_1}^{SGD} a_{e_2}^{SGD}$ and $h(x) \leq |r|\epsilon_{newrel}$ for any $x$, and $\hat{h}(S) = 0$. This is the first half of Item 1 of the lemma.*

*Proof.*

$$f_v(x; w^{\text{round}}) = b_v^{\text{round}} + (a_{e_1}^{\text{round}}(r_1 \chi_{S_1}(x) + h_1(x)) + a_{e_2}^{\text{round}} r_2 \chi_{S_2}(x))^2 \qquad \text{since } \epsilon_2 = 0$$

$$= b_v^{SGD} + (a_{e_1}^{SGD}(r_1 \chi_{S_1}(x) + h_1(x)) + a_{e_2}^{SGD} r_2 \chi_{S_2}(x))^2 \qquad \text{by Claim D.18}$$

$$= 2r_1 r_2 a_{e_1}^{SGD} a_{e_2}^{SGD} \chi_{S_1}(x)\chi_{S_2}(x) + T_1(x) + T_2(x) \qquad = r\chi_S(x) + h(x),$$

where we have defined $r = 2r_1r_2a_{e_1}^{SGD}a_{e_2}^{SGD}$, $h(x) = T_1(x) + T_2(x)$, and

$$T_1(x) = b_v^{SGD} + (r_1a_{e_1}^{SGD})^2 + (r_2a_{e_2}^{SGD})^2$$
$$T_2(x) = (a_{e_1}^{SGD}h_1(x))(2r_1a_{e_1}^{SGD}\chi_{S_1}(x) + a_{e_1}^{SGD}h_1(x) + 2r_2a_{e_2}^{SGD}\chi_{S_2}(x)).$$

Note that $\hat{h}(S) = 0$ since $\hat{T}_1(S) = 0$ and $\hat{T}_2(S) = 0$ and by linearity of the Fourier transform $\hat{h}(S) = \hat{T}_1(S) + \hat{T}_2(S)$. In particular, $\hat{T}_1(S) = 0$ because $T_1(x)$ is a constant and $S \neq \emptyset$ because since $S_1$ and $S_2$ are disjoint we have $S = (S_1 \cup S_2) \setminus (S_1 \cap S_2) = S_1 \cup S_2 \supset S_2 \neq \emptyset$. Further, $\hat{T}_2(S) = 0$ since, first of all, $h_1(x)\chi_{S_1}(x)$ and $h_1(x)^2$ depend only on variables in $S_1$ so they cannot be correlated with $\chi_S$ because, which depends on all the variables in $S_2$, which is nonempty. And, secondly, $\hat{h}_1(S_1) = 0$ by Assumption D.1, so $h_1(x)\chi_{S_2}(x)$ cannot be correlated to $\chi_S(x)$ because $S_2$ is nonempty.

In order to bound $|T_1(x)|$, let us first compute the derivative of the regularized loss with respect to $b_v$:

$$\frac{\partial \ell_R}{\partial b_v} = \frac{\partial \ell}{\partial b_v}$$
$$= \frac{\partial}{\partial b_v}\mathbb{E}_{x\sim\{-1,1\}^n}[\frac{1}{2}(b_v + (\sum_{i\in[2]} a_{e_i}f_{u_i}(x;w^0))^2 + \zeta(x;w^0))^2]$$
$$= \mathbb{E}_{x\sim\{-1,1\}^n}[b_v + (\sum_{i\in[2]} a_{e_i}f_{u_i}(x;w^0))^2 + \zeta(x;w^0)]$$
$$= b_v + \hat{\zeta}(\emptyset;w^0) + \mathbb{E}_{x\sim\{-1,1\}^n}[(\sum_{i\in[2]} a_{e_i}f_{u_i}(x;w^0))^2]$$
$$= b_v + \hat{\zeta}(\emptyset;w^0) + (r_1a_{e_1})^2 + (r_2a_{e_2})^2$$
$$+ \mathbb{E}_{x\sim\{-1,1\}^n}[2r_2a_{e_1}a_{e_2}(r_1\chi_{S_1}(x) + h_1(x))\chi_{S_2}(x) + 2r_1(a_{e_1})^2\chi_{S_1}(x)h_1(x) + (a_{e_1}h_1(x))^2]$$
$$= b_v + \hat{\zeta}(\emptyset;w^0) + (r_1a_{e_1})^2 + (r_2a_{e_2})^2 + \mathbb{E}_{x\sim\{-1,1\}^n}[(a_{e_1}h_1(x))^2],$$

where in the last line we use that $\mathbb{E}_{x\sim\{-1,1\}^n}[\chi_{S_1}(x)h_1(x)] = 0$ by Assumption D.1. We also use that $\mathbb{E}_{x\sim\{-1,1\}^n}[(\chi_{S_1}(x) + h_1(x))\chi_{S_2}(x)] = 0$ since $\chi_{S_1}(x) + h_1(x)$ only depends on $\{x_i\}_{i\in S_i}$ and $S_1 \cap S_2 = \emptyset$, and $S_2 \neq \emptyset$ by assumption. Therefore,

$$|T_1(x)| = |b_v + (r_1a_{e_1}^{SGD})^2 + (r_2a_{e_2}^{SGD})^2|$$
$$\leq \left|\frac{\partial\ell_R}{\partial b_v}|_{w=w^{SGD}}\right| + |\hat{\zeta}(\emptyset;w^0)| + |\mathbb{E}_{x\sim\{-1,1\}^n}[(a_{e_1}^{SGD}h_1(x))^2]|$$
$$\leq 2\epsilon_{stop} + |\hat{\zeta}(\emptyset;w^0)| + |\mathbb{E}_{x\sim\{-1,1\}^n}[(a_{e_1}^{SGD}h_1(x))^2]| \qquad \text{by Lemma D.5}$$
$$\leq 2\epsilon_{stop} + |\hat{\zeta}(\emptyset;w^0)| + (a_{e_1}^{SGD}r_1\epsilon_1)^2 \qquad \text{since } |h_1(x)| \leq |r_1\epsilon_1|$$
$$\leq 2\epsilon_{stop} + |\hat{\zeta}(\emptyset;w^0)| + 16\frac{\lambda_2}{\lambda_1}|\hat{\zeta}(S;w^0)||r_1\epsilon_1|^2/|r_1r_2| \qquad \text{by Eq. (55)}$$
$$\leq 2\epsilon_{stop} + |\hat{\zeta}(\emptyset;w^0)| + 16\frac{\lambda_2}{\lambda_1}|\hat{\zeta}(S;w^0)||\epsilon_1|^2|r_1/r_2|$$

We now bound $|T_2(x)|$. Since $|h_1(x)| \leq |r_1|\epsilon_1$, and $|\chi_{S_1}(x)|, |\chi_{S_2}(x)| \leq 1$,

$$|T_2(x)| \leq 2\epsilon_1|r_1a_{e_1}^{SGD}|(|r_1a_{e_1}^{SGD}|(1 + \epsilon_1) + |r_2a_{e_2}^{SGD}|).$$

By the above bounds on $T_1(x)$ and $T_2(x)$, we have

$$
\begin{aligned}
|h(x)| &\leq |T_1(x)| + |T_2(x)| \\
&\leq |T_1(x)| + 2\epsilon_1 |r_1 a_{e_1}^{SGD}|(|r_1 a_{e_1}^{SGD}|(1+\epsilon_1) + |r_2 a_{e_2}^{SGD}|) \\
&\leq |r|(|T_1(x)|/|r| + \epsilon_1(\frac{|r_1 a_{e_1}^{SGD}|}{|r_2 a_{e_2}^{SGD}|}(1+\epsilon_1) + 1)) && \text{by definition of } r \\
&\leq |r|(|T_1(x)|/|r| + \epsilon_1(2\frac{|r_1 a_{e_1}^{SGD}|}{|r_2 a_{e_2}^{SGD}|} + 1)) && \text{by } \epsilon_1 \leq 1 \\
&\leq |r|(|T_1(x)|/|r| + \epsilon_1(8\frac{r_1}{r_2}\sqrt{\frac{\gamma_2}{\gamma_1}} + 1) && \text{by Eq. (52)} \\
&= |r|(|T_1(x)|/|2r_1 r_2 a_{e_1}^{SGD} a_{e_2}^{SGD}| + \epsilon_1(8\frac{r_1}{r_2}\sqrt{\frac{\gamma_2}{\gamma_1}} + 1)) && \text{by definition of } r \\
&\leq |r|(2|T_1(x)|/|\hat{\zeta}(S;w^0)| + \epsilon_1(8\frac{r_1}{r_2}\sqrt{\frac{\gamma_2}{\gamma_1}} + 1)) && \text{by Eq. (51)} \\
&\leq |r|((4\epsilon_{stop} + 2|\hat{\zeta}(\emptyset;w^0)|)/|\hat{\zeta}(S;w^0)| \\
&\qquad + 32\frac{\lambda_2}{\lambda_1}|\epsilon_1|^2|r_1/r_2| + \epsilon_1(8\frac{r_1}{r_2}\sqrt{\frac{\gamma_2}{\gamma_1}} + 1)) \\
&:= |r|\epsilon_{newrel}.
\end{aligned}
$$

$\square$

**Claim D.21.** *The error in the direction of $\chi_S$ is greatly reduced to close to zero:*

$$|r + \hat{\zeta}(S;w^0)| \leq \frac{4\sqrt{\gamma_1 \gamma_2}}{|r_1 r_2|}. \tag{56}$$

*This proves the second part of Item 1.*

*Proof.* By Eq. (32), for any distinct $i, j \in \{1, 2\}$, we have

$$\rho \leq \frac{\gamma_i |a_{e_i}^{SGD}| + \epsilon^{(2)}}{|a_{e_j}^{SGD}|}.$$

By the bound in Eq. (52) on the ratio of $|a_{e_i}^{SGD}|$ and $|a_{e_j}^{SGD}|$, this means:

$$
\begin{aligned}
\rho &\leq 4\sqrt{\gamma_j \gamma_i} + \frac{\epsilon^{(2)}}{|a_{e_j}^{SGD}|} \\
&\leq 4\sqrt{\gamma_1 \gamma_2} + \frac{\epsilon^{(2)}}{|a_{e_j}^{SGD}|}.
\end{aligned}
$$

By the lower bound on $\max_i |a_{e_i}^{SGD}|$ in Eq. (48), this implies

$$
\begin{aligned}
\rho &\leq 4\sqrt{\gamma_1 \gamma_2} + \frac{\epsilon^{(2)}}{\sqrt{|\hat{\zeta}(S;w^0)|/|4r_1 r_2|}} \\
&\leq 8\sqrt{\gamma_1 \gamma_2} && \text{by Eq. (40).}
\end{aligned}
$$

Since $\rho = (2r_1 r_2 a_{e_1}^{SGD} a_{e_2}^{SGD} + \hat{\zeta}(S;w^0))(2r_1 r_2) = (r + \hat{\zeta}(S;w^0))(2r_1 r_2)$, this proves Eq. (56):

$$|r + \hat{\zeta}(S;w^0)| \leq \frac{4\sqrt{\gamma_1 \gamma_2}}{|r_1 r_2|}.$$

$\square$

**Claim D.22.** *We now prove Item 3, which controls the final bias of the error:* $|\hat{\zeta}(\emptyset; w^{\text{round}})| \leq 2\epsilon_{stop}.$

*Proof.* Using Eq. (21),

$$\frac{\partial \ell_R}{\partial b_v} \big|_{w=w^{SGD}}$$

$$= \frac{\partial \ell}{\partial b_v} \big|_{w=w^{SGD}} \qquad\qquad\qquad\qquad\qquad R \text{ does not depend on } b_v$$

$$= \frac{\partial}{\partial b_v} \frac{1}{2} \mathbb{E}_{x\sim\{-1,1\}^n}[(f(x;w^0) + b_v^{SGD} + (\sum_{i\in[2]} a_{e_i}^{SGD} f_{u_i}(x;w^0))^2 - g(x))^2]$$

$$= \mathbb{E}_{x\sim\{-1,1\}^n}[(f(x;w^0) + b_v^{SGD} + (\sum_{i\in[2]} a_{e_i}^{SGD} f_{u_i}(x;w^0))^2 - g(x))]$$

$$= \mathbb{E}_{x\sim\{-1,1\}^n}[(f(x;w^0) + b_v^{\text{round}} + (\sum_{i\in[2]} a_{e_i}^{\text{round}} f_{u_i}(x;w^0))^2 - g(x))] \qquad \text{by Claim D.18}$$

$$= \mathbb{E}_{x\sim\{-1,1\}^n}[\zeta(x;w^{\text{round}})]$$

$$= |\hat{\zeta}(\emptyset;w^{\text{round}})|.$$

By event $E_{stat}$ and the stationarity guarantee in Item 2 of Lemma D.5, we have $\|\nabla_{w_v}\ell_R(w^{SGD})\|_\infty \leq 2\epsilon_{stop}$. $\qquad\square$

$\square$

### D.7.2 Proof of Lemma D.13

*Proof of Lemma D.13.* Let $E_{goodinit}$ be the event that the following conditions Eqs. (57) to (59) hold. These conditions imply that the random perturbation at initialization in Line 2 is "good," and ensure that the optimization in NEURONSGD will not fall into a saddle point or spurious local minimum:

$$|a_{e_1}^{perturb}|, |a_{e_2}^{perturb}| > \eta/2, \tag{57}$$

$$\text{sgn}(r_1 r_2 a_{e_1}^{perturb} a_{e_2}^{perturb}) = \text{sgn}(-\hat{\zeta}(S;w^0)) \tag{58}$$

$$|b_v^{perturb}| \leq \sqrt{|r_1 r_2 \hat{\zeta}(S;w^0)|} \eta/8. \tag{59}$$

Since $a_{e_1}^{perturb}, a_{e_2}^{perturb}, b_v^{perturb}$ are chosen i.i.d. uniformly at random from $[-\eta,\eta]$, the events that Eqs. (57) to (59) hold are independent of each other and of $w^0$. So

$$\mathbb{P}[E_{goodinit} \mid w^0] \geq (1/4)^2 \cdot (1/2) \cdot \min(1, \sqrt{|r_1 r_2 \hat{\zeta}(S;w^0)|}/8).$$

We make the following claim:

**Claim D.23.** *If* $(\neg E_{newactive}) \cap E_{stat} \cap E_{goodinit}$ *holds, then* $\ell_R(w^{perturb}) < \ell_R(w^{SGD})$.

On the other hand, Lemma D.5 guarantees that under the event $E_{stat}$ we have $\ell_R(w^{perturb}) \geq \ell_R(w^{SGD})$, so to avoid a contradiction we must have

$$\mathbb{P}[(\neg E_{newactive}) \cap E_{stat} \cap E_{goodinit} \mid w^0] = 0.$$

So by a union bound,

$$\mathbb{P}[E_{newactive} \cap E_{stat} \cap E_{goodinit} \mid w^0]$$
$$= \mathbb{P}[E_{stat} \cap E_{goodinit} \mid w^0] \geq 1 - \mathbb{P}[\neg E_{goodinit}] - \mathbb{P}[\neg E_{stat} \mid w^0]$$
$$\geq \min(1, \sqrt{|2r_1 r_2 \hat{\zeta}(S;w^0)|})/C - (1 - \mathbb{P}[E_{stat} \mid w^0]).$$

$\square$

It only remains to prove the helper claim:

*Proof of Claim D.23.* We begin by comparing $\tilde{\ell}(w^{SGD})$ and $\tilde{\ell}(w^{perturb})$. First, we lower-bound $\tilde{\ell}(w^{SGD})$ under event $(\neg E_{newactive}) \cap E_{stat}$:

$$\tilde{\ell}(w^{SGD}) - \frac{1}{2} \sum_{\substack{S' \subset [n] \\ S' \neq \emptyset, S}} (\hat{\zeta}(S'; w^0))^2$$

$$= \frac{1}{2}(2r_1 r_2 a_{e_1}^{SGD} a_{e_2}^{SGD} + \hat{\zeta}(S; w^0))^2 + \frac{1}{2}(r_1^2 (a_{e_1}^{SGD})^2 + r_2^2 (a_{e_2}^{SGD})^2 + b_v^{SGD} + \hat{\zeta}(\emptyset; w^0))^2$$

$$\geq \frac{1}{2}(2r_1 r_2 a_{e_1}^{SGD} a_{e_2}^{SGD} + \hat{\zeta}(S; w^0))^2$$

$$\geq \frac{1}{2}(-2|r_1 r_2|\tau^2 + |\hat{\zeta}(S; w^0)|)^2$$

where in the last line we use that that under event $(\neg E_{newactive}) \cap E_{stat}$ we have $|a_{e_1}^{SGD}|, |a_{e_2}^{SGD}| < \tau$ by Lemma D.11, and also $2|r_1 r_2|\tau^2 \leq |\hat{\zeta}(S; w^0)|$ by Eq. (45). On the other hand, under event $E_{goodinit}$:

$$(2r_1 r_2 a_{e_1}^{perturb} a_{e_2}^{perturb} + \hat{\zeta}(S; w^0))^2 \leq (2|r_1 r_2|(\eta/2)^2 - |\hat{\zeta}(S; w^0)|)^2,$$

since $|a_{e_1}^{perturb}|, |a_{e_2}^{perturb}| \leq \eta/2$ by Eq. (57), $\text{sgn}(2r_1 r_2 a_{e_1}^{perturb} a_{e_2}^{perturb}) = -\text{sgn}(\hat{\zeta}(S; w^0))$ by Eq. (58), and $2|r_1 r_2|(\eta/2)^2 \leq |\hat{\zeta}(S; w^0)|$ by Eq. (44).

Furthermore, by the fact that $|a_{e_1}^{perturb}|, |a_{e_2}^{perturb}| < \eta$,

$$(r_1^2 (a_{e_1}^{perturb})^2 + r_2^2 (a_{e_2}^{perturb})^2 + b_v^{perturb} + \hat{\zeta}(\emptyset; w^0))^2 \leq (r_1^2 \eta^2 + r_2^2 \eta^2 + |b_v^{perturb}| + |\hat{\zeta}(\emptyset; w^0)|)^2.$$

So combining the above bounds we obtain:

$$\tilde{\ell}(w^{perturb}) - \frac{1}{2} \sum_{\substack{S' \subset [n] \\ S' \neq \emptyset, S}} (\hat{\zeta}(S'; w^0))^2$$

$$= \frac{1}{2}(2r_1 r_2 a_{e_1}^{perturb} a_{e_2}^{perturb} + \hat{\zeta}(S; w^0))^2 + \frac{1}{2}(r_1^2 (a_{e_1}^{perturb})^2 + r_2^2 (a_{e_2}^{perturb})^2 + b_v^{perturb} + \hat{\zeta}(\emptyset; w^0))^2$$

$$\leq \frac{1}{2}(-|2r_1 r_2|(\eta/2)^2 + |\hat{\zeta}(S; w^0)|)^2 + \frac{1}{2}(r_1^2 \eta^2 + r_2^2 \eta^2 + |b_v^{perturb}| + |\hat{\zeta}(\emptyset; w^0)|)^2.$$

This implies that:

$$\tilde{\ell}(w^{SGD}) - \tilde{\ell}(w^{perturb})$$

$$\geq \frac{1}{2}|2r_1 r_2 \tau^2|^2 - |2r_1 r_2 \tau^2 \hat{\zeta}(S; w^0)| - \frac{1}{2}|r_1 r_2 \eta^2/2|^2 + |r_1 r_2 \eta^2 \hat{\zeta}(S; w^0)/2|$$

$$\quad - \frac{1}{2}(r_1^2 \eta^2 + r_2^2 \eta^2 + |b_v^{perturb}| + |\hat{\zeta}(\emptyset; w^0)|)^2$$

$$\geq \frac{1}{2}\left(-|2r_1 r_2 \tau^2 \hat{\zeta}(S; w^0)| + |r_1 r_2 \eta^2 \hat{\zeta}(S; w^0)/2|\right) - \frac{1}{2}(r_1^2 \eta^2 + r_2^2 \eta^2 + |b_v^{perturb}| + |\hat{\zeta}(\emptyset; w^0)|)^2,$$

where for the second inequality we use Eqs. (44) and (45). Thus, using $\eta \geq 4\tau$ by Eq. (43), we have

$$\tilde{\ell}(w^{SGD}) - \tilde{\ell}(w^{perturb})$$

$$\geq \frac{1}{8}|r_1 r_2 \eta^2 \hat{\zeta}(S; w^0)| - \frac{1}{2}(r_1^2 \eta^2 + r_2^2 \eta^2 + |b_v^{perturb}| + |\hat{\zeta}(\emptyset; w^0)|)^2$$

$$\geq \frac{1}{8}|r_1 r_2 \eta^2 \hat{\zeta}(S; w^0)| - (r_1^2 \eta^2 + r_2^2 \eta^2 + |\hat{\zeta}(\emptyset; w^0)|)^2 - 2|b_v^{perturb}|^2$$

$$\geq \frac{1}{16}|r_1 r_2 \eta^2 \hat{\zeta}(S; w^0)| - |b_v^{perturb}|^2 \qquad \text{by Eq. (46)}$$

$$\geq \frac{1}{32}|r_1 r_2 \eta^2 \hat{\zeta}(S; w^0)|. \qquad \text{by Eq. (59)} \qquad (60)$$

This now lets us prove that $\ell_R(w^{SGD}) \geq \ell_R(w^{perturb})$. In the first inequality we use that $w^0_{-v} = w^{SGD}_{-v} = w^{perturb}_{-v}$ and $w^0_v = \vec{0}$.

$$\ell_R(w^{SGD}) - \ell_R(w^{perturb}) = \ell(w^{SGD}) - \ell(w^{perturb}) + R(w^{SGD}) - R(w^{perturb})$$

$$\geq \ell(w^{SGD}) - \ell(w^{perturb}) + R(w^0) - R(w^0) - \frac{1}{2}\max(\lambda_1, \lambda_2) \sum_{e=(u,v)\in E} |a_e^{perturb}|^2$$

$$= \ell(w^{SGD}) - \ell(w^{perturb}) - \frac{1}{2}\max(\lambda_1, \lambda_2) \sum_{e=(u,v)\in E} |a_e^{perturb}|^2$$

$$\geq \ell(w^{SGD}) - \ell(w^{perturb}) - \max(\lambda_1, \lambda_2)W\eta^2$$

In the last line we have used $|a_e^{perturb}| \leq \eta$ for all $e = (u,v) \in E$, and also there are at most $2W$ possible edges feeding into $v$: $|\{(u,v) \in E\}| \leq 2W$. Now, by the triangle inequality and since the idealized loss $\tilde{\ell}$ is close to the true loss $\ell$,

$$\ell(w^{SGD}) - \ell(w^{perturb})$$
$$\geq \tilde{\ell}(w^{SGD}) - \tilde{\ell}(w^{perturb}) - |\tilde{\ell}(w^{SGD}) - \ell(w^{SGD})| - |\tilde{\ell}(w^{perturb}) - \ell(w^{perturb})|$$
$$\geq \tilde{\ell}(w^{SGD}) - \tilde{\ell}(w^{perturb})$$
$$\quad - C\max_{u\in V}\max_{x\in\{-1,1\}^n}(\|w_v^{perturb}\|_\infty^4 + \|w_v^{SGD}\|_\infty^4 + 1)(|f_u(x;w^0)|^3 + 1)(\max_{i\in\{1,2\}}\epsilon_i) \quad \text{by Eq. (22)},$$

for some large constant $C > 0$. Plugging in the bound $\|w_v^{perturb}\|_\infty \leq \eta \leq 1$ by construction, and $\|w_v^{SGD}\|_\infty < \tau \leq 1$ from Lemma D.11 under event $(\neg E_{newactive}) \cap E_{stat}$, we have

$$\ell(w^{SGD}) - \ell(w^{perturb})$$
$$\geq \tilde{\ell}(w^{SGD}) - \tilde{\ell}(w^{perturb}) - C'\max_{u\in V}\max_{x\in\{-1,1\}^n}(|f_u(x;w^0)|^3 + 1)(\max_{i\in\{1,2\}}\epsilon_i),$$

for some large enough constant $C' > 0$. So combining the above bounds:

$$\ell_R(w^{SGD}) - \ell_R(w^{perturb}) \geq \frac{1}{32}|r_1 r_2 \eta^2 \hat{\zeta}(S;w^0)| - \lambda_2 W\eta^2 - C'\max_{u\in V}\max_{x\in\{-1,1\}^n}(|f_u(x;w^0)|^3 + 1)(\max_{i\in\{1,2\}}\epsilon_i)$$
$$> 0,$$

by Eq. (47), taking a large enough constant in Eq. (47). Thus, we conclude that $\ell_R(w^{perturb}) < \ell_R(w^{SGD})$. □

# E   Correctness of TRAINNETWORKLAYERWISE (proof of Theorem B.1)

In this section, we prove Theorem B.1 by using Lemmas D.7, D.8 and D.11 to D.13 to prove that certain events hold with high probability during the execution of TRAINNETWORKLAYERWISE (Algorithm 1). A key property that we will prove is maintained throughout training is that every active neuron computes some monomial up to a good relative approximation. Let us formalize the notion of approximation, as it will be needed later:

**Definition E.1.** *Let $v \in V$ be a neuron, and let $S \subset [n]$ be a subset of indices. We say that $v$ computes the monomial $\chi_S$ up to relative error $\epsilon_{rel}$ if there is some $r \in \mathbb{R}$ such that for all $x \in \{-1,1\}^n$ we have*

$$|f_v(x) - r\chi_S| \leq \epsilon_{rel}|r|.$$

*We call $r$ the "scaling" factor for neuron $v$'s approximation.*

As an example, for any $i \in [n]$, the input $v_{\text{in},i}$ computes $x_i = \chi_{\{i\}}$ with zero relative error, since $f_{v_{\text{in},i}}(x) = x_i$ for all $x$. And furthermore $v_{\text{in},0}$ computes the monomial $1 = \chi_\emptyset$ with zero relative error, since $f_{v_{\text{in},0}}(x) = 1$ for all $x$.

## E.1 Definition of events

The basis of our proof is showing that certain events and invariants hold with high probability during training. We now define them. In order to do this, recall the definition of the error function at iteration $t \in \{0, \ldots, WL\}$ of TRAINNETWORKLAYERWISE:

$$\zeta(x; w^t) = f(x; w^t) - g(x),$$

and recall its Fourier coefficients:

$$\hat{\zeta}(S; w^t) = \mathbb{E}_{x \sim \{-1,1\}^n}[\zeta(x; w^t)\chi_S(x)].$$

### E.1.1 Representation of monomials events

The first group of events states that all of the neurons in the neural network are either blank or represent a monomial approximately. Furthermore, they state that all low-order monomials in $g$ of degree at most $i + 1$ are represented in the network after the first $i$ layers have been trained.

**Definition E.2.** *We say that a subset $S \subset [n]$ is represented at iteration $t$ with scaling $r$ and relative error $\epsilon_{rel}$ if there is a neuron $u \in V \setminus V_{\text{in}}$ such that for all $x \in \{-1, 1\}^n$:*

- *$f_u(x; w^t) = r\chi_S(x) + h(x)$, where*

- *$|h(x)| \leq \epsilon_{rel}$ and $\hat{h}(S) = 0$.*

*We write that neuron $u$ represents (the monomial corresponding to) $S$.*

**Definition E.3** ($\epsilon_{rel,}, \epsilon_{fourmove}, \epsilon_{learned}$). *Let $\epsilon_{rel,0} = 0$, and for any $i \in [L]$ inductively define*

$$\epsilon_{rel,i} = 16M\epsilon_{stop} + 128M^2\frac{\lambda_2}{\lambda_1}(\epsilon_{rel,i-1})^2 + (32M^2\sqrt{\frac{\lambda_1}{\lambda_2}} + 1)\epsilon_{rel,i-1}$$

*Furthermore, define*

$$\epsilon_{fourmove} = 100M^2L\epsilon_{stop}$$

*And let*

$$\epsilon_{learned} = 16M^2\lambda_2$$

For any $i \in \{0, 1, \ldots, L\}$, let

$$t_i = Wi$$

be the iteration at which layers $1, \ldots, i$ have been trained in TRAINNETWORKLAYERWISE.

**Definition E.4.** *For any $t \in (t_{i-1}, t_i]$, let $E_{rep,S,t}$ be the event that at time $t$ there is exactly one neuron $u_S$ representing $S$, with relative error $\epsilon_S \leq \epsilon_{rel,i}$, and with scaling factor $r_S$ such that $|r_S - \hat{g}(S)| \leq \epsilon_{fourmove}t + \epsilon_{learned}$.*

**Definition E.5.** *For any $i \in [L]$ and $t \in (t_{i-1}, t_i]$, let $E_{nobadactive,t}$ be the event that, for any neuron $v \in V \setminus V_{\text{in}}$ that is active at iteration $t$ (i.e., such that there exists $x$ with $f_v(x; w^t) \neq 0$), $v$ represents $S$ such that $\hat{g}(S) \neq 0$ and $|S| \leq i + 1$.*

**Definition E.6** (Event: first $i$ layers represent all monomials of degree at most $i+1$). *For convenience of notation, let $E_{replayer,0}$ to be an event that always occurs. For any $i \in \{1, \ldots, L\}$, let $E_{replayer,i}$ be the event that: $E_{rep,S,t_i}$ holds for each $S \subset [n]$ such that $|S| \leq i + 1$ and $\hat{g}(S) \neq \emptyset$, and that $E_{nobadactive,t_i}$ holds.*

**Definition E.7** (Polarization of Fourier coefficients). *For any $t \in [WL]$, let $E_{pol,t}$ be the event that for any $S \subset [n]$;*

- *If $\hat{g}(S) = 0$, then $|\hat{\zeta}(S; w^t)| \leq \epsilon_{fourmove}t$.*

- *If $\hat{g}(S) \neq 0$ and $S$ is represented in the network at time $t$, then $|\hat{\zeta}(S; w^t)| \leq \epsilon_{learned} + \epsilon_{fourmove}t$.*

- *If $\hat{g}(S) \neq 0$ and $S$ is not represented in the network at time $t$, then $|\hat{\zeta}(S; w^t) + \hat{g}(S)| \leq \epsilon_{fourmove}t$.*

### E.1.2 Boundedness of bias and network parameters invariants

In order to apply the guarantees for TRAINNEURON, we also need to maintain certain technical events that ensure that the parameters and weights of the network do not blow up too much. This ensure smoothness of the objective during training.

**Definition E.8.** *For any $t \in \{0, \ldots, t_L\}$, let $E_{bias,t}$ be the event that $|\hat{\zeta}(\emptyset; w^t)| \leq 2\epsilon_{stop} \leq \epsilon_{stat}$. In other words, this is the event that on iteration $t$ the error is unbiased.*

**Definition E.9.** *For any $t \in \{0, \ldots, t_L\}$, the $E_{neurbound,t}$ event is that all neurons at iteration $t$ have magnitude upper-bounded by $2M$: i.e.,*

$$\max_{u \in V} \max_{x \in \{-1,1\}^n} |f_u(x; w^t)| \leq 2M. \tag{61}$$

**Definition E.10.** *For any $t \in \{0, \ldots, t_L\}$, the $E_{parambound,t}$ event is that at iteration $t$ we have the following bound on the trained weights:*

$$\max_{e \in E} |a_e^t| \leq 16M^2\sqrt{\lambda_2/\lambda_1}. \tag{62}$$

### E.1.3 Network connectivity events

Finally, we have certain events that control the connectivity structure of the network. First, we ensure (because of the sparsity of the network), that every neuron has at most two active inputs and if it has two then one of them is from $V_{\text{in}}$.

**Definition E.11.** *For simplicity of the definition, let $V_0 = \emptyset$.*

*For any $i \in \{0, \ldots, L-1\}$, $E_{nothree,i}$ be the event that after training layers $1, \ldots, i$ (i.e., at iteration $t_i$), there is no $v_i \in V_i$ such that*

$$|\{(u, v_i) \in E \text{ such that } u \text{ is active at iteration } t_i\}| \geq 3,$$

*and also there is no $v_i \in V_i$ such that*

$$|\{(v_{i-1}, v_i) \in E \text{ such that } v_{i-1} \text{ is active at iteration } t_i \text{ and } v_{i-1} \in V_{i-1}\}| \geq 2.$$

Second, we also ensure that the network architecture is sufficiently connected that the product of any pair of trained neurons can be learned.

**Definition E.12.** *For simplicity of the definition, let $V_0 = \emptyset$. Let*

$$n_{shared} = 64M^2 \log(16sL/\delta) \tag{63}$$

*For any $i \in \{0, \ldots, L-1\}$, let $E_{conn,i}$ be the event that, at iteration $t_i$, for all distinct pairs of neurons $u, u' \in \{u \in V_{i-1} \cup V_{\text{in}} \text{ s.t. } u \text{ is active at iteration } t_i\}$,*

$$|\{v_i \in V_i : (u, v_i) \in E, (u', v_i) \in E\}| \geq n_{shared}.$$

### E.2 $E_{replayer,L}$ suffices to ensure learning

We now show that the $E_{replayer,L}$ event is enough to prove that the loss is bounded by $\epsilon$ at the final iteration $t_L$ (proved in Lemma E.15). Thus, the goal of the remainder of the proof will be to show that $E_{replayer,L}$ occurs with high probability.

**Claim E.13** (Bounded relative error during training)**.** *For all $i \in \{0, \ldots, L\}$, $\epsilon_{rel,i} \leq (1 + 1/L)^i(16Mi\epsilon_{stop})$. In particular, $\epsilon_{rel,i} \leq 45ML\epsilon_{stop} \leq \epsilon/(2Ms) \leq 1/2$.*

*Proof.* The proof is by induction on $i$. In the base case $i = 0$ we have $\epsilon_{rel,0} = 0$, satisfying the bound. For $i \in \{1, \ldots, L\}$, we have by the inductive hypothesis

$$\epsilon_{rel,i} = 16M\epsilon_{stop} + 128M^2\frac{\lambda_2}{\lambda_1}((1 + 1/L)^{i-1}(16M(i-1)\epsilon_{stop}))^2$$

$$+ (32M^2\sqrt{\frac{\lambda_1}{\lambda_2}} + 1)(1 + 1/L)^{i-1}(16M(i-1)\epsilon_{stop}).$$

Since

$$128M^2\frac{\lambda_2}{\lambda_1}((1+1/L)^{i-1}(16M(i-1)\epsilon_{stop}))$$

$$\leq 128M^2\frac{\lambda_2}{\lambda_1}((1+1/L)^L(16ML\epsilon_{stop}))$$

$$= 128M^2(64M^2L)^2((1+1/L)^L(16ML\epsilon_{stop})) \qquad \text{by Eq. (11)}$$

$$= 2^{25}M^7L^3\epsilon_{stop}$$

$$\leq 1/(2L) \qquad \text{by Eq. (12)}$$

and

$$32M^2\sqrt{\frac{\lambda_1}{\lambda_2}} = 1/(2L),$$

this means

$$\epsilon_{rel,i} \leq 16M\epsilon_{stop} + (1/(2L)+1/(2L))(1+1/L)^{i-1}(16M(i-1)\epsilon_{stop})$$

$$\leq 16M\epsilon_{stop} + (1+1/L)^i(16M(i-1)\epsilon_{stop})$$

$$\leq (1+1/L)^i(16Mi\epsilon_{stop}).$$

This proves the first part of the claim. To see the second part, note that, for any $i \in \{0, \ldots, L\}$

$$\epsilon_{rel,i} \leq (1+1/L)^L(16ML\epsilon_{stop}) \leq 45ML\epsilon_{stop} \leq \epsilon/(2Ms) \leq 1/2.$$

$\square$

**Claim E.14** (Bounded error in Fourier coefficients during training). *For any $t \leq WL$,*

$$\epsilon_{fourmove}t + \epsilon_{learned} \leq 32M^2\lambda_2 \leq \epsilon/(4Ms) \leq 1/(4M).$$

*Proof.*

$$\epsilon_{fourmove}t + \epsilon_{learned} \leq \epsilon_{fourmove}WL + \epsilon_{learned}$$

$$= 100M^2WL^2\epsilon_{stop} + 16M^2\lambda_2$$

$$\leq 32M^2\lambda_2 \qquad \text{by } \epsilon_{stop} \leq \lambda_2/(100WL^2) \text{ in Eq. (12)}$$

$$\leq \epsilon/(4Ms) \qquad \text{by } \lambda_2 \leq \epsilon/(128M^3s) \text{ in Eq. (10)}$$

$$\leq 1/(4M). \qquad \text{since } s \geq 1, \epsilon \leq 1$$

$\square$

**Lemma E.15.** *If $E_{replayer,L}$ holds, then $\ell(w^{t_L}) \leq \epsilon$.*

*Proof.* Since $L \geq n-1$, the event $E_{replayer,L}$ states that the active neurons of the network at iteration $t_L$ are in bijective correspondence with the subsets $S \subset [n]$ such that $\hat{g}(S) \neq 0$. In other words, for each $S$ such that $\hat{g}(S) \neq 0$ there is exactly one neuron $u_S$ such that $f_{u_S}(x; w^{t_L}) = r_S\chi_S(x) + h_S(x)$ where $|h_S(x)| \leq \epsilon_{rel,L}|r_S|$ and $|r_S - \hat{g}(S)| \leq \epsilon_{fourmove}t_i + \epsilon_{learned}$. Furthermore, there are no other active neurons, meaning that:

$$f(x; w^{t_L}) = \sum_{S:\hat{g}(S)\neq 0} f_{u_S}(x; w^{t_L}).$$

This implies that the error function is always bounded:

$$
\begin{aligned}
|\zeta(x; w^{t_L})| &= |f(x; w^{t_L}) - g(x)| \\
&= |\sum_{S:\hat{g}(S)\neq 0} f_{u_S}(x; w^{t_L}) - \hat{g}(S)\chi_S(x)| \\
&= |\sum_{S:\hat{g}(S)\neq 0} (r_S - \hat{g}(S))\chi_S(x) + h_S(x)| \\
&\leq \sum_{S:\hat{g}(S)\neq 0} |r_S - \hat{g}(S)| + |h_S(x)| \\
&\leq \sum_{S:\hat{g}(S)\neq 0} |r_S - \hat{g}(S)| + \epsilon_{rel,L}|r_S| \\
&\leq \sum_{S:\hat{g}(S)\neq 0} |r_S - \hat{g}(S)|(1 + \epsilon_{rel,L}) + \epsilon_{rel,L}|\hat{g}(S)| \\
&\leq \sum_{S:\hat{g}(S)\neq 0} 2|r_S - \hat{g}(S)| + \epsilon_{rel,L}|\hat{g}(S)| && \text{by Claim E.13} \\
&\leq \sum_{S:\hat{g}(S)\neq 0} 2(\epsilon_{fourmove}t_L + \epsilon_{learned}) + M\epsilon_{rel,L} \\
&\leq 2s(\epsilon_{fourmove}t_L + \epsilon_{learned}) + Ms\epsilon_{rel,L} \\
&\leq 2s(\epsilon_{fourmove}t_L + \epsilon_{learned}) + \epsilon/2 && \text{by Claim E.13} \\
&\leq \epsilon/2 + \epsilon/2 && \text{by Claim E.14} \\
&= \epsilon.
\end{aligned}
$$

As a consequence, we may bound the loss at the final time step $t_L$:

$$
\begin{aligned}
\ell(w^{t_L}) &= \mathbb{E}_{x\sim\{-1,1\}^n}[\ell(x; w^{t_L})] \\
&= \mathbb{E}_{x\sim\{-1,1\}^n}[\zeta(x; w^{t_L})^2] \\
&\leq \mathbb{E}_{x\sim\{-1,1\}^n}[\epsilon^2] \\
&\leq \epsilon^2 \\
&\leq \epsilon.
\end{aligned}
$$

$\square$

## E.3   $E_{replayer,L}$ occurs with high probability

In this section, we prove that $E_{replayer,L}$ occurs with high probability, essentially concluding the proof of the theorem because of Lemma E.15. First, we define the intersection of the events defined above, which we will show holds with high probability by induction on the iteration number.

**Definition E.16.** *Define the event*

$$
E_{stepgood,0} = E_{pol,0} \cap E_{neurbound,0} \cap E_{bias,0} \cap E_{parambound,0} \cap E_{nobadactive,0}.
$$

*For any $t \in \{1, \ldots, t_L\}$, inductively define the event*

$$
E_{stepgood,t} = E_{stepgood,t-1} \cap E_{pol,t} \cap E_{neurbound,t} \cap E_{parambound,t} \cap E_{bias,t} \cap E_{nobadactive,t}.
$$

**Definition E.17.** *Define the event*

$$
E_{layergood,0} = E_{conn,0} \cap E_{nothree,0} \cap E_{replayer,0}.
$$

*For any $i \in \{1, \ldots, L\}$, inductively define the event*

$$
E_{layergood,i} = E_{layergood,i-1} \cap E_{replayer,i} \cap E_{conn,i} \cap E_{nothree,i}.
$$

### E.3.1 $E_{stepgood,t}$ follows from $E_{stepgood,t-1}$ and $E_{layergood,i-1}$ with high probability

The first element of our induction is given by Lemma E.20 below. It bounds the runtime of an iteration of TRAINNETWORKLAYERWISE and proves that with high probability the event $E_{stepgood,}$ continues to hold. First, we prove a couple of helper claims.

**Claim E.18.** *Let $t \in \{0, \ldots, t_L\}$. Under the event $E_{stepgood,t}$, we have*

$$\ell_R(w^t) \leq 2^{22} W^2 L^3 M^8 s^2 \tag{64}$$

*Proof.* We bound the regularized loss at $w^t$, using that under the event $E_{stepgood,t}$, both $E_{neurbound,t}$ and $E_{parambound,t}$ hold:

$$
\begin{aligned}
\ell_R(w^t) &= \ell(w^t) + R(w^t) \\
&= \mathbb{E}_{x \sim \{-1,1\}^n}[\ell(x; w^t)] + R(w^t) \\
&\leq \max_x |\ell(x; w^t)| + R(w^t) \\
&\leq \max_x (f(x; w^t) - g(x))^2 + R(w^t) \\
&\leq \max_x 2g(x)^2 + 2f(x; w^t)^2 + R(w^t) \\
&\leq (Ms)^2 + 2f(x; w^t)^2 + R(w^t) && \text{by Claim C.5} \\
&\leq (Ms)^2 + 2WL \max_u \max_x f_u(x; w^t)^2 + R(w^t) \\
&\leq (Ms)^2 + 8WLM^2 + R(w^t) && \text{by Eq. (61), since } E_{neurbound,t} \text{ holds} \\
&\leq (Ms)^2 + 8WLM^2 + \sum_{e \in E} (a_e^t)^2 && \text{since } \lambda_1, \lambda_2 \leq 1 \\
&\leq (Ms)^2 + 8WLM^2 + 2W^2 L \max_e |a_e^t|^2 \\
&\leq (Ms)^2 + 8WLM^2 + 2W^2 L (16M^2 \sqrt{\lambda_2/\lambda_1})^2 && \text{by Eq. (62), since } E_{parambound,t} \text{ holds} \\
&\leq (Ms)^2 + 8WLM^2 + 2W^2 L (2^{20} M^8 L^2) && \text{by Eq. (11)} \\
&\leq 2^{22} W^2 L^3 M^8 s^2.
\end{aligned}
$$

$\square$

Let $E_{stat,t}$ denote the event that on call $t$ to TRAINNEURON, the event $E_{stat}$ from Lemma D.5 holds.

**Lemma E.19.** *Let*

$$\delta_{stat} = \delta/(64LWsM^2). \tag{65}$$

*For any $t \in (t_{i-1}, \ldots, t_i]$, if the event $E_{stepgood,t-1} \cap E_{layergood,i-1}$ holds, then $\mathbb{P}[E_{stat,t} \mid w^{t-1}] \geq 1 - \delta_{stat}$. Furthermore, if $E_{stat}t$ holds then the call to TRAINNEURON exits after at most $O(\kappa^{2393})$ time.*

*Proof.* The lemma follows by applying Lemma D.5. Indeed, for some large enough constant $C$ so that we can apply Lemma D.5, we bound the learning rate by taking $c_\alpha > 0$ small enough:

$$
\begin{aligned}
\alpha &\leq 1/(C(\lambda_1\lambda_2)^{-5} 2^{56} \kappa^{72}) && \text{by Eq. (13)} \\
&\leq 1/(C(\lambda_1\lambda_2)^{-5} \kappa^{16}((2M)^{32} + 1)(\kappa^8((2M)^{16} + 1) + 2^{22}\kappa^8) \\
&\leq 1/(C(\lambda_1\lambda_2)^{-5} \kappa^{16}((2M)^{32} + 1)(\kappa^8((2M)^{16} + 1) + \ell_R(w^0)^4)) && \text{by Eq. (64)} \\
&< \min_{u \in V} \min_{x \in \{-1,1\}^n} (C(\lambda_1\lambda_2)^{-5}\kappa^{16})^{-1}(|f_u(x; w^0)|^{32} + 1)^{-1} \\
&\qquad\qquad \cdot (\kappa^8(|f_u(x; w^0)|^{16} + 1) + \ell_R(w^0)^4)^{-1} && \text{by } E_{neurbound,t-1}
\end{aligned}
$$

the bound on the number of iterations in Lemma D.5 is at most:

$$t_{max} = C(\kappa^2(\max_{u \in V} \max_{x \in \{-1,1\}^n} |f_u(x; w^0)|^4 + 1) + \ell_R(w^{t-1}))/(\alpha(\epsilon_{stop})^2)$$

$$\leq C(\kappa^2((2M)^4 + 1) + \ell_R(w^{t-1})/(\alpha(\epsilon_{stop})^2) \qquad \text{by } E_{neurbound,t-1}$$

$$\leq C(\kappa^2((2M)^4 + 1) + 2^{22}\kappa^8)/(\alpha(\epsilon_{stop})^2) \qquad \text{by Eq. (64)}$$

$$\leq 2^{23}C\kappa^{10}/(\alpha(\epsilon_{stop})^2)$$

$$\leq C^2(\lambda_1\lambda_2)^{-5}\kappa^{82}/(\epsilon_{stop})^2 \qquad \text{by Eq. (13)}$$

$$\leq C^3(\lambda_1\lambda_2)^{-5}\kappa^{942} \qquad \text{by Eq. (12)}$$

and the minibatch size satisfies the following because the constant $c_B$ is large enough:

$$B \geq C^4(\lambda_1\lambda_2)^{-4}\kappa^{910} \qquad \text{by Eq. (14)}$$

$$\geq C^3(\lambda_1\lambda_2)^{-4}\kappa^{909}\log(1/\delta_{stat}) \qquad \text{by Eq. (65)}$$

$$\geq C^2(\lambda_1\lambda_2)^{-3}(2^{98}\kappa^{908})\log(2t_{max}/\delta_{stat}) \qquad \text{by } t_{max} \text{ bound}$$

$$\geq C(\lambda_1\lambda_2)^{-3}(2^{98}\kappa^{48})\log(2t_{max}/\delta_{stat})/\epsilon_{stop}^2 \qquad \text{by Eq. (12)}$$

$$\geq C(\lambda_1\lambda_2)^{-3}\kappa^8((2M)^8 + 1)(\kappa^8(2M)^{16} + 2^{88}\kappa^{32})\log(2t_{max}/\delta_{stat})/\epsilon_{stop}^2$$

$$\geq C(\lambda_1\lambda_2)^{-3}\kappa^8((2M)^8 + 1)(\kappa^8(2M)^{16} + \ell_R(w^0)^4)\log(2t_{max}/\delta_{stat})/\epsilon_{stop}^2 \qquad \text{by Eq. (64)}$$

$$\geq C(\lambda_1\lambda_2)^{-3}\kappa^8(\max_{u \in V} \max_{x \in \{-1,1\}^n} |f_u(x; w^0)|^8 + 1)$$

$$\cdot (\kappa^8(\max_u |f_u(x; w^0)|^{16} + 1) + \ell_R(w^0)^4)\log(2t_{max}/\delta_{stat})/\epsilon_{stop}^2,$$

Thus, we can apply Lemma D.5 and derive the claimed bounds, including the runtime bound of $O(\kappa B t_{max}) = O((\lambda_1\lambda_2)^{-9}\kappa^{1853}) = O(\kappa^{2393})$. $\qquad \square$

Now we are ready to prove the main result of this subsection, which is the inductive step showing that $E_{stepgood,t}$ is maintained with high probability. The proof calls on the guarantees on TRAINNEURON proved in Lemmas D.7, D.8 and D.11 to D.13.

**Lemma E.20.** *For any layer $i \in [L]$ and iteration $t \in [WL]$ such that $t \in [t_{i-1} + 1, t_i]$ and $E_{stepgood,t-1} \cap E_{layergood,i-1}$ holds, then*

$$\mathbb{P}[E_{stepgood,t} \cap E_{stat,t} \mid w^{t-1}] \geq 1 - \delta_{stat}.$$

*Furthermore, if the neuron $v \in V_i$ trained in iteration $t$ has exactly two active parents representing $S_1$ and $S_2$, and $\hat{g}(S) \neq \emptyset$, and $\neg E_{rep,S,t-1}$ holds for $S = (S_1 \cup S_2) \setminus (S_1 \cap S_2)$, then with lower-bounded probability $v$ is trained to be a neuron that represents $S$:*

$$\mathbb{P}[E_{rep,S,t} \mid w^{t-1}] \geq 1/(64M^2).$$

*Proof.* We prove that if $E_{stat,t}$ occurs then $E_{stepgood,t}$ also occurs. This suffices to prove the first part of the claim since $\mathbb{P}[E_{stat,t} \mid w^{t-1}] \geq 1 - \delta_{stat}$ by Lemma E.19.

Let $v \in V_i$ in layer $i$ be the neuron that we update with TRAINNEURON on iteration $t$. Then by event $E_{nothree,i-1}$, $v$ must have at most two active parents at iteration $t - 1$: i.e.,

$$|\{u \in V : (u, v) \in E \text{ and } \exists x \text{ s.t. } f_u(x; w^{t-1}) \neq 0\}| \leq 2.$$

For ease of notation, let $u_1, u_2 \in V_{i-1} \cup V_{in}$ be two parents of $v$ such that $(u_1, v), (u_2, v) \in E$, and such that all other parents are blank[4]: if $(u, v) \in E$ and $u \notin \{u_1, u_2\}$, then $f_u(x; w^{t-1}) \equiv 0$. Write the functions computed at $u_1, u_2$ as

$$f_{u_j}(x; w^{t-1}) = r_j \chi_{S_j}(x) + h_j(x),$$

where $r_1, r_2 \in \mathbb{R}$, $S_1, S_2 \subset [n]$, and $\hat{h}_j(S_j) = 0$. Since $E_{layergood,i-1}$ implies $E_{replayer,i-1}$, we know that for any active neuron $u' \in V_{i-1} \cup V_{in}$ we have $f_{u'}(x; w^{t-1}) = r'\chi_{S'}(x) + h'(x)$, where

---

[4]This notation assumes that $v$ has at least two parents, but this is only for the sake of convenience since the case where $v$ has no parents or one parent essentially follows by the same arguments, letting $r_1 = r_2 = 0$ or $r_2 = 0$.

$\hat{h}'(S') = 0$ and $|r'| \le |r' - \hat{g}(S')| + |g(\hat{S}')| \le \epsilon_{learned} + \epsilon_{fourmove}(t-1) + M \le 1/(4M) + M \le 2M$ by Claim E.14, so

$$|r_1|, |r_2| \in \{0\} \cup [1/2M, 2M], \tag{66}$$

and there are $\epsilon_1, \epsilon_2 > 0$ satisfying

$$\epsilon_1, \epsilon_2 \le \epsilon_{rel,i-1} \tag{67}$$

such that for all $x \in \{-1,1\}^n$, $|h_j(x)| \le |r_j|\epsilon_j$. Therefore, we may bound $\epsilon_{stat}(w^{t-1}, \epsilon_1, \epsilon_2)$ and $U_{stat}(w^{t-1})$:

**Claim E.21.** *Under $E_{stepgood,t-1}$ and $E_{layergood,i-1}$, we have*

$$\epsilon_{stat}(w^{t-1}, \epsilon_1, \epsilon_2) \lesssim (\lambda_1\lambda_2)^{-3}\kappa^{67}\epsilon_{stop} \le 1 \tag{68}$$

$$U_{stat}(w^{t-1}) \lesssim (\lambda_1\lambda_2)^{-1}\kappa^{20} \tag{69}$$

*Proof.* We bound $\epsilon_{stat}(w^{t-1}, \epsilon_1, \epsilon_2)$, first recalling that by the definition in Lemma D.5,

$\epsilon_{stat}(w^{t-1}, \epsilon_1, \epsilon_2)$

$\lesssim \epsilon_{stop} + \max\limits_{u \in V} \max\limits_{x \in \{-1,1\}^n} (\lambda_1\lambda_2)^{-3}\kappa^{12}(\ell_R(w^{t-1})^3 + 1)(|f_u(x; w^{t-1})|^{30} + 1)(\max\limits_{i \in \{1,2\}} \epsilon_i)$

$\lesssim \epsilon_{stop} + \max\limits_{u \in V} \max\limits_{x \in \{-1,1\}^n} (\lambda_1\lambda_2)^{-3}\kappa^{12}(\ell_R(w^{t-1})^3 + 1)(|f_u(x; w^{t-1})|^{30} + 1)\epsilon_{rel,i-1}$ by Eq. (67)

$\lesssim \epsilon_{stop} + (\lambda_1\lambda_2)^{-3}\kappa^{12}(W^2 L^3 M^8 s^2)^3(|f_u(x; w^{t-1})|^{30} + 1)\epsilon_{rel,i-1}$ by Eq. (64)

$\lesssim \epsilon_{stop} + (\lambda_1\lambda_2)^{-3}\kappa^{12}(W^2 L^3 M^8 s^2)^3 M^{30}\epsilon_{rel,i-1}$ by Eq. (61)

$\le \epsilon_{stop} + (\lambda_1\lambda_2)^{-3}\kappa^{12+24+30}\epsilon_{rel,i-1}$

$= \epsilon_{stop} + (\lambda_1\lambda_2)^{-3}\kappa^{66}\epsilon_{rel,i-1}$

$\lesssim (\lambda_1\lambda_2)^{-3}\kappa^{67}\epsilon_{stop}$ by Claim E.13.

In the above bounds, we have used that Eq. (64) holds because $E_{stepgood,t-1}$ holds, and Eq. (61) holds because $E_{neurbound,t-1}$ holds by $E_{stepgood,t-1}$.

Similarly, we bound $U_{stat}(w^{t-1}, \epsilon_1, \epsilon_2)$, recalling the definition in Lemma D.5:

$U_{stat}(w^{t-1}) \lesssim (\lambda_1\lambda_2)^{-1}\kappa^4(\max\limits_u \max\limits_{x \in \{-1,1\}^n} |f_u(x; w^{t-1})|^8 + 1)(\ell_R(w^{t-1}) + 1)$

$\lesssim (\lambda_1\lambda_2)^{-1}\kappa^4(\max\limits_u \max\limits_{x \in \{-1,1\}^n} |f_u(x; w^{t-1})|^8 + 1)\kappa^8$ by Eq. (64)

$\lesssim (\lambda_1\lambda_2)^{-1}\kappa^4 M^8 \kappa^8$ by Eq. (61)

$= (\lambda_1\lambda_2)^{-1}\kappa^{20}.$

Here again, we have used Eq. (64) and Eq. (61) because $E_{stepgood,t-1}$ holds. $\square$

We may now break the analysis into cases, writing $\epsilon_{stat} = \epsilon_{stat}(w^{t-1}, \epsilon_1, \epsilon_2)$ and $U_{stat} = U_{stat}(w^t)$ for shorthand.

**Case 1: At most one active input** If $v$ has at most one active parent at iteration $t-1$, then $f_{u_2}(x; w^{t-1}) \equiv 0$ without loss of generality.

*Checking preconditions of Lemma D.7.* In this case, we apply Lemma D.7, first checking that the preconditions apply. In the below, let taking $C > 0$ to be a large enough universal constant. Eq. (25) applies since

$2\epsilon_{stat}/\min(\lambda_1, \lambda_2) \le 2\epsilon_{stat}/(\lambda_1\lambda_2)$

$\le C(\lambda_1\lambda_2)^{-4}\kappa^{67}\epsilon_{stop}$ by Eq. (68)

$\le 1/(2^{21} M^7 L)$ by $\epsilon_{stop} \le (\lambda_1\lambda_2)^4\kappa^{-74}/(2^{21}C)$ in Eq. (12)

$< \tau$ by Eq. (8)

Furthermore, Eq. (26) applies, since

$$
\begin{aligned}
|\hat{\zeta}(\emptyset; w^0)| &+ \epsilon_{stat} + (r_1)^2 |2\epsilon_{stat}/\min(\lambda_1, \lambda_2)|^2 \\
&\leq 2\epsilon_{stat} + (r_1)^2 |2\epsilon_{stat}/\min(\lambda_1, \lambda_2)|^2 && \text{by } E_{bias,t-1} \\
&\leq C(\lambda_1\lambda_2)^{-3}\kappa^{67}\epsilon_{stop}(1 + (r_1)^2|2/\min(\lambda_1, \lambda_2)|^2) && \text{by Eq. (68)} \\
&\leq C(\lambda_1\lambda_2)^{-3}\kappa^{67}\epsilon_{stop}(1 + (4M/\lambda_1)^2) && \text{since } |r_1| \leq 2M \\
&\leq 64C(\lambda_1\lambda_2)^{-3}\kappa^{67}\epsilon_{stop}/\lambda_1^2 && \text{by Eq. (8)} \\
&\leq \tau
\end{aligned}
$$

Finally, Eq. (27) applies since

$$
\begin{aligned}
\epsilon_{stat} &< C(\lambda_1\lambda_2)^{-3}\kappa^{67}\epsilon_{stop} && \text{by Eq. (68)} \\
&\leq \lambda_1/(4M)^2 && \text{by } \epsilon_{stop} \leq (\lambda_1\lambda_2)^4\kappa^{-69}/(4C) \text{ in Eq. (12)} \\
&\leq \min(\lambda_1, \lambda_2)/(2r_1)^2 && \text{by } |r_1| \leq 2M
\end{aligned}
$$

Thus, all the preconditions of Lemma D.7 hold.

*Applying Lemma D.7.* Therefore, under event $E_{stat,t}$ after running TRAINNEURON$(v, w^{t-1})$ we have $w^t = w^{t-1}$. So since the weights are unchanged and we assume that $E_{stat,t}$ holds, $E_{stepgood,t}$ follows from $E_{stepgood,t-1}$.

**Case 2: Exactly two active inputs**  If $v$ has exactly two active parents at iteration $t - 1$, then by Eq. (66), we must have $|r_1|, |r_2| \in [1/(2M), 2M]$. Furthermore, let $S = S_1 \cup S_2 \setminus (S_1 \cap S_2)$, so that $\chi_S(x) = \chi_{S_1}(x)\chi_{S_2}(x)$. We further subdivide into two cases, depending on whether $S$ is represented by a neuron in the network and whether $\hat{g}(S) = 0$.

In order to analyze this section, let us first upper-bound the quantities $\epsilon^{(1)}, \epsilon^{(2)}, \epsilon^{(3)}$,

$$
\begin{aligned}
\epsilon^{(1)} &= 2\epsilon_{stat}(w^{t-1}, \epsilon_1, \epsilon_2)/\min(\lambda_1, \lambda_2) \\
&\lesssim (\lambda_1\lambda_2)^{-3}\kappa^{67}\epsilon_{stop}/\min(\lambda, \lambda_2) \\
&\leq (\lambda_1\lambda_2)^{-4}\kappa^{67}\epsilon_{stop}. && (70)
\end{aligned}
$$

$$
\begin{aligned}
\epsilon^{(2)} &= (1 + \max_i r_i^2 U_{stat}(w^{t-1}))\epsilon_{stat}(w^{t-1}, \epsilon_1, \epsilon_2) \\
&\leq (1 + (2M)^2)U_{stat}(w^{t-1}))\epsilon_{stat}(w^{t-1}, \epsilon_1, \epsilon_2) && \text{by Eq. (66)} \\
&\lesssim (1 + (2M)^2)(\lambda_1\lambda_2)^{-1}\kappa^{20}\epsilon_{stat}(w^{t-1}, \epsilon_1, \epsilon_2) && \text{by Eq. (69)} \\
&\lesssim (1 + (2M)^2)(\lambda_1\lambda_2)^{-1}\kappa^{20}(\lambda_1\lambda_2)^{-3}\kappa^{67}\epsilon_{stop} && \text{by Eq. (68)} \\
&\lesssim (\lambda_1\lambda_2)^{-4}\kappa^{89}\epsilon_{stop} && (71)
\end{aligned}
$$

And since $E_{stepgood,t-1}$ implies $E_{pol,t-1}$, we have

$$
\begin{aligned}
|\hat{\zeta}(S; w^{t-1})| &\leq |\hat{g}(S)| + (t-1)\epsilon_{fourmove} + \epsilon_{learned} \\
&\leq |\hat{g}(S)| + (M/4) && \text{by Claim E.14} \\
&\leq 2M && \text{by } |\hat{g}(S)| \leq M \quad (72)
\end{aligned}
$$

$$
\begin{aligned}
\epsilon^{(3)} &= 8\epsilon^{(2)}(1 + (U_{stat})^2 + |\hat{\zeta}(S; w^{t-1})|)\max(1, |r_1 r_2|^2)/\min(\lambda_1, \lambda_2)^2 \\
&\lesssim ((\lambda_1\lambda_2)^{-4}\kappa^{89}\epsilon_{stop})(1 + (U_{stat})^2 + |\hat{\zeta}(S; w^{t-1})|)\frac{\max(1, |r_1 r_2|^2)}{\min(\lambda_1, \lambda_2)^2} && \text{by Eq. (71)} \\
&\leq ((\lambda_1\lambda_2)^{-6}\kappa^{89}\epsilon_{stop})(1 + (U_{stat})^2 + |\hat{\zeta}(S; w^{t-1})|)\max(1, |r_1 r_2|^2) \\
&\lesssim ((\lambda_1\lambda_2)^{-6}\kappa^{93}\epsilon_{stop})(1 + (U_{stat})^2 + |\hat{\zeta}(S; w^{t-1})|) && \text{by Eq. (66)} \\
&\lesssim ((\lambda_1\lambda_2)^{-6}\kappa^{93}\epsilon_{stop})((\lambda_1, \lambda_2)^{-2}\kappa^{40} + |\hat{\zeta}(S; w^{t-1})|) && \text{by Eq. (69)} \\
&\lesssim (\lambda_1\lambda_2)^{-8}\kappa^{133}\epsilon_{stop}. && \text{by Eq. (72)} \quad (73)
\end{aligned}
$$

**Case 2a: Exactly two active inputs, product is not useful**  Consider the case in which either $\hat{g}(S) = 0$ or $S$ is already represented by some neuron at iteration $t-1$ (i.e., $E_{rep,S,t-1}$ holds). Since $E_{stepgood,t-1}$ implies $E_{pol,t-1}$, we have the following bound

$$|\hat{\zeta}(S; w^{t-1})| \le (t-1)\epsilon_{fourmove} + \epsilon_{learned}$$
$$\le 32M^2\lambda_2. \qquad \text{by Claim E.14}$$

Using this, we will prove that the TRAINNEURON$(v; w^{t-1})$ will with high probability leave the neuron $v$ blank and the weights unchanged after training. Intuitively, this is because if neuron $v$ were trained to represent the monomial $\chi_S$, then this would not reduce the loss significantly since the Fourier coefficient $\hat{\zeta}(S; w^{t-1})$ of the error is small. Therefore the regularization term dominates and pushes the trained weights on this iteration to close to zero.

*Checking preconditions of Lemma D.8.* We apply Lemma D.8 to conduct our analysis. In order to check that the preconditions are satisfied, let us first upper bound the quantity $\epsilon^{(4)}$.

$$\epsilon^{(4)} = 2\left(\sqrt{\frac{\max(\lambda_1, \lambda_2)}{\min(\lambda_1, \lambda_2)}}\sqrt{\max(\lambda_1, \lambda_2) + |\hat{\zeta}(S; w^{t-1})|} + \epsilon^{(2)}\frac{\max(\lambda_1, \lambda_2))}{\min(1, |r_1 r_2|^2)}\right)$$

$$= 2(\sqrt{\lambda_2/\lambda_1}\sqrt{\lambda_2 + |\hat{\zeta}(S; w^{t-1})|} + \lambda_2\epsilon^{(2)})/\min(1, |r_1 r_2|^2)$$

$$\lesssim \kappa^4(\sqrt{\lambda_2/\lambda_1}\sqrt{\lambda_2 + |\hat{\zeta}(S; w^{t-1})|} + \lambda_2\epsilon^{(2)}) \qquad \text{by Eq. (66)}$$

$$\lesssim \kappa^4(\sqrt{\lambda_2/\lambda_1}\sqrt{\lambda_2 + |\hat{\zeta}(S; w^{t-1})|} + (\lambda_1\lambda_2)^{-4}\kappa^{89}\epsilon_{stop}) \qquad \text{by Eq. (71)}$$

$$\lesssim \kappa^4(\sqrt{\lambda_2/\lambda_1}\sqrt{\lambda_2 + |\hat{\zeta}(S; w^{t-1})|}) + (\lambda_1\lambda_2)^{-5}\kappa^{93}\epsilon_{stop}$$

$$\lesssim \kappa^4\sqrt{\lambda_2/\lambda_1}\sqrt{\lambda_2 + 32M^2\lambda_2} + (\lambda_1\lambda_2)^{-5}\kappa^{93}\epsilon_{stop}$$

$$\lesssim \kappa^6\sqrt{\lambda_2 + 32M^2\lambda_2} + (\lambda_1\lambda_2)^{-5}\kappa^{93}\epsilon_{stop} \qquad \text{by Eq. (11)}$$

$$\lesssim \kappa^7\sqrt{\lambda_2} + (\lambda_1\lambda_2)^{-5}\kappa^{93}\epsilon_{stop} \qquad (74)$$

In the following, let $C > 0$ be some large enough universal constant. Eq. (28) holds, because

$$\max(\epsilon^{(1)}, \epsilon^{(3)}, \epsilon^{(4)}) < C(\kappa^7\sqrt{\lambda_2} + (\lambda_1\lambda_2)^{-8}\kappa^{133}\epsilon_{stop}) \qquad \text{by Eqs. (70), (73) and (74)}$$

$$\le 1/(2^{21}M^7 L) + C(\lambda_1\lambda_2)^{-8}\kappa^{133}\epsilon_{stop} \qquad \text{since } \sqrt{\lambda_2} \le 1/(2^{21}\kappa^{14}C) \text{ by Eq. (10)}$$

$$\le 1/(2^{20}M^7 L) \qquad \text{since } \epsilon_{stop} \le 1/(2^{21}\kappa^{140}C) \text{ by Eq. (12)}$$

$$= \tau \qquad \text{by Eq. (8)}$$

And Eq. (29) holds,

$$|\hat{\zeta}(\emptyset; w^0)| + 2(r_1^2 + r_2^2)(\max(\epsilon^{(1)}, \epsilon^{(3)}, \epsilon^{(4)}))^2 + \epsilon_{stat}$$

$$\le 2(r_1^2 + r_2^2)(\max(\epsilon^{(1)}, \epsilon^{(3)}, \epsilon^{(4)}))^2 + 2\epsilon_{stat} \qquad \text{by } E_{bias,t-1}$$

$$\le 8M^2(\max(\epsilon^{(1)}, \epsilon^{(3)}, \epsilon^{(4)}))^2 + 2\epsilon_{stat} \qquad \text{by Eq. (66)}$$

$$\le 8M^2/(2^{20}M^7 L)^2 + 2\epsilon_{stat} \qquad \text{by Eqs. (70), (73) and (74)}$$

$$\le 1/(2^{21}M^7 L) + 2\epsilon_{stat}$$

$$\le 1/(2^{21}M^7 L) + C(\lambda_1\lambda_2)^{-3}\kappa^{67}\epsilon_{stop} \qquad \text{by Eq. (68)}$$

$$\le 1/(2^{20}M^7 L) \qquad \text{by } \epsilon_{stop} \le (\lambda_1\lambda_2)^3\kappa^{-74}/(2^{21}C) \text{ in Eq. (12)}$$

$$< \tau \qquad \text{by Eq. (8)}$$

Finally, Eq. (30) applies, since

$$4(r_1^2 + r_2^2)\epsilon_{stat}/\min(\lambda_1, \lambda_2)$$

$$= 4(r_1^2 + r_2^2)\epsilon_{stat}/\lambda_2$$

$$\le 32\kappa^2\epsilon_{stat}/\lambda_2 \qquad \text{by Eq. (66)}$$

$$\lesssim C(\lambda_1\lambda_2)^{-4}\kappa^{69}\epsilon_{stop} \qquad \text{by Eq. (68)}$$

$$\le 1/2 \qquad \text{since } \epsilon_{stop} \le (\lambda_1\lambda_2)^4\kappa^{-69}/(2C) \text{ by Eq. (12)}$$

Thus, all the preconditions to Lemma D.8 hold.

*Applying Lemma D.8.* So we conclude that if the event $E_{stat,t}$ for the $t$th call to TRAINNEURON holds, then we have $w^t = w^{t-1}$. The network weights are unchanged and neuron $v$ remains blank. Thus $E_{stepgood,t}$ follows from $E_{stepgood,t-1}$ in this case.

**Case 2b: Exactly two active inputs, product is useful**    The final case is if $\hat{g}(S) \neq 0$ and $S$ is not represented by some neuron at iteration $t-1$. Since $E_{stepgood,t-1}$ implies $E_{pol,t-1}$, we have

$$|\hat{\zeta}(S; w^{t-1}) - \hat{g}(S)| \leq \epsilon_{fourmove} t$$
$$\leq 1/(4M) \qquad\qquad \text{by Claim E.14}$$

So since $\hat{g}(S) \neq 0$ implies that $|\hat{g}(S)| \in [1/M, M]$ by assumption,

$$|\hat{\zeta}(S; w^{t-1})| \in [3/(4M), (5/4)M] \subset [1/(2M), 2M] \tag{75}$$

In this case, we will prove that with polynomially-lower bounded probability TRAINNEURON trains $v$ to approximately represent the monomial $\chi_S(x)$. And otherwise, with high probability it leaves the weights unchanged: $w^t = w^{t-1}$. To show this, we will use the guarantees for TRAINNEURON proved inLemmas D.11 to D.13. In order to apply these, we must first verify the preconditions.

*Checking preconditions of Lemmas D.11 to D.13.* By $E_{nothree,i-1}$, we know that $v$ cannot have two active parents on the previous layer. Therefore, we must have $u_2 \in V_{in}$ without loss of generality, so $\gamma_2 = \lambda_1$, $\epsilon_2 = 0$, and $|S_2| \leq 1$ because $v_2$ is an input. Now, if $u_1 \in V_{in}$ then the preconditions of the lemma with respect to the sets $S_1, S_2$ also hold because $u_1$ and $u_2$ are distinct inputs and therefore $S_2 \neq \emptyset$ without loss of generality and $S_1 \cap S_2 = \emptyset$.

On the other hand, suppose that $i > 1$ and $u_1 \in V_{i-1}$. Then $|S_1| \leq i$ by $E_{replayer,i-1}$. Since $\hat{g}(S) \neq 0$ and $E_{rep,S,t-1}$ does not hold, $S$ is not represented by a neuron in the first $i-1$ layers so by $E_{replayer,i-1}$ we conclude that $|S| > i$. Therefore since $|S_2| \leq 1$ and $S \neq S_1$ we have $|S_2| = 1$ and $|S_1| = i$ and $|S| = i + 1$. Thus, $f_{u_1}(x; w^{t-1})$ only depends on the variables in $S_1$. This is because by $E_{layergood,i-1}$, we must have $E_{nothree,0} \cap \dots E_{nothree,i-1}$, so the predecessors of neuron $u_1$ all have in-degree at most 2, and at least one of the parents is in $V_{in}$.

We conclude that in all cases $S_2 \neq \emptyset$, $\epsilon_2 = 0$, $S_1 \cap S_2 = \emptyset$, and $f_{u_1}(x; w^{t-1})$ depends only on variables $\{x_j\}_{j \in S_1}$ and $f_{u_2}(x; w^{t-1})$ depends only on variables $\{x_j\}_{j \in S_2}$. It only remains to verify Eqs. (36) to (47), which we do below. First, Eqs. (36) to (38) hold, since $\tau > \max(\epsilon^{(1)}, \epsilon^{(3)})$ and $\tau > |\hat{\zeta}(\emptyset; w^0)| + 2(|r_1|^2 + |r_2|^2)(\epsilon^{(3)})^2 + \epsilon_{stat}$ by the same reasoning as in Case 2a.

In the arguments below, let $C > 0$ be a sufficiently large universal constant. Eq. (39) holds, since

$$\sqrt{3} \max(\lambda_1, \lambda_2)/|r_1 r_2|$$
$$\leq \sqrt{3}\lambda_2/(1/(2M))^2 \qquad\qquad \text{by Eq. (66)}$$
$$\leq 32\lambda_2 M^2$$
$$\leq 1/(2M) \qquad\qquad \text{by } \lambda_2 \leq 1/(64M^3) \text{ in Eq. (10)}$$
$$\leq |\hat{\zeta}(S; w^0)| \qquad\qquad \text{by Eq. (75)}$$

Eq. (40) holds, since

$$\min(\lambda_1, \lambda_2)\sqrt{|\hat{\zeta}(S; w^0)|/|r_1 r_2|}/8$$
$$= \lambda_1 \sqrt{|\hat{\zeta}(S; w^0)|/|r_1 r_2|}/8$$
$$\geq \lambda_1/(8\sqrt{|r_1 r_2|2M}) \qquad\qquad \text{since } |\hat{\zeta}(S; w^0)| \geq 1/(2M) \text{ by Eq. (75)}$$
$$\geq \lambda_1/(8 \cdot (2M)^{3/2}) \qquad\qquad \text{since } |r_1|, |r_2| \leq 2M \text{ by Eq. (66)}$$
$$\geq C(\lambda_1 \lambda_2)^{-4} \kappa^{89} \epsilon_{stop} \qquad\qquad \text{since } \epsilon_{stop} \leq (\lambda_1 \lambda_2)^5 \kappa^{-91}/(32C) \text{ by Eq. (12)}$$
$$\geq \epsilon^{(2)}. \qquad\qquad \text{by Eq. (71)}$$

Eq. (41) holds because

$$
\frac{1}{8}\sqrt{\frac{\lambda_1}{\lambda_2}|\hat\zeta(S;w^0)|/|r_1 r_2|} \geq \frac{1}{32M^2}\sqrt{\frac{\lambda_1}{\lambda_2}|\hat\zeta(S;w^0)|} \qquad \text{since } |r_1|,|r_2| \leq 2M \text{ by Eq. (66)}
$$

$$
\geq \frac{1}{32\sqrt{2}M^{(3/2)}}\sqrt{\frac{\lambda_1}{\lambda_2}} \qquad \text{since } |\hat\zeta(S;w^0)| \geq 1/(2M) \text{ by Eq. (75)}
$$

$$
\geq \frac{1}{2048\sqrt{2}M^{(7/2)}L} \qquad \text{by Eq. (11)}
$$

$$
> \tau \qquad \text{by Eq. (71)},
$$

Eq. (42) holds because

$$
(|\hat\zeta(S;w^0)|/4)(\min_i |r_i|)/(\max_i |r_i|) - |\hat\zeta(\emptyset;w^0)| - \epsilon_{stat}
$$

$$
\geq 1/(8M)(\min_i |r_i|)/(\max_i |r_i|) - |\hat\zeta(\emptyset;w^0)| - \epsilon_{stat} \qquad \text{since } |\hat\zeta(S;w^0)| \geq 1/(2M) \text{ by Eq. (75)}
$$

$$
\geq 1/(32M^3) - |\hat\zeta(\emptyset;w^0)| - \epsilon_{stat} \qquad \text{by Eq. (66)}
$$

$$
\geq 1/(32M^3) - 2\epsilon_{stat} \qquad \text{by } E_{bias,t-1}
$$

$$
\geq 1/(32M^3) - C(\lambda_1\lambda_2)^{-3}\kappa^{67}\epsilon_{stop} \qquad \text{by Eq. (68)}
$$

$$
\geq 1/(64M^3) \qquad \text{by } \epsilon_{stop} \leq (\lambda_1\lambda_2)\kappa^{-70}/(64C) \text{ in Eq. (12)}
$$

$$
> \tau \qquad \text{by Eq. (8)}
$$

Eq. (43) holds because $\eta \geq 4\tau$ because $\eta = 4\tau$ by definition in Eq. (9).

Eqs. (44) and (45) hold because

$$
|2r_1 r_2 \tau^2| < |2r_1 r_2 \eta^2| \qquad \text{by Eq. (9)}
$$

$$
\leq 8M^2\eta^2 \qquad \text{by Eq. (66)}
$$

$$
\leq 1/(32M) \qquad \text{since } \eta = 4\tau \leq 1/(16M^2) \text{ by Eq. (9)}
$$

$$
\leq |\hat\zeta(S;w^0)|/16 \qquad \text{since } |\hat\zeta(S;w^0)| \geq 1/(2M) \text{ by Eq. (75)}
$$

In order to show Eq. (46), we first prove

$$
\epsilon_{stat} \leq C(\lambda_1\lambda_2)^{-3}\kappa^{67}\epsilon_{stop} \qquad \text{by Eq. (68)}
$$

$$
\leq 1/(2^{36}M^{14}L^2) \qquad \text{since } \epsilon_{stop} \leq (\lambda_1\lambda_2)^3\kappa^{-81}/(2^{36}C) \text{ by Eq. (12)}
$$

$$
= \eta^2 \qquad \text{by Eq. (9)}
$$

Therefore, Eq. (46) holds because

$$
|r_1 r_2 \eta^2 \hat\zeta(S;w^0)| \geq 1/(4M^2)\eta^2|\hat\zeta(S;w^0)| \qquad \text{by Eq. (66)}
$$

$$
\geq 1/(8M^3)\eta^2 \qquad \text{by Eq. (75)}
$$

$$
\geq 2^{12}M^4\eta^4 \qquad \text{by } \eta^2 \leq 2^{-15}M^{-7} \text{ in Eq. (9)}
$$

$$
\geq 2^{10}M^4(\eta^2 + \epsilon_{stat})^2 \qquad \text{by } \epsilon_{stat} < \eta^2 \text{ proved above}
$$

$$
\geq 16(4M^2\eta^2 + 4M^2\eta^2 + \epsilon_{stat})^2
$$

$$
\geq 16(r_1^2\eta^2 + r_2^2\eta^2 + \epsilon_{stat})^2 \qquad \text{by Eq. (66)}
$$

$$
\geq 16(r_1^2\eta^2 + r_2^2\eta^2 + |\hat\zeta(\emptyset;w^0)|)^2 \qquad \text{by } E_{bias,t-1}
$$

Finally, in order to show Eq. (47), we first prove the following two bounds:

$$
\frac{1}{32}|r_1 r_2 \eta^2 \hat\zeta(S;w^0)| \geq \frac{1}{256M^3}\eta^2 \qquad \text{by Eqs. (66) and (75)}
$$

$$
> 2\lambda_2 W\eta^2 \qquad \text{since } \lambda_2 W < 1/(512M^3) \text{ by Eq. (10)}
$$

and

$$\frac{1}{32}|r_1 r_2 \eta^2 \hat{\zeta}(S; w^0)|$$

$$\geq \frac{1}{256M^3}\eta^2 \qquad\qquad \text{by Eqs. (66) and (75)}$$

$$= 2^{-44}M^{-17}L^{-2} \qquad\qquad \text{by Eq. (9)}$$

$$\geq 2^{11}CM^4 L\epsilon_{stop} \qquad\qquad \text{by } \epsilon_{stop} < \kappa^{21}/(2^{55}C) \text{ in Eq. (12)}$$

$$\geq 2C((2M)^3 + 1)(45ML\epsilon_{stop})$$

$$\geq 2C((2M)^3 + 1)\epsilon_{rel,i-1} \qquad\qquad \text{by Claim E.13}$$

$$\geq 2C((2M)^3 + 1)(\max_{i\in\{1,2\}} \epsilon_i) \qquad\qquad \text{by } E_{rep,i-1}$$

$$> 2C\max_{u\in V}\max_{x\in\{-1,1\}^n}(|f_u(x; w^0)|^3 + 1)(\max_{i\in\{1,2\}} \epsilon_i) \qquad\qquad \text{by } E_{neurbound,t-1}$$

Eq. (47) holds by combining the above two bounds, since we take $C$ greater than or equal to the constant from Lemma D.13:

$$\frac{1}{32}|r_1 r_2 \eta^2 \hat{\zeta}(S; w^0)| > \lambda_2 W \eta^2 + C\max_{u\in V}\max_{x\in\{-1,1\}^n}(|f_u(x; w^0)|^3 + 1)(\max_{i\in\{1,2\}} \epsilon_i)$$

Therefore, the preconditions of Lemmas D.11 to D.13 all hold.

*Applying Lemmas D.11 to D.13.* Let $E_{newactive,t}$ be the event defined in Definition D.10 for the iteration $t$ call to TRAINNEURON. Lemma D.11 states that if $(\neg E_{newactive,t}) \cap E_{stat,t}$ holds, then $w^t = w^{t-1}$. In this case $E_{stepgood,t}$ follows from $E_{stepgood,t-1}$ because the parameters of the neural network are unchanged.

On the other hand, if $E_{newactive,t} \cap E_{stat,t}$ holds, then Lemma D.12 states that the weights $w_v$ corresponding to neuron $v$ are trained so that neuron $v$ becomes an active neuron. In particular, Item 1 of Lemma D.12 states that $f_v(x; w^t) = r\chi_S(x) + h(x)$, where

$$|r + \hat{\zeta}(S; w^{t-1})| \leq \frac{4\sqrt{\gamma_1\gamma_2}}{|r_1 r_2|} \leq 4M\lambda_2 \leq \epsilon_{learned},$$

and $h(x) \leq |r|\epsilon_{newrel}$ for

$$\epsilon_{newrel} = (4\epsilon_{stop} + 2|\hat{\zeta}(\emptyset; w^{t-1})|)/|\hat{\zeta}(S; w^{t-1})| + 32\frac{\lambda_2}{\lambda_1}|\epsilon_1|^2|r_1/r_2| + \epsilon_1(8\frac{|r_1|}{|r_2|}\sqrt{\frac{\gamma_2}{\gamma_1}} + 1).$$

Recall that $\gamma_2 = \lambda_1$ since we have assumed without loss of generality that $u_2 \in V_{in}$. If $u_1 \in V_{in}$ then we also have $\epsilon_1 = 0$ because $f_{u_1}(x)$ computes either the constant 1 or an input monomial in $x_1, \ldots, x_n$. Therefore,

$$\epsilon_{newrel} = (4\epsilon_{stop} + 2|\hat{\zeta}(\emptyset; w^{t-1})|)/|\hat{\zeta}(S; w^{t-1})| \leq 16M\epsilon_{stop} \leq \epsilon_{rel,i}$$

by $E_{bias,t-1}$ and Eq. (75). On the other hand, if $u_1 \notin V_{in}$ then we have $u_1 \in V_{i-1}$ because it must be in the previous layer, and the regularization is $\gamma_1 = \lambda_2$ because $u_1$ is not an input in $V_{in}$. Also, by $E_{replayer,i-1}$ we must have that the relative error $f_{u_1}(x)$ is $\epsilon_1 \leq \epsilon_{rel,i-1}$. So if $u_1 \notin V_{in}$ then

$$\epsilon_{newrel} \leq \frac{(4\epsilon_{stop} + 2|\hat{\zeta}(\emptyset; w^{t-1})|)}{|\hat{\zeta}(S; w^{t-1})|} + 32\frac{\lambda_2|r_1|}{\lambda_1|r_2|}|\epsilon_{rel,i-1}|^2 + \epsilon_1(8\frac{|r_1|}{|r_2|}\sqrt{\frac{\lambda_1}{\lambda_2}} + 1)$$

$$\leq 16M\epsilon_{stop} + 32\frac{\lambda_2}{\lambda_1}|\epsilon_{rel,i-1}|^2|r_1/r_2| + (8\frac{|r_1|}{|r_2|}\sqrt{\frac{\lambda_1}{\lambda_2}} + 1)\epsilon_{rel,i-1} \qquad \text{by } E_{bias,t-1} \text{ and Eq. (75)}$$

$$\leq 16M\epsilon_{stop} + 128M^2\frac{\lambda_2}{\lambda_1}|\epsilon_{rel,i-1}|^2 + (32M^2\sqrt{\frac{\lambda_1}{\lambda_2}} + 1)\epsilon_{rel,i-1} \qquad \text{by Eq. (66)}$$

$$= \epsilon_{rel,i}$$

In both cases $\epsilon_{newrel} \leq \epsilon_{rel,i}$, and so $E_{rep,S,t}$ holds because the network has been updated so that neuron $v$ now computes $\chi_S$ with at most $\epsilon_{rel,i}$ relative error.

Finally, Lemma D.12 allows us to prove that $E_{stepgood,t}$ holds. First, we show that $E_{pol,t}$ holds. The condition on $\hat{\zeta}(S; w^t)$ follows since $\hat{\zeta}(S; w^t) = \hat{\zeta}(S; w^{t-1}) + r$, and so we have $|\hat{\zeta}(S; w^t)| \leq \epsilon_{learned} \leq \epsilon_{fourmove}t + \epsilon_{learned}$. Furthermore, since $\neg E_{rep,S,(t-1)}$, by $E_{pol,t-1}$ we have $|\hat{\zeta}(S; w^{t-1}) + \hat{g}(S)| \leq \epsilon_{fourmove}(t-1)$, and so combining by triangle inequality with the bound on $|r + \hat{\zeta}(S; w^{t-1})|$, we have

$$|r - \hat{g}(S)| \leq \epsilon_{fourmove}t + \epsilon_{learned} \leq M/4$$

This means that $|r| \leq |\hat{g}(S)| + M/4 \leq 5M/4$. So $|h(x)| \leq |r|\epsilon_{rel,i} \leq (5M/4)(45ML\epsilon_{stop}) \leq 100ML^2\epsilon_{stop} = \epsilon_{fourmove}$ by Claim E.13. Since for any $S' \subset [n]$, we have $|\hat{\zeta}(S'; w^t) - \hat{\zeta}(S'; w^{t-1})| = |\hat{f}_v(S'; w^t)| = |\hat{h}(S')| = |\mathbb{E}_{x \sim \{-1,1\}^n}[h(x)\chi_{S'}(x)]| \leq \max_x |h(x)| \leq \epsilon_{fourmove}$. Therefore, $E_{pol,t}$ follows from $E_{pol,t-1}$ and this bound.

To prove that $E_{neurbound,t}$ holds, note that $|f_u(x; w^t)| = |f_u(x; w^{t-1})| \leq 2M$ for all $u \neq v$ by $E_{neurbound,t-1}$. And $|f_v(x; w^t)| \leq |r| + |h(x)| \leq (5/4M)(1 + 45ML\epsilon_{stop}) \leq 2M$ since $45ML\epsilon_{stop} \leq 1/2$ by Claim E.13.

$E_{parambound,t}$ holds because

$$\begin{aligned}
\max_{e \in E} |a_e^t| &= \max(\max_{e \in E} |a_e^{t-1}|, \max_{e=(u,v) \in E} |a_e^t|) \\
&\leq \max(16M^2\sqrt{\lambda_2/\lambda_1}, \max_{e=(u,v) \in E} |a_e^t|) \qquad \text{by } E_{parambound,t-1} \\
&\leq \max(16M^2\sqrt{\lambda_2/\lambda_1}, 4\sqrt{\frac{\lambda_2}{\lambda_1}|\hat{\zeta}(S; w^0)|/|r_1 r_2|}) \qquad \text{by Item 2 of Lemma D.12} \\
&\leq 16M^2\sqrt{\lambda_2/\lambda_1} \qquad \text{by Eqs. (66) and (75)}
\end{aligned}$$

$E_{bias,t}$ holds by Item 3 of Lemma D.12. And $E_{nobadactive,t}$ holds because the active neuron that has been created represents $S$, where $\hat{g}(S) \neq 0$ and $|S| = i + 1$.

Thus, in this case $E_{stepgood,t} = E_{stepgood,t-1} \cap E_{pol,t} \cap E_{neurbound,t} E_{parambound,t} \cap E_{bias,t}$ holds. Therefore, our analysis shows that if $\hat{g}(S) \neq 0$ and $(\neg E_{rep,S,t-1}) \cap E_{stepgood,t-1} \cap E_{layergood,i-1}$ holds, then

$$\begin{aligned}
\mathbb{P}[E_{rep,S,t} \mid w^{t-1}] &\geq \mathbb{P}[E_{newactive,t} \cap E_{stat,t} \mid w^{t-1}] \\
&\geq \min(1, \sqrt{|r_1 r_2 \hat{\zeta}(S; w^{t-1})|}/8) - \mathbb{P}[\neg E_{stat,t} \mid w^{t-1}] \qquad \text{by Lemma D.13} \\
&\geq \min(1, \sqrt{|\hat{\zeta}(S; w^{t-1})|}/(16M)) - \mathbb{P}[\neg E_{stat,t} \mid w^{t-1}] \qquad \text{by Eq. (66)} \\
&\geq \min(1, 1/(32M^2)) - \mathbb{P}[\neg E_{stat,t} \mid w^{t-1}] \qquad \text{by Eq. (75)} \\
&\geq 1/(32M^2) - \mathbb{P}[\neg E_{stat,t} \mid w^{t-1}] \\
&\geq 1/(64M^2),
\end{aligned}$$

since $\mathbb{P}[\neg E_{stat,t} \mid w^{t-1}] \leq \delta_{stat} \leq 1/(64M^2)$ by Lemma E.19. $\qquad \square$

### E.3.2 $E_{replayer,i} \cap E_{stepgood,t_i}$ follows from $E_{stepgood,t_{i-1}} \cap E_{layergood,i-1}$ with high probability

Another ingredient in the induction is showing that the updates from iterations $t_{i-1} + 1$ through $t_i$ suffice for $E_{replayer,i}$ to hold with high probability. Essentially, if the degree at most $i$ monomials were represented after training layers 1 through $i - 1$, then with high probability the degree at most $i + 1$ monomials are represented after training layers 1 through $i$.

**Lemma E.22.** $\mathbb{P}[E_{replayer,i} \cap E_{stepgood,t_i} \mid E_{stepgood,t_{i-1}} \cap E_{layergood,i-1}] \geq 1 - Ws\delta_{stat} - \delta/(8L)$

*Proof.* $E_{stepgood,t_i}$ implies $E_{nobadactive,t_i}$. Therefore it remains to show that for any $S \subset [n]$ with $|S| \leq i + 1$ and $\hat{g}(S) \neq 0$ that $E_{rep,S,t_i}$ holds with high probability.

Suppose that $i > 1$, then for any $S$ with $|S| \leq i$, we have that $E_{rep,S,t_i}$ holds by the inductive hypothesis $E_{layergood,i-1}$. Therefore, it remains to prove $E_{rep,S,t_i}$ holds with high probability for any $S \subset [n]$ such that $|S| = i + 1$ and $\hat{g}(S) \neq 0$.

Fix such a subset $S$ with $|S| = i + 1$ and $\hat{g}(S) \neq 0$. Since $g$ satisfies the staircase property in Definition 1.1, there must be a set $S_1 \subset S$ such that $\hat{g}(S') \neq 0$ and $|S \setminus S_1| = 1$. By the event $E_{replayer,i-1}$, which is implied by $E_{layergood,i-1}$, there is a neuron $u_1 \in V_{i-1}$ such that $u_1$ represents $S_1$. On the other hand, letting $S_2 = S \setminus S_1$, because $|S_2| = 1$ there is a neuron $u_2 \in V_{\text{in}}$ such that $u_2$ represents $S_2$. Therefore, by $E_{conn,i-1}$, it holds that $|\{v \in V_i : (u_1, v), (u_2, v) \in E\}| \geq n_{shared}$.

Let $t^{(1)} \leq \cdots \leq t^{(k)}$ be the iterations such that the neuron $v \in V_i$ trained at iteration $t^{(j)}$ satisfies $(u_1, v), (u_2, v) \in E\}$. By the above argument, $k \geq n_{shared}$.

For any $t \in (t_{i-1}, t_i]$ if $w^{t-1}$ is such that $(\neg E_{rep,S,t-1}) \cap E_{stepgood,t-1} \cap E_{layergood,i-1}$ holds, then by Lemma E.20 we have that $E_{stepgood,t} \cap E_{layergood,i-1}$ holds with probability at least $1 - \delta_{stat}$. In addition, if $t \in \{t^{(1)}, \ldots, t^{(k)}\}$, then $E_{rep,S,t} \cap E_{stepgood,t} \cap E_{layergood,i-1}$ holds with probability at least $1/(64M^2)$. Since once $E_{rep,S,t}$ holds, it is also true that $E_{rep,S,t'}$ holds for all $t' \geq t$, analyzing the Markov chain implies

$$\mathbb{P}[E_{rep,S,t_i} \cap E_{stepgood,t_i} \mid E_{stepgood,t_{i-1}} \cap E_{layergood,i-1}] \geq 1 - W\delta_{stat} - (1 - 1/(64M^2))^k$$
$$\geq 1 - W\delta_{stat} - \delta/(8sL),$$

since $k \geq n_{shared} \geq 64M^2 \log(16sL/\delta)$ by Eq. (63).

By a union bound over all $S$ such that $|S| = i+1$ and $\hat{g}(S) \neq 0$, we have $\mathbb{P}[E_{replayer,i} \cap E_{stepgood,t_i} \mid E_{stepgood,t_{i-1}} \cap E_{stepgood,i-1}] \geq 1 - Ws\delta_{stat} - \delta/(8L)$.

The case where $i = 1$ is similar: here it suffices to show that $E_{rep,S,t_1}$ holds with high probability for any $S \subset [n]$ such that $|S| \leq 2$ and $\hat{g}(S) \neq 0$. An analogous argument to the above works, appealing to $E_{conn,0}$ and the fact that for each $S' \subset [n]$ with $|S'| \leq 1$, there is a neuron $u \in V_{\text{in}}$ computing $\chi_S$. $\square$

### E.3.3 $E_{conn,i} \cap E_{nothree,i}$ follows from $E_{replayer,i}$ with high probability

The final element of the inductive step is to guarantee that the network connectivity events for the edges to layer $i$ after the training of layers 1 through $i - 1$ has concluded. The idea behind proof here is that the edges to layer $i$ are independent of the state of the network parameters at iteration $t_{i-1}$, and since $E_{replayer,i-1}$ guarantees that there are at most $s$ active neurons at iteration $t_{i-1}$ we may ensure these events hold with high probability.

**Lemma E.23.** *For any $i \in \{0, \ldots, L - 1\}$, conditioned on $E_{replayer,i}$ and $w^{t_i}$, the event $E_{conn,i}$ holds with probability at least $1 - \delta/(8L)$.*

*Proof.* First we consider shared children of pairs of inputs in $V_{\text{in}}$. By Eqs. (5) and (6),

$$(p_1)^2 W \geq 10 \log(4WL/\delta)n_{shared}$$
$$\geq 10 \log(4(n + 1)L/\delta)n_{shared},$$

for any distinct $u, u' \in V_{\text{in}}$ we have

$$\mathbb{P}[|\{v_1 \in V_1 : (u, v_1), (u', v_1) \in E\}| \geq n_{shared}] \geq 1 - \delta/(4(n + 1)L)^2$$

by a Hoeffding bound, as all edges from $V_{\text{in}}$ to $V_{i+1}$ are i.i.d. with probability $p_1$ and independent of $w^{t_i}$. Therefore, by a union bound, for all pairs of distinct $u, u' \in V_{\text{in}}$, with probability at least $1 - \delta/(16L)$ we have that $|\{v_1 \in V_1 : (u, v_1), (u', v_1)\}| \geq n_{shared}$.

Now, we consider the number of children of an input and a neuron on the previous layer. For $i \geq 1$, note that $E_{replayer,i}$ implies that the number of active neurons in $V_i$ after iteration $t_i$ must be at most $s$ – because each active neuron corresponds to a unique nonzero Fourier coefficient of $g$. Furthermore, these active neurons are trained independently of the edges from layer $i$ to layer $i + 1$. Hence for any active neuron $v_i \in V_i$ that is active at iteration $t_i$ and any input $u \in V_{\text{in}}$, the expected number of neurons $v_{i+1} \in V_{i+1}$ that have $v_i$ and $u$ as parents is

$$p_1 p_2 W \geq 10 \log(4WL/\delta)n_{shared},$$

by Eqs. (5) to (7). So by a Hoeffding bound,

$$\mathbb{P}[|\{v_{i+1} \in V_{i+1} : (v_i, v_{i+1}) \in E, (u, v_{i+1}) \in E\}| \geq n_{shared}] \geq 1 - \delta/(4WL)^2$$
$$\geq 1 - \delta/(16W(n + 1)L).$$

Finally, a union bound over the at most $W(n+1)$ pairs of an input $u \in V_{\text{in}}$ and a neuron on layer $V_i$ imply that with probability at least $1 - \delta/(16L)$ for all such pairs $|\{v_{i+1} \in V_{i+1} : (v_i, v_{i+1}) \in E, (u, v_{i+1}) \in E\}| \geq n_{shared}$.

The lemma follows by a union bound of the above two results. □

**Lemma E.24.** *For any* $i \in \{0, \ldots, L-1\}$*, conditioned on* $E_{replayer,i}$ *and on* $w^{t_i}$*, the event* $E_{nothree,i}$ *holds with probability at least* $1 - \delta/(8L)$*.*

*Proof.* By $E_{replayer,i}$, there are at most $s$ active neurons in $V_i$, because each one corresponds to a distinct nonzero Fourier coefficient of $g$. For any $v \in V_{i+1}$ and distinct $u_1, u_2, u_3 \in V_{\text{in}} \cup \{u \in V_i : u\text{is active at iteration } t_i\}$, call the tuple $(v, u_1, u_2, u_3)$ "bad" if $(u_1, v), (u_2, v), (u_3, v) \in E$. The probability that a tuple $(v, u_1, u_2, u_3)$ is bad is at most $\max(p_1, p_2)^3 = (p_1)^3$, since these edges are independent of $w^{t_i}$, because by the layerwise training $w^{t_i}$ depends only on presence or absence the edges up to the layer $V_i$. The number of bad tuples is thus at most

$$(p_1)^3 W(s+n+1)^3 \leq$$
$$\leq \delta/(16L)$$

in expectation by Eqs. (5) and (6). Therefore, by a Markov bound there are no bad tuples with probability at least $1 - \delta/(16L)$.

Furthermore, the number of neurons $v \in V_{i+1}$ that have at least two active $u_1, u_2 \in V_i$ is in expectation at most

$$(sp_2)^2 W \leq \qquad\qquad \leq \delta/(16L)$$

, by a similar argument and Eqs. (5) and (7). So a Markov bound shows there are no such neurons with probability at least $1 - \delta/(16L)$.

So by a union bound $E_{nothree,i}$ holds with probability at least $1 - \delta/(8L)$. □

### E.3.4 Proof of Theorem B.1

We conclude by combining the inductive steps Lemmas E.20 and E.22 to prove that $E_{replayer,L}$ holds with high probability, and then recalling this is sufficient by Lemma E.15.

**Lemma E.25.** $E_{replayer,L}$ *holds with probability at least* $1 - \delta$*.*

*Proof.* We prove by induction on $i$ that for any $i \in \{0, \ldots, L-1\}$,

$$\mathbb{P}[E_{layergood,i} \cap E_{stepgood,t_i}] \geq 1 - (2i+1)\delta/(4L).$$

For the base case $i = 0$, note that $E_{bias,0}$ holds because $\hat{\zeta}(\emptyset; w^0) = -\hat{g}(\emptyset) = 0$. Further, $E_{neurbound,0}$, $E_{parambound,0}$, and $E_{nobadactive,0}$ follow from the fact that the network is initialized to all zeros, and $E_{pol,0}$ holds because $\hat{\zeta}(S; W^0) = -\hat{g}(S)$ for all $S \subset [n]$. $E_{replayer,0}$ holds because by definition it always holds. Finally, given $E_{replayer,0}$, Lemmas E.23 and E.24 imply that $E_{conn,0} \cap E_{nothree,0}$ hold with probability at least $1 - \delta/(4L)$. Combining these with the definition of $E_{layergood,0}$ and $E_{stepgood,0}$ in Definitions E.16 and E.17, it follows that

$$\mathbb{P}[E_{layergood,0} \cap E_{stepgood,t_0}] \geq 1 - \delta/(4L).$$

For the inductive step for any $i > 1$, Lemma E.22 implies that

$$\mathbb{P}[E_{replayer,i} \cap E_{stepgood,t_i} \mid E_{layergood,i-1} \cap E_{stepgood,t_{i-1}}] \geq 1 - Ws\delta_{stat} - \delta/(8L) \geq 1 - \delta/(4L).$$

Also, Lemmas E.23 and E.24 imply that

$$\mathbb{P}[E_{conn,i} \cap E_{nothree,i} \mid E_{replayer,i} \cap E_{stepgood,t_i} \cap E_{layergood,i-1}] \geq 1 - \delta/(4L).$$

Since

$$E_{layergood,i} = E_{layergood,i-1} \cap E_{replayer,i} \cap E_{conn,i} \cap E_{nothree,i},$$

we conclude that

$$\mathbb{P}[E_{layergood,i} \cap E_{stepgood,t_i}] \geq \mathbb{P}[E_{layergood,i-1} \cap E_{stepgood,t_{i-1}}] - \delta/(2L) \geq (2i+1)\delta/(4L).$$

This concludes the induction.

Applying the claim with $i = L - 1$, we obtain

$$\mathbb{P}[E_{layergood,L-1} \cap E_{stepgood,t_{L-1}}] \geq 1 - (2L-1)\delta/(4L),$$

and so by one final application of Lemma E.22 we have $\mathbb{P}[E_{replayer,L} \cap E_{stepgood,t_L}] \geq 1 - (2L)\delta/2 \geq 1 - \delta/2$. □

*Proof of Theorem B.1.* By Lemma E.15 that if $E_{replayer,L}$ holds then $\ell(w^{t_L}) < \epsilon$. Further, Lemma E.25 proves that $\mathbb{P}[E_{replayer,L}] \geq 1 - \delta$. The runtime bound follows because there are $t_L = WL = O(\kappa)$ iterations, each of which can be implemented in $O(\kappa^{2393})$ time and samples by Lemma D.5. □