# OpenReview forum: "The staircase property: How hierarchical structure can guide deep learning"
_NeurIPS.cc/2021/Conference — NeurIPS 2021 Poster_

### Official Review · Reviewer_dLyL · 2021-07-14

**Rating:** 7
**Confidence:** 4

**Summary:**

This paper identifies a class of functions with hierarchical structures which can be efficiently learned by a neural network trained layerwise with stochastic coordinate descent. The authors introduce a sparsely connected neural network architecture and provide a corresponding layerwise training algorithm. Theoretically, they prove that for a class of hierarchical Boolean functions, the population loss can be minimized to be close to zero to illustrate the generalization ability.

**Limitations And Societal Impact:**

Yes.

**Main Review:**

Strength:

The motivation to study the functions with hierarchical structures which can be learned by neural networks is clearly described and the literature reviews are exhaustive. The generalization results for the hierarchical Boolean functions based on the population loss are quite strong. Numerical experiments on ReLU ResNets also support their claim.

Weakness and comments:

The definition of the homogeneous neural network is quite different from the definition in [a]. It will be helpful to discuss the similarity and differences somewhere in the paper. Besides, it seems that the homogeneous neural network discussed in the paper is firstly studied by the reference [3]. However, the preprint [3] does not appear on its first author’s  website. It will be helpful to provide other easy-to-access references.

[a] Lyu and Li, Gradient Descent Maximizes the Margin of Homogeneous Neural Networks.

The organization of the paper can be improved. For instance, the general hierarchical architecture described in section 2 is rarely discussed in the rest of the paper. The authors may move this part into the appendix or discuss how the proposed neural network architecture, training algorithm, and generalization results can modify to learn functions with the general hierarchical architecture.

The algorithm description is not very clear. Specifically, the authors only tell that each of the subroutine TrainNeuron runs SGD. They shall specify how many samples are drawn in each iteration and how many iterations SGD will be executed in the subroutine. Besides, it will be better to discuss the overall number of required samples in the training instead of directly stating the sample access to the data distribution. This will be helpful to generalize the results to learning general functions beyond those with the domain of {-1,1}^n.

The numerical comparison part is a bit confusing. The authors claim that they use the same ReLU ResNet. However, the architecture introduced in section 3.1 is not directly related to the ResNet architecture. The authors shall explain the relation between ResNet and the architecture in section 3.1.

Besides, the authors also say that they train the neural network using SGD instead of using their proposed layerwise training introduced in section 3.3. It will be helpful to provide the comparisons between these two training methods. The width of ResNet in the experiment is only 40. The authors may try testing on wider ResNet, for instance, 400.

Minors:

For the equation above line 165, y is not defined. I guess that it shall be defined by y=Ax+b.

In line 172, the norm $\mathcal{S}$ is only defined via the pseudo metric $d_{\mathcal{S}}$. It will be helpful to properly define $\mathcal{S}$.


**Time Spent Reviewing:**

5

---

> ### Author Response · Authors · 2021-08-10
> **Response to the reviewer**
>
> Thank you for the kind and thoughtful comments which have helped improve the paper. We respond to your questions below.
>
> > “The definition of the homogeneous neural network is quite different from the definition in [a]. It will be helpful to discuss the similarity and differences somewhere in the paper.”
>
> The definition of a homogeneous neural network in [a] is unrelated to our definition of homogeneous. In that work, the definition is that if you scale the weights by $c$ then the output gets scaled by $c$ to some power. In contrast, in our work, “homogeneous” means that the initialization is i.i.d. and that the architecture is such that the edges between a pair of layers are included independently and with equal probability. We chose the word “homogeneous” because it is the same word used to describe Erdos-Renyi random graphs where each edge is included with equal probability. However, in order to avoid confusion, we will change the terminology to “regular” in the revision. We wonder if you have suggestions on terminology that could make sense.
>
> > “Besides, it seems that the homogeneous neural network discussed in the paper is firstly studied by the reference [3]. However, the preprint [3] does not appear on its first author’s website.”
>
> Reference [3] has now been submitted on arXiv and is available at https://arxiv.org/pdf/2108.04190.pdf. However, there is no definition of “homogeneous” networks in that paper. Instead, what we claim is that the results in [3] (and also the result of [6], which is available online) require initializations that are very specific to the problem at hand, and that it is unknown how to obtain these results in the case of homogeneous initializations such as the one studied in the present paper. We will make these statements clearer in the revision.
>
> > “The organization of the paper can be improved. For instance, the general hierarchical architecture described in section 2 is rarely discussed in the rest of the paper. The authors may move this part into the appendix or discuss how the proposed neural network architecture, training algorithm, and generalization results can modify to learn functions with the general hierarchical architecture.”
>
>
> Thank you for the helpful suggestions. In the revision, we will add more motivation for the general hierarchical structure in Section 2 -- namely, that it is a natural extension beyond the Boolean case and that it encompasses function classes of interest in classical learning theory, including decision trees in a smoothed complexity setting, and sparse biased parities ([Ref 3]). We refer the reviewer to points 1 and 2 in the general comments for detailed discussions on why these function classes have important implications.
> To tie this more general hierarchical structure into the theoretical results of the paper, we will add a section discussing how the architecture could be slightly modified so that we could extend our result to this more general class of functions. Please see our response to all reviewers for a more detailed discussion of the changes we will make in our revision.
>
> > “The algorithm description is not very clear”
>
> Thank you for the suggestions about the exposition of the algorithm, which we will implement in the revision.
>
> > “The numerical comparison part is a bit confusing. The authors claim that they use the same ReLU ResNet. However, the architecture introduced in section 3.1 is not directly related to the ResNet architecture. The authors shall explain the relation between ResNet and the architecture in section 3.1.
> Besides, the authors also say that they train the neural network using SGD instead of using their proposed layerwise training introduced in section 3.3. It will be helpful to provide the comparisons between these two training methods. The width of ResNet in the experiment is only 40. The authors may try testing on wider ResNet, for instance, 400.”
>
> First of all we apologize if there was confusion between the theory and empirical parts; The paper has two contributions that are both grounded on the same concept of learning functions with the staircase property. For the empirical work, we considered what we think is a very standard class of ResNET without modification, and also used the most common form of SGD for training (to the best of our knowledge, we could add more simulations if relevant). For the theoretical work, we prove the results about functions with the staircase property not for such standard ResNET with SGD,  but for a homogeneous/regular net with layerwise SGD, simply because this framework is simpler to analyze rigorously, even though the current proof is already hairy. Our goal was to establish the first formal result that shows that “regular nets with gradient-based training can learn staircases,” but of course it would be great to be able to prove the same result for the standard ResNET with SGD. We believe that this is in fact one of the nice future directions of work that our submission puts forward. We will make this discussion more explicit in the paper; note for the record that we mentioned this in lines 281-288 in the Limitations section.
>
> > “For the equation above line 165, $y$ is not defined. I guess that it shall be defined by $y=Ax+b$. In line 172, the norm $\mathcal{S}$ is only defined via the pseudo metric $d_\mathcal{S}$. It will be helpful to properly define $\mathcal{S}$.”
>
> Thank you for this note. You are correct about $y$. We will fix it in the revision. We will clarify the notation regarding the seminorm $\mathcal{S}$.
>
> References
>
> [Ref 3] Malach, Kamath, Abbe and Srebro - “Quantifying the Benefit of Using Differentiable Learning over Tangent Kernels”, ICML 2021.

---

> > ### Comment · Reviewer_dLyL · 2021-08-20
> > **Response to the authors**
> >
> > The authors address most of my questions. I would like to raise the score to 7.

---

### Official Review · Reviewer_1A1u · 2021-07-17

**Rating:** 7
**Confidence:** 1

**Summary:**

The authors prove a theoretical result on the ability of deep neural networks to learn hierarchical boolean functions. Intuitively, the authors argue that neural networks are able to learn functions that contain low-level features that can be hierarchically combined into high-level features (aka staircase property).

**Limitations And Societal Impact:**

The authors discussed relevant limitations of their work.

**Main Review:**

**Disclaimer**: I am not an expert in the are of this paper (boolean functions and deep learning theory). Consequently, I had a hard time reviewing this paper. I try to make useful comments about the paper, but assign a low confidence score to my review.

## Overview

The paper presents an interesting view on generalization in deep learning. In particular, I have not previously seen a similar analysis, even the results in Figure 2. I have seen arguments that neural networks learn functions of increasing complexity, as shown in Figure 2(d), but not in the application to boolean functions; the result in Fig. 2 (a), (c) are interesting: the network is unable to learn the function that lacks hierarchical structure.

My understanding is that the theory explains why NNs *can* learn the functions that have a hierarchical structure. Would the theory also predict that the network *cannot* learn a function that lacks hierarchical structure?

## Limitations

The main limitations of the work are probably that the model architecture (a sparsely and randomly connected network) and the training method (layer-wise optimization) are quite different from what is used in practice.

## Clarity

The paper is quite technical, and not easy to follow for an outsider to the field. Even understanding the definition 1.1 was not trivial for me. Specifically, what are $M_1$, $M_2$? Are these just real-valued numbers? What is the intuitive meaning of $\hat g(S)$ being equal to zero? Overall, I would encourage the authors to provide more intuition behind definition 1.1 and other formal definitions in the paper.

**Time Spent Reviewing:**

3

---

> ### Author Response · Authors · 2021-08-10
> **Response to the reviewer**
>
> Thank you for the kind and thoughtful comments which have helped improve the paper.
>
> ### Re: clarifications
>
> **Clarification questions** We will make the paper more friendly to the non-expert reader by addressing your clarification questions in the revision:
> $M_1$ and $M_2$ are positive real-valued numbers that upper bound the magnitude of the Fourier coefficients and bound the nonzero Fourier coefficients away from zero, respectively. Any PAC-learning algorithm for functions with the staircase property of Definition 1.1 must depend polynomially on $\frac{M_2 }{ M_1}$. We will clarify this in the revision.
>
> If $\hat{g}(S)$ is zero, then this means that $g(x)$ is uncorrelated with the parity $\chi_S(x) = \prod_{i \in S} x_i$. In this sense, $\chi_S(x)$ is an uninformative feature for learning g.
>
> **Extra justification** We will also add more explanation and motivation for the hierarchical structure in Definition 1.1 and Section 2 considered in this paper. This will clarify its significance. This extra motivation includes an explanation of how hierarchical functions in the sense of Section 2 include $\log(n)$-depth decision trees in a smoothed complexity setting and sparse biased parities. Thus, an algorithm for learning hierarchical functions in the sense of Section 2 would also be able to learn decision trees in the smoothed complexity setting. We refer the reviewer to the general comments for a further discussion.
>
> ### Re: limitations of the model and training algorithm
>
> We agree with the limitations, which we have stated clearly in the manuscript, and we believe it is a very interesting open problem to overcome. We refer the reviewer to the general comments for more discussion.
>
> ### Re: necessary condition
>
> While we show that hierarchical structure is sufficient for learnability, there are functions like the indicator $\chi_S(x) = \prod_{i \in S} x_i$ , where $|S| = \log n$ which cannot be learned with a polynomial number of statistical queries (Corollary 14 in [5]). This lacks a hierarchical structure and it cannot be learned efficiently. We also refer the reviewer to point 3 in the general comments for our discussion on necessity of hierarchical structure.

---

> > ### Comment · Reviewer_1A1u · 2021-08-23
> > **Thank you for the response!**
> >
> > Thank you for the response and clarifications! I think including these in the paper would really help non-expert readers understand the contributions and their significance. I am happy to raise my score to 7, as I couldn't identify any significant issues in the paper beyond the limiting assumptions, but my confidence is still low, as I am not an expert in the area.

---

### Official Review · Reviewer_n6Y2 · 2021-07-17

**Rating:** 7
**Confidence:** 3

**Summary:**

The paper discussed the learnability of function by neural networks.

The paper introduces a "staircase function": a boolean function composed of monomials of increasing order such that higher order monomials are recursively composed of lower-order monomials. The authors argue that this is a prototype of a hierarchical feature structure present in natural data, and they show by experiment that this function, unlike a parity function, is efficiently learnable by standard ReLU MLPs.

The paper extends the formally the staircase function to a class of "Hierarchical Boolean Function" which it further extends to continous functions. As its main theoretical result, the paper introduces a somewhat non-standard neural network architecture and proves that it can efficiently learn hierarchical boolean functions under sparsity conditions, using a non-standard neuron-wise training algorithm.


**Limitations And Societal Impact:**

I'd like more discussion of how the paper claims reconcile with the observed non-hierarchical behavior.

**Main Review:**

The paper provides an interesting learnability result which attempts to further our understanding on why neural networks seem to learn natural data even though some simple synthetic functions such as parity functions are hard to learn.

The authors argue that natural data is better understood as having "staircase" hierarchy, where more complex factors are composed of simpler factors enabling a neural network to learn them in sequence starting from a linear-like model and increasingly adding non-linear factors. This claim is not proven, as it would hard to, but empirical evidence is presented in the form of the interpretable hierarchy of features that is sometimes observed in the layers of trained neural networks. The authors mention the NTK regime where neural networks behave non-hierarchically, however they fail to mention practical cases where non-hierarchical behavior is observed (e.g. training example memorization even with completely noisy labels).

The theoretical proof has the benefit of considering a homogeneous architecture and initialization, unlike previous work, however the architecture and traning algorithm are still non-standard, so they don't necessarily transfer to practical neural networks.

Overall I consider the findings of this paper interesting although not groundbreaking.

EDIT:

I'm satisfied with the authors response, I've increased my score.

**Time Spent Reviewing:**

2

---

> ### Author Response · Authors · 2021-08-10
> **Response to the reviewer**
>
> Thank you for the kind and thoughtful comments which have helped improve the paper.
>
>
> Regarding non-hierarchical learning in the NTK regime:
>
> In the revision, we will add more discussion of the relation of our work to the NTK regime and non-hierarchical learning.  As the reviewer notes, non-hierarchical learning is an interesting phenomenon that is often observed in practice. And in the highly overparameterized regime, the NTK approximation is valid and training the network may be understood as learning with a kernel method that merely interpolates all the data points and memorizes their labels in a non-hierarchical way. As the reviewer points out, overparameterized networks may even memorize random labels over data points.
>
> In contrast, in our setting the sample space is of the size $2^n$, so there are actually fewer parameters in the network than the number of possible data samples -- this means that naive memorization of the data samples would require an exponential number of samples and neurons. Thus, in this work, we are mainly concerned with generalization over unseen data. In an effort to study what is tractably learnable we investigate what kind of structures present in the data can be broken down and efficiently learned by neural networks.
>
>
> Indeed, the class of staircase functions that we consider is provably not learnable by kernel methods or neural networks in the NTK regime. We will add discussion of this fact in the revision. Namely, the general staircase structure in Section 2 contains the class of  $\log(n)$-depth decision trees over the biased hypercube [implied by Lemma 3 of Ref 1], and as a special case contains sparse parities over the biased hypercube. It was shown in [Ref 3] that sparse-biased-parities cannot be learned (to any nontrivial loss) by any kernel method with polynomially many features. Therefore, NTK/non-hierarchical learning cannot learn the functions satisfying the staircase property in Section 2, and the phenomenon that we study is entirely non-hierarchical.
>
> References:
>
> [Ref 1]  Kalai, Samorodnitsky, and Teng - “Learning and Smoothed Analysis”, FOCS 2009.
>
> [Ref 3] Malach, Kamath, Abbe and Srebro - “Quantifying the Benefit of Using Differentiable Learning over Tangent Kernels”, ICML 2021.

---

### Official Review · Reviewer_JJ8b · 2021-07-24

**Rating:** 6
**Confidence:** 3

**Summary:**

In order to formalize the intuition that neural nets learn by building hierarchical features, this paper considers a model of learning Boolean functions shoe spectrum satisfies a certain  "staircase" structure. Roughly speaking, for every non-zero coefficient in the support, there is a subset of it that also has a non-zero coefficient. This paper shows that a certain family of neural networks will successfully learn such functions. In contrast, earning parity functions along is hard.

**Limitations And Societal Impact:**

Yes

**Main Review:**

The strength of the paper is that it comes up with a nice clean abstraction of hierarchical structure in functions, and proves an elegant theoretical result, which is borne out by practical experiments.

One can ask whether this structure really models the more complex kind of structures you would find in images etc. It is hard to argue too strongly here and the paper does not try (which is fine, this simpler setting means you can actually give formal guarantees).

However, one could still ask if the class of Boolean functions that they consider is a rich and natural class. How does it relate to other known classes of Boolean functions (halfspace, dnfs, decision trees, monotone functions etc). Is it known to include any class of functions that has been studied previously in the literature? I would have like to see the paper make a stronger case for this.  I don't think structure in the Fourier space lone qualifies a class  as natural.

There are some other choices that struck me as a little non-standard, including the choice of squared activation functions and the architecture. The authors do discuss these under limitations.

Overall, I liked reading this paper. I thing it makes a nice contribution towards a rigorous analysis of how hierarchical concept learning happens. I will be wiling to support it more strongly if the case can be made that this is a natural and interesting class of functions.

 In general the writing is good, except there seems to be a typo/omission at a rather crucial place. If you look at line 164, the affine transform $A$ is not used anywhere in the definition. I suspect it was intended to be used in defining the sequence $h_k$.

**Time Spent Reviewing:**

2.5

---

> ### Author Response · Authors · 2021-08-10
> **Response to the reviewer**
>
> We thank the reviewer for the kind and thoughtful comments which have helped improve the paper.
>
> The reviewer asks whether the class of functions studied in this paper is significant in the sense that it is expressive, and includes classes that have been previously studied in the literature. The answer to this question is yes, as we discuss in our general comments. We believe that the reviewer will especially appreciate points 1 and 2 in the general comments where we discuss applications to learning decision trees in a smoothed complexity setting, and sparse biased parities. We will add these in our revision.
>
> We agree with the fact that the architecture is non-standard, which we have stated clearly in the manuscript, and we believe it is a very interesting open problem. We refer the reviewer to the general comments for more discussion.

---

> > ### Comment · Reviewer_JJ8b · 2021-08-24
> > **Question about the role of $A$.**
> >
> > Can you clarify the role $A$ is meant to play in the definition?

---

> > > ### Author Response · Authors · 2021-08-25
> > > **The role of A**
> > >
> > > A is the matrix in the affine transform. We let y = Ax + b, where x is the input and the hierarchical structure is defined with respect to y (In the display in page 5, the LHS depends on x and the RHS depends on y). We left out the definition of y by mistake and will include it in the revision.

---

### Author Response · Authors · 2021-08-10
**General response to all reviewers**

We thank all of the reviewers for the kind and thoughtful comments which have greatly helped improve the paper. We address some of the reviewers’ common questions here.

### Richness of the class of functions

This paper puts forward a structural property of functions -- the staircase property -- which we believe plays an important role in understanding the class of functions that can (or cannot) be learned from regular neural networks with descent algorithms. The main theoretical result of the paper focuses on the staircase functions on the unbiased binary hypercube, i.e., Definition 1.1. In our paper, we are able to prove a first result showing that “a staircase function” is learned from “a regular network” using “a gradient-based” training algorithm.

Furthermore, additional experiments provided in the appendix show that this phenomenon is more general than just unbiased distributions. Thus another contribution of the paper is to put forward several natural extensions that result from the simplest unbiased case, namely the general staircase structure defined in Section 2. This opens up several interesting extensions to pursue, because the class of functions satisfying the general staircase structure is rich:

**1. Decision trees**: In a smoothed complexity setting (i.e., biased input distribution), the class of $\log(n)$-depth decision trees satisfy the general staircase structure [implied by Lemma 3 in Ref 1]. Since this is the staircase property over a biased input distribution, it does not quite fall under our main theoretical result which holds only for unbiased inputs. In the revision we will add a discussion for how our theoretical result could be modified to show that a slightly less natural architecture learns staircase functions with biased binary hypercube inputs, and therefore would learn decision trees in the smoothed complexity setting.

(Note: this question is of interest elsewhere in the literature. After our initial submission, we became aware of [Ref 2] which leaves as an open problem in Section 1.3 whether neural networks can learn $\log(n)$-juntas in a smoothed complexity setting, which are a special case of $\log(n)$-depth decision trees.)

**2. Sparse biased parities**: it was recently shown [Ref 4​​] that sparse biased parities can be learned with (S)GD on neural networks of polynomial size with an “emulation architecture” - where specifically designed initialization and architecture emulates a specifically designed statistical query algorithm via. SGD - which is far away from the type of “regular” architectures used in practice. However, learning sparse-biased-parities is an important problem as it was also shown in [Ref 3] that sparse-biased-parities cannot be learned (to any nontrivial loss) by any kernel method with polynomially many features. So a natural question is: can we also learn sparse-biased-parities with descent algorithms on neural networks that are more regular than [Ref 4], in particular, that are i.i.d. in their layers (a property that the current emulation networks do not have). The connection with staircases is now simple: it turns out that sparse biased parities satisfy the staircase property over biased inputs, as defined in Section 2 (for the same reason as in the previous point, since sparse biased parities are a special case of $\log(n)$-depth decision trees). Our paper shows that one can learn such staircases with regular networks in the unbiased case. We will also add a discussion on how our result can be extended to this biased setting.

**3. Necessity**: Finally, we believe that some notion of staircase is also ‘necessary’ in order for descent algorithms to learn functions that are sparse polynomials on the Boolean hypercube. For instance, we conjecture that in the unbiased hypercube truncated staircases starting from a $\log(n)$-degree monomial, such as $x_1\dots x_{\log(n)}+ \ldots +x_1\dots x_n$, are not learnable from regular networks in $\mathsf{poly}(n)$ time, due to the fact that the network will not be able to learn the lowest-degree monomials and thus to “climb” the staircase. However, this is still a class of functions that is learnable by SQ. Thus it is also plausible that a modification of the staircase class identifies the class of sparse polynomials that you can learn with a regular network, and this can be strictly less than the SQ-learnable class of functions (which are those learnable from non-regular emulation networks). In that sense, we believe that staircases are relevant to understanding deep learning, since deep learning succeeds with fairly blackbox/regular architectures (as also illustrated with our empirical results). We will add discussion about this conjecture in the revision.

### Regarding the model and training algorithm
We have stated the limitations of architecture clearly in Lines 281-288 the manuscript, and we believe it is a very interesting open problem to overcome these. We would like to emphasize the fact that using homogeneous initialization and architecture is a very important conceptual step in understanding hierarchical learning. The carefully-crafted architecture and initialization for simulating SQ learners considered in prior works essentially simulate the computation and memory component of any learning algorithm designed to solve this specific problem -- thus, we see our result as a first stepping stone to understanding more standard architectures and we believe that our architecture captures real world neural network phenomena more accurately than these prior works that rely on emulation arguments. We intend to analyze more standard neural network architectures in future works and for the aforementioned reasons believe that the present paper is an important first step.



Refs:

[Ref 1]  Kalai, Samorodnitsky, and Teng - “Learning and Smoothed Analysis”, FOCS 2009.

[Ref 2] Brutzkus, Daniely, and Malach - “ID3 Learns Juntas for Smoothed Product Distributions”, COLT 2020.

[Ref 3] Malach, Kamath, Abbe and Srebro - “Quantifying the Benefit of Using Differentiable Learning over Tangent Kernels”, ICML 2021.

[Ref 4] ​​Abbe, Kamath, Malach, Sandon, and Srebro - “On the Power of Differentiable Learning versus PAC and SQ Learning”, https://arxiv.org/pdf/2108.04190.pdf 2021.

---

### Decision · Program_Chairs · 2021-09-27

**Decision:**

Accept (Poster)

**Comment:**

This paper studies hierarchical structures that can be efficiently learned by neural networks to formalize the intuition that deep learning works by building feature hierarchies. It was shown that a class of sparsely connected networks trained in a neuron-wise and layer-wise manner can learn hierarchical boolean functions as defined in the paper. As all the reviewers agree, the paper contains strong theoretical results towards this direction and I recommend acceptance.

However, please note that the writing and notation can be improved to make the paper more accessible for a wider audience. Please take into account the updated reviews when preparing the final version to accommodate the requested changes including the necessary clarifications and typos pointed out by the reviewers.

An additional typo/notation issue: In section 3.2, the regularized loss \ell_R is defined as the population loss. However, in the gradient calculation step of Algorithm 2, \ell_R is evaluated at a batch of data points. This seems like a notational error.